



# A near-global multiyear climate data record of the fine-mode and coarse-mode components of atmospheric pure-dust

Emmanouil Proestakis[1], Antonis Gkikas[2], Thanasis Georgiou[1,3], Anna Kampouri[1,4], Eleni Drakaki[1,5], Claire L. Ryder[6], Franco Marenco[7,8], Eleni Marinou[1], Vassilis Amiridis[1]

[1]Institute for Astronomy, Astrophysics, Space Applications and Remote Sensing, National Observatory of Athens, Athens, Greece, 15236.
[2]Research Centre for Atmospheric Physics and Climatology, Academy of Athens, Athens, Greece.
[3]School of Physics, Faculty of Sciences, Aristotle University of Thessaloniki.
[4]Department of Meteorology and Climatology, School of Geology, Aristotle University of Thessaloniki, Thessaloniki, Greece.
[5]Harokopion University of Athens (HUA), Department of Geography, Athens, 17671.
[6]Department of Meteorology, University of Reading, Reading, RG6 6BB, UK.
[7]The Cyprus Institute, 20 Konstantinou Kavafi St., 2121, Aglantzia, Nicosia, Cyprus.
[8]Formerly at the Met Office, Fitzroy Road, Exeter, Devon, EX1 3PB, United Kingdom.

*Correspondence to*: Emmanouil Proestakis (proestakis@noa.gr)

## Abstract

A new four-dimensional, multiyear, and near-global climate data record of the fine-mode (submicrometer in terms of diameter) and coarse-mode (supermicrometer in terms of diameter) components of atmospheric pure-dust, is presented. The separation of the two modes of dust in detected atmospheric dust layers is based on a combination of (1) the total pure-dust product provided by the well-established European Space Agency (ESA) - "LIdar climatology of Vertical Aerosol Structure" (LIVAS) database and (2) the coarse-mode component of pure-dust provided by the first-step of the two-step POlarization LIdar PHOtometer Networking (POLIPHON) technique, developed in the framework of European Aerosol Research Lidar Network (EARLINET). Accordingly, the fine-mode component of pure-dust is extracted as the residual between the LIVAS total pure-dust and the coarse-mode component of pure-dust. Intermediate steps involve the implementation of regionally-dependent lidar-derived lidar-ratio values and AErosol RObotic NETwork (AERONET) based climatological extinction-to-volume conversion factors, facilitating conversion of dust backscatter into extinction and subsequently extinction into mass concentration. The decoupling scheme is applied to Cloud-Aerosol Lidar and Infrared Pathfinder Satellite Observations (CALIPSO) observations at 532 nm. The final products consist of the fine-mode and coarse-mode of atmospheric pure-dust, of quality-assured profiles of backscatter coefficient at 532 nm, extinction coefficient at 532 nm, and mass concentration for each of the two components. The datasets are established primarily with the original L2 horizontal (5 km) and vertical (60 m) resolution of Cloud-Aerosol Lidar with Orthogonal Polarization (CALIOP) along the CALIPSO orbit-path, and secondly in averaged profiles of seasonal-temporal resolution, 1°×1° spatial resolution, and with the original vertical resolution of CALIPSO, focusing on the latitudinal band extending between 70°S and 70°N and covering more than 15-years of Earth Observation (06/2006-12/2021). The quality of the dust products is justified by using AERONET fine-mode and coarse-mode aerosol optical thickness (AOT) interpolated to 532 nm and AERosol properties – Dust (AER-D) campaign airborne in-situ particle size distributions (PSDs) as reference datasets, during atmospheric conditions characterized by dust presence. The near-global fine-mode and coarse-mode pure-dust climate data record is considered unique with respect to a wide range of potential applications, including climatological, time-series, and trend analysis over extensive geographical domains and temporal periods, validation of atmospheric dust models and reanalysis datasets, assimilation activities, and investigation of the role of airborne dust on radiation and air quality.



## 1. Introduction

Mineral dust particles dispersed in the atmosphere play a key role in the Earth's radiation budget, climate system, environmental conditions, and human health. Constituting a major component of the global aerosol mass burden (Gliß et al., 2021; Kok et al., 2017), airborne dust perturbates through scattering and absorption of solar shortwave and longwave radiation the Earth's radiative budget (Tegen et al., 1996; Ramanathan et al., 2001; Adeyemi and Kok, 2020; Ito et al., 2021), an instantaneous process known as the direct radiative effect (Sokolik and Toon, 1996). Moreover, depending on the chemical composition and atmospheric conditions, dust aerosols serve as effective Cloud Condensation Nuclei (CCN; Hatch et al., 2008) and/or Ice Nuclei (IN; DeMott et al., 2009; Marinou et al., 2019). Through the induced indirect and the semi-direct effects on the radiation balance, airborne dust aerosols modify clouds' microphysical, macrophysical, and optical properties (e.g., albedo), precipitation patterns, atmospheric stability, cloud formation, lifetime, and coverage (Twomey, 1977; Albrecht, 1989; Rosenfeld et al., 2008), with adverse effects on weather and eventually climate (Haywood and Bucher, 2000; Huang et al., 2006). However, the impact of atmospheric dust extends beyond the Earth's energy balance. The aeolian transport of dust particles over large distances (van der Does et al., 2018; Drakaki et al., 2022) uniquely influences both marine and terrestrial ecosystems. Through wet and dry deposition of mineral nutrients, iron, and phosphorus (Okin et al., 2004), dust regulates oceanic productivity, affects the ecosystems' biogeochemical cycles, and in addition the carbon dioxide budget (Jickells et al., 2005; Li et al., 2018). Dust is related to a wide range of anthropogenic economic activities, including among others, agriculture (Stefanski and Sivakumar, 2009), solar energy production (Kosmopoulos et al., 2018; Masoom et al., 2021; Papachristopoulou et al., 2022), and aviation safety (Papagiannopoulos et al., 2020). Finally, atmospheric dust and human health are closely linked. More specifically, depending on Particles' Size Distribution (PSD) and mass concentration over inhabited areas and within the Planetary Boundary Layer (PBL), airborne dust is associated with degradation of air quality (Kanakidou et al., 2011, Dione et al., 2022) and induced negative disorders on human health (Du et al., 2015).

The intensity of these effects depends strongly on the complex nature of mineral dust, related to large uncertainties not fully determined and known. According to the Intergovernmental Panel on Climate Change - Fourth Assessment Report (IPCC AR4, 2007; Foster et al., 2007), the "natural variability" of aerosols is a significant factor of uncertainty in climate change predictions. Despite the considerable progress by the scientific community in observing and modelling climate-relevant aerosol properties in the following years (IPCC AR5, 2014) and until nowadays (Forster et al., 2021), the overall aerosol uncertainties, although better quantified and of improved confidence level, remain high. Moreover, in addition to natural dust the anthropogenic dust is estimated to contribute about 25% to the global atmospheric dust load (Ginoux et al., 2012), a component that consists an additional important source of uncertainty.

Towards reducing these uncertainties, proper consideration and better understanding of the different contributing factors of dust life circle, from dust emission to transport and eventually deposition, is required. Regarding mineral dust natural mobilization mechanisms, they involve dust devils (Koch and Renno, 2005), "haboobs" (Knippertz et al., 2007), pressure gradients (Klose et al., 2010) and low-level jets (LLJ; Fiedler et al., 2013), developed over hyper-arid, arid and semi-arid regions (Fig.1) of easily erodible dry soils or areas of little vegetation (Prospero et al., 2002), triggering dust emission and suspension into the atmosphere (Marticoréna, 2014). Uncertainties related to emission and mobilization mechanisms, inhomogeneous both in time and space and of variable production strength (Knippertz et al., 2009, 2011), propagate into our understanding of mineral dust role in the climate system, environmental conditions, and human health. In addition, mineral dust originating from different regions is characterized by substantially different chemical composition (Krueger et al., 2004), thus of different scattering and absorption properties (Müller et al., 2007a; Nisantzi et al., 2015). Therefore, to better assess, understand, and model the complex role of atmospheric dust in the climate system and its impact on the environment, accurate information on the highly variable temporal evolution and three-dimensional distribution and of dust is required, with particular focus on profiling.



Of particular interest is the fact that dust-associated properties are associated to the particle size distribution. Upon entering the Free-Troposphere the lifetime of atmospheric dust transport highly depends on the particles' size, with coarse mineral particles more efficiently removed through dry deposition (e.g., gravitational settling) close to the source regions (Schepanski et al., 2009b) and fine dust particles more prominent to long-range transport, prior their removal via dry deposition or wet scavenging (Ginoux et al., 2004). Recent studies, however, stress the still not fully understood impact of dust size distribution on transport, the longer atmospheric lifetime of giant dust particles prior to removal (van der Does et al., 2018; Drakaki et al., 2022), and the substantial underestimation of dust transport range simulated by state-of-the-art climate models (Adebiyi and Kok, 2020). Dust vertical distribution, dust particle size-distribution, and dust transport are intimately interlinked and a poor characterization of one of these aspects in a model has direct repercussions on the other two: for this reason, spaceborne datasets of the vertical dust distribution can fill a crucial observational gap (O'Sullivan et al, 2020). Moreover, dust optical depth is controlled by both the fine-mode and coarse-mode dust components, while at the same time, optical properties, such as scattering and linear particle depolarization ratio, are also subject to the size distribution (Sakai et al., 2010; Järvinen et al., 2016). Regarding the Earth's energy balance, not all dust modes contribute to aerosol radiative forcing in the same way, in sign and magnitude, with warming and cooling effects reported for dust particles larger and smaller than 5 μm in diameter, respectively (Miller et al., 2006; Kok et al., 2017). Furthermore, coarse dust particles act more efficiently as CCN and/or IN than fine-mode dust particles (DeMott et al., 2009; Adebiyi et al., 2023), while the effect of dust on health is widely controlled by the fine-mode of dust particles (Goudie et al., 2014). To address these multiple uncertainties, observing, monitoring, modelling, and quantifying the spatial, vertical, and temporal distribution of mineral dust suspended in the atmosphere, with the potential to further distinguish between fine-mode and coarse-mode, over extended regions and temporal periods, is an important step towards a more realistic understanding of the complex role on dust in Earth's system and human health, and towards better constrains in prediction models.

Light detection and ranging (lidar) is among the most prominent and powerful techniques for remote sensing of the atmosphere, able to provide the vertical structure of the aerosol field and related optical properties at high vertical resolution. In particular, lidar systems employing polarization measurements greatly contribute to our knowledge of atmospheric dust, as irregular particles perturb the polarization state of lidar-emitted polarized light pulses (Freudenthaler et al., 2009). Moreover, when individual ground-based multiwavelength-Raman-polarization lidars are assembled under a network architecture, the on-parallel operation greatly expands the capacity of aerosol remote sensing, both in time and space, on a regular basis, or even continually (e.g., Ansmann et al., 2003; Amiridis et al, 2005; Mona et al., 2006; Mattis et al., 2008; Papayannis et al., 2008). Motivated by the lidar capability of profiling aerosol optical properties with high vertical resolution and the distinct signature of non-spherical dust particles on the particle linear depolarization ratio, comprehensive efforts have been made to implement lidars to develop sophisticated methodologies for identifying, and accordingly decoupling, the pure-dust component from the total atmospheric aerosol load. To date, the Lidar-Radiometer Inversion Code (LIRIC; Chaikovsky et al., 2016) and the Generalized Aerosol Retrieval from Radiometer and Lidar Combined data (GARRLiC; Lopatin et al., 2013) algorithms implement coincident elastic-backscatter lidar measurements synergistically with the sun–sky-scanning radiometer observations obtained by the global Aerosol Robotic Network (AERONET; https://aeronet.gsfc.nasa.gov/; last access: 03/07/2023; Holben et al., 1998) for the retrieval of vertically resolved aerosol properties, including differentiation between the fine-mode and coarse-mode contributions. In addition to LIRIC and GARRLiC algorithms and under the EARLINET efforts of aerosol characterization, a polarization-based algorithm for decoupling the atmospheric pure-dust component from the total aerosol load has been developed, initially established by Shimizu et al. (2004) and accordingly expanded through the one-step polarization Lidar Photometer Networking (one-step POLIPHON; Tesche et al., 2009; Ansmann et al., 2012) advancements. The technique consists of a stand-alone lidar polarization-dependent approach capable of decoupling the pure-dust and non-dust aerosol components, during both daytime and nighttime illumination conditions, and even under the presence of thin clouds. Moreover, a multistep extension of the one-step POLIPHON, namely the two-step POLIPHON, allows for





further decoupling between the fine-mode and coarse-mode components of atmospheric dust (Mamouri and Ansmann, 2014; 2017), based on the observation that these two components of the total dust load have distinct characteristic particle depolarization ratio properties.

Observation and characterization of atmospheric dust highly depends on lidar systems employing polarization measurements. To date, a significant number of polarization lidar systems have been deployed for both services and aerosol research purposes to permanent locations around the globe, frequently operating as integral components of ground-based lidar networks. Presently, lidar networks of continental-scale contributing to the dust observational efforts include, among others, the pioneer European Aerosol Research Lidar Network (EARLINET; www.earlinet.org/; last access: 03/07/2023; Pappalardo et al. 2014), PollyNET (http://polly.tropos.de/; last access: 03/072023; Baars et al., 2016), the (east) Asian Dust and Aerosol Lidar Observation Network (AD-NET; https://www-lidar.nies.go.jp/AD-Net/; last access: 03/07/2023; Shimizu et al. 2017), the Latin America Lidar Network (LALINET; http://lalinet.org/; last access: 03/07/2023; Antuna-Marrero et al., 2017), the Micro-Pulse Lidar Network (MPLNET; http://mplnet.gsfc.nasa.gov; last access: 03/07/2023; Welton et al. 2001) and NOAA's Cooperative Remote Sensing Science and Technology Lidar Network (CREST-CLN; https://www.cessrst.org/about/facilities/crest-lidar-network; last access: 03/07/2023). The ground-based lidar networks are further organized as a network of lidar networks under the Global Atmosphere Watch (GAW) Aerosol Lidar Observation Network (GALION). However, harmonized global vertically-resolved observation of the atmospheric aerosol components collected via ground-based lidar systems still remains challenging. Among the significant challenges are the different instrumental designs, the manually researcher/user-dependent operation (e.g. EARLINET three times per week operation; Pappalardo et al., 2014), the need for 24/7 continuous provision of atmospheric observations (e.g. PollyNET; Baars et al., 2016; Engelmann et al., 2016), and the challenge of automatic data processing chains and quality-assurance and calibration procedures to harmonize lidar measurements (e.g. efforts towards the EARLINET Single Calculus Chain; D'Amico et al., 2015). Furthermore, due to the Earth's surface, being ~70% covered by water, the geographical coverage of deployed lidar systems is mainly over land, and low.

To date, the challenges of ground-based lidar networks have been addressed with spaceborne lidar systems, and especially by CALIOP (Cloud–Aerosol Lidar with Orthogonal Polarization), the primary instrument onboard the satellite CALIPSO (Cloud–Aerosol Lidar and Infrared Pathfinder Satellite Observation, Winker et al., 2009). More specifically, CALIPSO provided multi-year observations of aerosol and cloud optical properties (i.e., attenuated backscatter and volume depolarization ratio at 532 nm), operating on a near-global scale (i.e., 82ºS and 82ºN) and nearly continuously, between June 2006 and August 2023. As such CALIPSO has provided an unprecedented long-term Earth Observation (EO) dataset of atmospheric aerosols. In addition, due to the distinct signature of dust on particulate depolarization ratio (Gobbi et al., 2000; Freudenthaler et al., 2009), the long-term CALIPSO vertically-resolved polarization measurements have allowed for global monitoring, and quantification of the horizontal, vertical, and temporal distribution of the mineral dust aerosol component (Amiridis et al., 2013; Marinou et al., 2017; Proestakis et al., 2018).

Here we present the first attempt to adapt and apply the general concept of the two-step POLIPHON method to CALIOP polarization lidar measurements. The study is motivated by laboratory studies reporting on the distinct light-depolarizing properties of fine-mode and coarse-mode dust (Sakai et al., 2010; Järvinen et al., 2016), accordingly expanded in the framework of EARLINET (Mamouri and Ansmann, 2014; 2017). Overarching objective of the present study consists of the separation of the pure-dust submicrometer (fine-mode) and supermicrometer (coarse-mode) components of the dust aerosol load, in order to provide an accurate near-global and multiyear description of (1) the temporal distribution, (2) the three-dimensional spatial and vertical distribution, and (3) the seasonal and spatial transition of fine/coarse-mode dust transport pathways in terms of range, height and intensity (Fig.1). Moreover, this study aims to contribute to the next generation of dust air quality geo-information products, with the overarching objective to advance our EO-based capacity to provide the PBL fine-mode component of dust at a near-global scale and over long-term periods.



The paper is organized as follows. Section 2 provides a description of the implemented datasets (Sect. 2.1) and an overview of the applied methodology (Sect. 2.2) in order to realize the overarching technical and scientific objectives of the study. Section 3 provides justification of the validity of the fine-mode and coarse-mode components of pure-dust, on the basis of long-term AERONET observations (Sect. 3.1) and airborne in-situ measurements (Sect. 3.2) as reference datasets. Section 4

provides an overview of the four-dimensional (4D) reconstruction of the atmosphere in terms of fine-mode and coarse-mode pure-dust components, at a near-global scale and based on more than 15-years of EO. Finally, Section 5 provides a summary of the study along with the main concluding remarks.

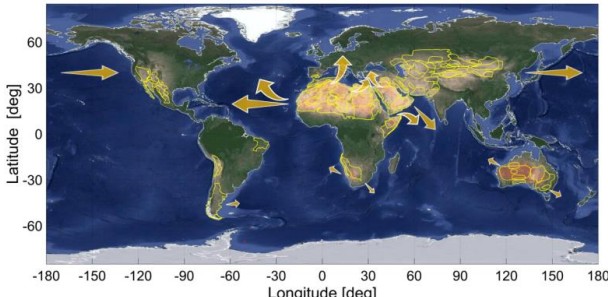

Figure 01: Hyper-arid, arid and semi-arid regions of easily erodible dry soils or areas of little vegetation (areas in yellow colour) and main aeolian dust transport pathways (source: Natural Earth Data: https://www.naturalearthdata.com/; last access: 03/07/2023).

## 2 Datasets and methodology

### 2.1 Datasets

The following sub-sections discuss the datasets implemented in the framework of the study, as to facilitate the realization of its overarching objectives. More specifically, Sect.2.1.1. provides an overview of the CALIPSO-CALIOP mission and

products, cornerstone of the near-global fine-mode and coarse-mode pure-dust climate data record. In addition, towards establishing the accuracy of the products and consistency checks, the study utilizes ISS-CATS optical products (Sect.2.1.2.) and AERONET retrievals (Sect.2.1.3.).

### 2.1.1 CALIPSO-CALIOP

The Cloud-Aerosol Lidar and Infrared Pathfinder Satellite Observation (CALIPSO) mission (Winker et al., 2010) was a joint satellite project, developed, operated, and maintained in collaboration between the National Aeronautics and Space Administration (NASA), the United States space agency, and the Centre National D'Études Spatiales (CNES), the French space agency. The satellite CALIPSO was launched on April 28[th], 2006, and integrated in the Afternoon-Train (A-Train)

constellation of sun-synchronous polar-orbit satellites (Stephens et al., 2018), hosting a suite of three Earth-Observing instruments, in a near-nadir-looking configuration: a single channel 645 wide field-of-view camera (WFC), a three channel (8.65, 10.6, 12.05 µm) Imaging Infrared Radiometer (IIR; Garnier et al., 2017), and the primal payload, the Cloud-Aerosol Lidar with Orthogonal Polarization (CALIOP) lidar (Hunt et al., 2009). CALIOP was a dual-wavelength polarization-sensitive elastic backscatter Nd:YAG lidar, capable of transmitting linear polarized light pulses at 532 and 1064 nm, and performing

range-resolved measurements of the backscattered signals by atmospheric features, and specifically, of the parallel and perpendicular components of the backscattered photons at 532 nm with respect to the polarization plane of CALIOP emitted beam, and the total backscatter intensity at 1064 nm (Winker et al., 2009).



CALIOP measurements and products are provided in different levels of processing. The received measurements of attenuated backscatter from molecules and particles are provided in 1/3 km horizontal and 30 m vertical resolution and reported in CALIOP Level 1 (L1). Subsequently, CALIOP L1 measurements are processed to CALIOP Level 2 (L2) products, following a sophisticated chain of algorithms (Winker et al., 2009) that performs a sequence of functions, including the fundamental for

the retrievals daytime and nighttime calibration of the three receiver channels (Powell et al., 2009; Getzewich et al., 2018; Kar et al., 2018; Vaughan et al., 2019), layer detection (Vaughan et al., 2009) and cloud-aerosol discrimination (Liu et al., 2009; Liu et al., 2019; Zeng et al., 2019). In addition, in the process of retrieving particulate extinction profiles (Young and Vaughan, 2009), an intermediate aerosol classification and lidar ratio (LR) selection algorithm for feature detection classifies atmospheric features between "clear-air", "tropospheric aerosol", "stratospheric aerosol", "cloud", "surface", "subsurface", "totally

attenuated", and aerosol and cloud features of "low/no confidence". The algorithm modules further classify atmospheric features categorized as "tropospheric aerosol" between "marine", "dust", "polluted continental/smoke", "clean continental", "polluted dust", "elevated smoke" and "dusty marine" (Omar et al., 2009; Kim et al., 2018), and in the case of "stratospheric aerosol" between "PSC aerosol", "volcanic ash", and "sulfate/other" (Kar et al., 2019).

       CALIOP L2 profiles of aerosols (APro) and clouds (CPro) provided continuous, vertically resolved measurements of optical

and geometrical properties of atmospheric features, detected along the CALIPSO orbit path on a near-global at uniform 5 km horizontal and 60 m vertical resolution over the altitude range from −0.5 to 20.2 km, and 180 m from 20.2 to 30 km height a.m.s.l. In this study, we use CALIPSO Version 4.2 (V4. 2) L2 profiles of altitude-resolved aerosol backscatter coefficient and particulate depolarization ratio at 532 nm, profile descriptors (e.g., longitude, latitude, time), the provided quality-assurance flags (e.g., Cloud-Aerosol-Discrimination – CAD – score), and the assigned atmospheric classification products,

between June 2006 and December 2021, to develop the CALIPSO-based three-dimensional multi-year global fine-mode and coarse-mode pure-dust products.

       The quality screening procedures used here to generate the quality-assured CALIPSO-based Lever 2 and Level 3 fine-mode and coarse-mode pure-dust aerosol products follow the quality control procedures used to generate the official CALIPSO Level 3 aerosol products (Winker et al. 2013; Tackett et al., 2018) and subsequent developments (Amiridis et al., 2013; Marinou et

al., 2017). More specifically, the quality screening methods are initially applied to CALIOP L2 backscatter coefficient profiles at 532 nm prior to decoupling the pure-dust, fine-mode pure-dust, and coarse-mode pure-dust components from the total aerosol load. The approach is conservative, weighting the removal of a significant number of erroneous features and retrievals over preservation of the dataset, to avoid introducing inconsistencies, weighting effects, and unrealistic shape of profiles due to significant reduction of the data set.

With respect to quality assurance procedures, the most aggressive quality control check is considered the cloud-free condition, applied in an entire profile-removal approach when cloud features are detected at CALIPSO profile level at 5 km, resulting in minimizing detection, classification and retrieval errors, and eventually avoiding attenuation and weighting effects (Tackett et al., 2018). Moreover, in the process of quality screening controls, backscatter coefficient of atmospheric features classified as "clear-air" is assumed equal to $0.0$ km$^{-1}$sr$^{-1}$. Aerosol layers detected at 80 km horizontal averaging resolution, due to low signal-

to-noise ratio (SNR), and not in contact with other quality-assured aerosol layers are rejected. To account for clouds misclassified as aerosol, and vice versa, aerosol features of CAD score in the range [-100,-20] are accepted, rejecting aerosol layers of CAD score outside this range due to high probability of erroneous feature classification. In addition, aerosol layers above 4 km a.m.s.l., adjacent to ice clouds of Top Temperature below 0 ºC are also rejected as cirrus fringes misclassified as aerosols. The series of quality assurance procedures include rejection of the backscatter coefficient at 532 nm in cases of low

quality in the retrieval of the corresponding extinction coefficient at 532 nm profiles. Level 2 aerosol features of extinction QC flags not equal to 0 (Lidar Ratio unchanged), 1 (Lidar Ratio measured), 16 (layer is opaque and the Lidar Ratio value unchanged), or 18 (layer is opaque and the Lidar Ratio value is reduced) are rejected, while in terms of random and systematic errors, aerosols feature of extinction uncertainty less or equal to 99.99 km$^{-1}$ are also rejected. Finally, the sequence of



backscatter coefficient at 532 nm quality assurance controls accounts for large signal anomalies in cases of surface-attached aerosol layers, reporting either significant negative (less than -0.2 km$^{-1}$) or large positive (higher than 2.0 km$^{-1}$) extinction coefficient at 532 nm, within 60 m a.g.l., are removed. Overall, the quality filtering methods and control procedures (Table 1) are applied to counteract and reduce the impact of noise and of clouds misclassified as aerosols, systematic and random errors

and artifacts, and retrieval issues, while at the same time affecting CALIPSO aerosol profiles by the smallest amount possible, and maintaining a high-quality extended dataset, suitable for study cases and longer-scale studies. Accordingly, the sequence of quality assurance procedures iterates through all CALIOP L2 cloud-free profiles to generate the fine-mode and coarse-mode pure-dust products along the CALIPSO orbit path.

Table 1: Quality control procedures and filtering applied in CALIPSO data.

| Quality Assurance procedures |
|---|
| 1  Screen out all cloud features. |
| 2  Aerosol extinction coefficient for "clear air" assigned equal 0.0 km$^{-1}$. |
| 3  Screen out atmospheric features of CAD score outside the range [-100, -20]. |
| 4  Screen out atmospheric features of Extinction QC flag ≠ 0, 1, 16 and 18. |
| 5  Screen out atmospheric features of aerosol extinction uncertainty ≤ 99.9 km$^{-1}$. |
| 6  Screen out misclassified cirrus fringes. |
| 7  Screen out isolated aerosol features of horizontal resolution 80 km. |
| 8  Features of large negative extinction coefficient values ≤ -0.2 km$^{-1}$, detected ≤ 60 m a.g.l., are removed. |
| 9  Features of large positive extinction coefficient values ≥ 2.0 km$^{-1}$, detected ≤ 60 m a.g.l., are removed. |
| 10  "Clear-Sky" Mode |

### 2.1.2 ISS-CATS

The Cloud-Aerosol Transport System (CATS) was a multiwavelength (355, 532, and 1064 nm) lidar system developed at NASA's Goddard Space Flight Center, operated as a scientific payload to the Japanese Experiment Module–Exposed Facility (JEM-EF) on the International Space Station (ISS). Due to technical issues, CATS operated primarily on Mode 2 (forward Field Of View – FFOV) towards acquiring near-real-time profile measurements of attenuated total backscatter and linear volume depolarization ratio at 1064 nm in the Earth's atmosphere (Yorks et al., 2016). CATS range-resolved observations of

aerosol and cloud optical properties along the ISS orbit track extended between the 10th of February, 2015 and the 30th of October, 2017, when the system suffered an unrecoverable power failure. CATS products are provided in different levels of processing. McGill et al. (2015) and Yorks et al. (2016) provide a comprehensive overview of CATS instrument and scientific goals, while CATS processing algorithms and validation of the L1 and L2 products are comprehensively provided by Yorks et al. (2016), Pauly et al. (2019), and Proestakis et al. (2019), as well as in the CATS Data Release Notes, Quality Statements

and Theoretical Basis documentation (https://cats.gsfc.nasa.gov/; last access: 23/06/2023). In this study, we use CATS Version 3.01 (V3.01) L2 profiles of altitude-resolved backscatter coefficient and particulate depolarization ratio at 1064 nm, including the available quality-assurance flags and atmospheric classification products (i.e., feature type, aerosol subtype, classification confidence; Yorks et al., 2019), provided with vertical and horizontal resolution of 60 m and 5 km (along-track) respectively. Table 2 provides the procedures applied to generate quality-assured profiles of backscatter coefficient and particulate

depolarization ratio at 1064 nm. In the framework of the study, CATS is utilized to demonstrate the performance of the methodology when applied to a satellite-based lidar system, and to establish the performance of the fine-mode and coarse-mode of pure-dust products in terms of mass concentration, against highly collocated airborne remote sensing and in-situ measurements conducted on the 7th of August 2015 (Sect.3.2).





Table 2: Quality control procedures and filtering applied in CATS data.

| Quality Assurance procedures |
| --- |
| 1 Screen out all cloud features. |
| 2 Backscatter coefficient and extinction coefficient for "clear air" assigned equal to 0.0 $km^{-1}sr^{-1}$ and $km^{-1}$. Particulate depolarization ratio for "clear air" assigned missing values (NaN). |
| 3 Screen out atmospheric features of classification confidence (Feature Type Score) outside the range [-10, -2]. |
| 4 Screen out atmospheric features of Extinction QC flag = 4, 8, or 9. |
| 5 Screen out atmospheric features of particulate depolarization range outside the nominal range [0, 1]. |
| 6 Screen out cloud contaminated profiles (Sky Condition 2 or 3). |
| 7 Features of large negative backscatter coefficient values $\leq$ -0.002 $km^{-1}sr^{-1}$ are removed. |
| 8 Features of large positive backscatter coefficient values $\geq$ 0.02 $km^{-1}sr^{-1}$ are removed. |
| 9 "Clear-Sky" Mode. |

### 2.1.3 The AERONET product

The validity of the applied methodology on the spaceborne retrievals for the decomposition of the total dust load to its size-related (i.e., fine and coarse) components is justified by utilizing the AERONET optical products as reference. Ground-based measurements suitable for the purposes of the current study are those derived via the Spectral Deconvolution Algorithm (SDA; O'Neill et al., 2001a; 2001b; 2003), initially implemented by Eck et al. (1999). In principle, SDA utilizes the sun-direct measurements of aerosol optical thickness (AOT) and Ångström exponent and it tries to reproduce, via an iterative process, the parameters of the log-normal aerosol-speciated size distribution which gives the best agreement between modeled and measured AOTs. The two primary SDA products are the fine and coarse AOT at 500 nm, which along with the Ångström exponent (expressing the spectral variation of AOT) have been processed for deriving the corresponding AOTs at 532 nm (CALIOP-CALIPSO). In the SDA, the decomposition of total AOT to its fine and coarse counterparts is defined optically under the assumption of a bi-modal aerosol particle size distribution (PSD) and the approximation of a neutral coarse mode spectral variation. In our study, we analyze the Level 2.0 data (quality assured and cloud screened) from the most updated version (Version 3; Giles et al., 2019; Sinyuk et al., 2020). Moreover, we use the AERONET data stored in the "All Points" files thus making feasible the optimum temporal collocation between ground-based and spaceborne retrievals. At each AERONET site we define a circle of 80 km radius and we average the CALIOP-CALIPSO vertical profiles of the extinction coefficient residing within the area. Then, the ground-based retrievals acquired within a time-window of 60 minutes centered at the satellite overpass time are averaged in temporal terms. In the framework of the study, AERONET retrievals of fine-mode and coarse-mode AOTs converted at 532 nm are utilized to evaluate the CALIPSO-based fine-mode and coarse-mode Dust Optical Depths (DODs) at 532 nm products (Sect.3.1).

### 2.2 Methodology

The present work aims to decouple the fine-mode (particles with diameter less than 1 μm) and coarse-mode (particles with diameter greater than 1 μm) components of atmospheric pure-dust, which is in turn is component of the total aerosol mixture, at a near-global scale. The decoupling methodology follows the series of the well-established polarization-based algorithms for decoupling the atmospheric pure-dust component from the total aerosol load, initially established by Shimizu et al. (2004) and accordingly expanded through the family of the POLIPHON algorithms. More specifically, the study is based on the one-step POLIPHON (Tesche et al., 2009) methodology for decoupling the pure-dust component from the total aerosol load (Shimizu et al., 2004), and accordingly on the conceptual approach of the two-step POLIPHON (Mamouri and Ansmann, 2014) for extracting the coarse-mode pure-dust component from the total aerosol load. Finally, in the framework of the present study, the submicrometer (fine-mode) component of pure-dust is extracted as the residual between the total pure-dust (one-step POLIPHON) and the supermicrometer (coarse-mode) component of pure-dust (first-step of the two-step POLIPHON).





The retrieval scheme is applicable to single-wavelength lidar observations, as long as profiling of calibrated linear-polarization is included. The methodology is applied to CALIPSO backscatter coefficient and particulate depolarization ratio profiles at 532 nm (Sect.2.1.1.), with the overarching objective to provide the fine-mode and coarse-mode pure-dust atmospheric components at near-global scale and for the temporal period extending between 06/2006 and 12/2021. Sect.2.2.1. presents the

5  decoupling methodology of the fine-mode and coarse-mode components of pure-dust, in terms of backscatter coefficient at 532 nm, extinction coefficient at 532 nm, and mass concentration, while sect.2.2.3. discusses the uncertainties of the established products.

**2.2.1 Pure-Dust, Coarse-Dust and Fine-Dust Backscatter Coefficient profiles at 532 nm**

The algorithm applied to decouple an external aerosol mixture of particles with distinct depolarizing properties (e.g. dust and non-dust, with $\delta_{dust} > \delta_{non-dust}$), thoroughly discussed in Shimizu et al. (2004) and Tesche et al., (2009), starts from the equation for particle depolarization ratio - "$\delta_{\lambda,p}(z)$" (Eq. (1)) and the consideration that the backscattered signal by an external aerosol mixture in a lidar system "$\beta_{\lambda,p}(z)$" corresponds to the summation of the cross and parallel return signals from the

15  different aerosol types (Eq. (2)).

$$\delta_{\lambda,p}(z) = \frac{\beta_{\lambda,nd}^{\perp}(z) + \beta_{\lambda,d}^{\perp}(z)}{\beta_{\lambda,nd}^{\parallel}(z) + \beta_{\lambda,d}^{\parallel}(z)} \tag{1}$$

$$\beta_{\lambda,p}(z) = \beta_{\lambda,nd}(z) + \beta_{\lambda,d}(z) \tag{2}$$

In Eq. (1) the parameters "$\beta_{\lambda,k}^{\perp}(z)$" and "$\beta_{\lambda,k}^{\parallel}(z)$" (k = "$d$" for dust particles or "$nd$" for non-dust particles) correspond to the

20  cross and parallel backscatter coefficient components, respectively, of the two aerosol subtypes with different depolarizing optical properties, given as functions of wavelength "$\lambda$" and height "$z$". Based on Eq. (1) and Eq. (2), "$\beta_{\lambda,k}^{\parallel}(z)$" and "$\beta_{\lambda,k}^{\perp}(z)$" can be expressed as functions of the total particle backscatter coefficient - "$\beta_{\lambda,k}(z)$" and the corresponding particle depolarization - "$\delta_{\lambda,k}(z)$".

$$\beta_{\lambda,k}^{\parallel}(z) = \frac{\beta_{\lambda,k}(z)}{1 + \delta_{\lambda,k}(z)} \tag{3}$$

$$\beta_{\lambda,k}^{\perp}(z) = \frac{\beta_{\lambda,k}(z)\delta_{\lambda,k}(z)}{1 + \delta_{\lambda,k}(z)} \tag{4}$$

Through Eq. (3), Eq. (4), and considering "$\beta_{\lambda,nd}(z)$" as "$\beta_{\lambda,p}(z) - \beta_{\lambda,d}(z)$" in Eq. (1), the pure-dust backscatter coefficient component "$\beta_{\lambda,d}(z)$" is expressed by Eq. (5).

$$\beta_{\lambda,d}(z) = \beta_{\lambda,p}(z) \frac{\left(\delta_{\lambda,p}(z) - \delta_{\lambda,nd}(z)\right)\left(1 + \delta_{\lambda,nd}(z)\right)}{\left(\delta_{\lambda,d}(z) - \delta_{\lambda,nd}(z)\right)\left(1 + \delta_{\lambda,p}(z)\right)} \tag{5}$$

In Eq. (5), under the special cases of $\delta_{\lambda,p}(z) \leq \delta_{\lambda,nd}(z)$ and $\delta_{\lambda,p}(z) \geq \delta_{\lambda,d}(z)$, we set $\beta_{\lambda,d}(z) = 0$ and $\beta_{\lambda,d}(z) = \beta_{\lambda,p}(z)$, respectively, accounting for the cases of negligible and dominant contribution of dust in the total aerosol mixture, respectively. However, for a proper implementation of Eq. (5), and to facilitate accurate quantification of the atmospheric pure-dust aerosol

component, proper definition of the non-dust and dust light-depolarization characteristics, thus of "$\delta_{nd}$" and "$\delta_d$", is a prerequisite.

For dust "$\delta_d$", typical particle depolarization ratios of lofted dust-dominated aerosol layers, measured in the framework of field activities conducted in the proximity of the Sahara (Esselborn et al., 2009; Freudenthaler et al., 2009; Ansmann et al. 2011),

Middle East (Mamouri et al., 2013; Filioglou et al., 2020), and Asian (Sugimoto et al., 2003; Hofer et al., 2017) dust sources, show similar values ranging between 29% and 35%. These findings agree with lidar measurements of particle depolarization ratio of airborne dust, studied during mid-range and long-range transport across Europe (Wiegner et al., 2011; Baars et al., 2016), North Atlantic Ocean (Gross et al., 2011; Gross et al., 2015; Tesche et al., 2011; Veselovskii et al., 2016; Haaring et al., 2017), and over the Pacific Ocean (Sakai et al., 2003; Shimizu et al., 2004). The studies corroborate on the assumption that

desert dust is characterized by a particle depolarization ratio around 0.31 ± 0.04 at 532 nm, a characteristic property both close to the emission sources of dust and following long-range atmospheric transport.

However, accurate implementation of the pure-dust decoupling methodology (Eq. (5)) requires, in addition to dust depolarization features ("$\delta_d$"), proper consideration of the depolarization features of the non-dust aerosol subtypes composing the aerosol mixture ("$\delta_{nd}$"). Broader aerosol subtype categories include sea salt, biomass-burning smoke, pollen, and volcanic

ash. Regarding marine aerosol, the particle linear depolarization ratio increases from 2-3% at 532 nm for wet spherical sea salt particles in marine environment of high relative humidity to about 10 to 15% at 532 nm for dry cubic-like sea salt particles close to the Marine Boundary Layer (MBL) – Free Troposphere (FT) entrainment zone (Haaring et al., 2017). The presence of sea salt in the FT is considered negligible. Other aerosol subtypes frequently encountered both in the Planetary Boundary Layer (PBL) and the FT include urban haze and biomass-burning smoke, with depolarizing effects of 1-4% at 532 nm (Müller

et al., 2007b; Nicolae et al., 2013). The pollen aerosol category relates to depolarization ratio in the range of 4–6% at 532 nm, although in extreme cases of significantly large particles (diameter ≥ 50 μm) this effect may reach as high as 15% at 532 nm (Noh et al.; 2013). The presence of pollen is usually confined within the PBL and manifests high seasonality, with higher values evident during spring and during atmospheric convection conditions. Finally, a less frequently observed aerosol category is volcanic ash, with depolarization ratio effect ranging between 30 and 40% at 532 nm, as reported by EARLINET

observational activities in the case of Eyjafjallajökull in 2010 (Ansmann et al., 2010; Groß et al., 2012). Here, and based on the above discussion, for the non-dust aerosol subtypes category "$\delta_{nd}$" equal to 0.05 ± 0.02 at 532 nm is assumed (Tesche et al. 2009, Mamouri and Ansmann, 2014; 2016; Marinou et al., 2017; Proestakis et al., 2018).

To date, the pure-dust decoupling methodology (Shimizu et al., 2004; Tesche et al., 2009) has been applied towards a robust global pure-dust product, established in the framework of the ESA - LIdar climatology of Vertical Aerosol Structure for space-

based lidar simulation studies activity (LIVAS; Amiridis et al., 2015). The CALIPSO-based pure-dust product implements suitable geographically dependent extinction-to-backscatter ratios for dust aerosols, while its performance has been established against AERONET-collocated measurements in the Saharan Desert broader region (Amiridis et al., 2013). The ESA-LIVAS pure-dust product constitutes of a cornerstone dataset of the present study (Fig.2 – "1st step").

As extensively and thoroughly discussed in Mamouri and Ansmann (2014; 2017), two successive POLIPHONs are required

in order to extract the fine-mode and coarse-mode components of pure-dust. The two-step POLIPHON technique assumes that the backscattered signal by an external aerosol mixture "$\beta_{\lambda,p}(z)$" corresponds to the summation of the cross and parallel return signals from the non-dust, the fine-mode dust, and the coarse-mode dust aerosol components (Eq. (6)).

$$\delta_{\lambda,p}(z) = \frac{\beta_{\lambda,nd}^{\perp}(z) + \beta_{\lambda,df}^{\perp}(z) + \beta_{\lambda,dc}^{\perp}(z)}{\beta_{\lambda,nd}^{\parallel}(z) + \beta_{\lambda,df}^{\parallel}(z) + \beta_{\lambda,dc}^{\parallel}(z)} = \frac{\beta_{\lambda,ncd}^{\perp}(z) + \beta_{\lambda,cd}^{\perp}(z)}{\beta_{\lambda,ncd}^{\parallel}(z) + \beta_{\lambda,cd}^{\parallel}(z)}$$

(6)





In Eq. (6), "$\beta_{\lambda,ncd}(z)$" and "$\beta_{\lambda,cd}(z)$" correspond to the non-coarse-mode aerosol (i.e., non-dust and fine-mode dust) and the coarse-mode dust components of the total aerosol mixture, respectively. Accordingly, "$\beta_{\lambda,cd}(z)$" is expressed by:

$$\beta_{\lambda,cd}(z) = \beta_{\lambda,p}(z) \frac{\left(\delta_{\lambda,p}(z) - \delta_{\lambda,ncd}(z)\right)\left(1 + \delta_{\lambda,cd}(z)\right)}{\left(\delta_{\lambda,cd}(z) - \delta_{\lambda,ncd}(z)\right)\left(1 + \delta_{\lambda,p}(z)\right)} \tag{7}$$

For a proper implementation of Eq. (7), and towards the accurate determination of the atmospheric coarse-mode dust component, proper knowledge of the non-coarse-mode aerosol and coarse-mode dust light-depolarization characteristics, thus of "$\delta_{ncd}$ and "$\delta_{cd}$", is required. In Eq. (7), under the special cases of "$\delta_{\lambda,p}(z) \leq \delta_{\lambda,cd}(z)$" and "$\delta_{\lambda,p}(z) \geq \delta_{\lambda,cd}(z)$", we set "$\beta_{\lambda,cd}(z) = 0$" and "$\beta_{\lambda,cd}(z) = \beta_{\lambda,p}(z)$", respectively, accounting for the cases of negligible and dominant contribution of coarse-mode dust in the total aerosol mixture.

Sakai et al. (2010) performed extensive chamber laboratory experiments with the overarching objective to determine the dependence of near-backscattering linear depolarization ratio of Saharan and Asian dust on particle size distribution. The authors reported on the significantly different polarization properties of the fine-mode and coarse-mode dust populations, with the submicrometer-dominated aerosol sample yielding depolarization ratios around 0.16 ± 0.03 at 532 nm, and the supermicrometer-dominated aerosol sample resulting in depolarization ratios of 0.39 ± 0.04 at 532 nm. The laboratory findings

of Sakai et al. (2010) were to an extent confirmed by size-segregated polarization measurements performed by Järvinen et al. (2016), for a large variety of desert-dust samples of Asian, African and American origin. The authors were able to inject well-constrained mono-modal dust populations into an aerosol laboratory chamber, and based on near-backscatter polarization measurements at 488 and 552 nm, reported on the dependence of dust linear depolarization ratio on dust size-distribution. More specifically, Järvinen et al. (2016) experimentally showed the distinct linear depolarization effects of the submicrometer

and supermicrometer dust modes, with depolarization ratios of fine-mode accumulating in the same region as reported by Sakai et al. (2010), and coarse-mode no lower than 0.18 and a mean value of 0.27. Therefore, according to the published laboratory studies we assume mean linear depolarization effects of "$\delta_{ncd}$" and "$\delta_{cd}$" equal to 0.16 ± 0.02 and 0.39 ± 0.03 respectively, at 532 nm – (Fig.2 – "2nd Step)".

    It should be mentioned that the selection of "$\delta_{\lambda,ncd}$" consists one of the main assumptions and challenges of the present study,

assumed equal 0.16 ± 0.02 for two main reasons. The first one relates to Järvinen et al. (2016) and the chamber laboratory experiments of near-backscattering linear depolarization ratios of fine-mode dust, reporting that "$\delta_{fd}$" greatly varies between 0.05 for dust particles of diameter around to 0.5 µm to values as high as 0.41 in the case of submicrometer particles with diameter close to 1 µm. The second one conceptually follows also the Järvinen et al. (2016) experimental measurements, reporting coarse-mode dust linear depolarization ratio values not lower than 0.18. In this case, implementation of "$\delta_{\lambda,ncd}(z)$"

equal to 0.16 is in closer agreement with the basic assumption that cases of "$\delta_{\lambda,p}(z) \leq \delta_{\lambda,ncd}(z)$" should yield "$\beta_{\lambda,cd}(z) = 0$". In the present study, and as a final step, the fine-mode component of pure-dust is estimated as the residual between the ESA-LIVAS pure-dust product, extracted from the total aerosol load based on the pure-dust decoupling methodology (Shimizu et al., 2004; Tesche et al., 2009), and the coarse-mode component of pure-dust, extracted from the total aerosol load based on the first step of the two-step POLIPHON (Mamouri and Ansmann, 2014; 2017) (Eq. (8)), both algorithm branches applied to

CALIOP optical products (Sect.2.1.1.) – (Fig.2 – "3rd Step)".

$$\beta_{\lambda,fd}(z) = \beta_{\lambda,d}(z) - \beta_{\lambda,cd}(z) \tag{8}$$

    It should be noted that the size of mineral dust particles suspended into the atmosphere spans more than three orders of magnitude, from less than 0.1 µm to more than 100 µm in diameter (Mahowald et al., 2014; Ryder et al., 2019). This extended





range of airborne dust size distribution is closely related to widespread inconsistencies in the terminology of dust size classes. The general consensus is that the size classification has to follow the broad modes apparent in the aerosol size distribution, defining a fine-mode and a coarse-mode component (Seinfeld and Pandis, 2006; Whitby, 1978). However significant inconsistencies are observed in the definition of the boundary diameter separating the two modes. To date, studies have applied

boundary separating diameter of 1 μm (Mahowald et al., 2014; Mamouri and Ansmann, 2014; 2017; Ansmann et al., 2017), 2 μm (Spurny, 1998; Whitby, 1978; Willeke and Whitby, 1975), 2.5 μm (Seinfeld and Pandis, 2006; Zhang et al., 2013; Perez García-Pando et al., 2016), 4 μm (Rajot et al., 2008), even 5 μm (Kok et al., 2017; Adebiyi and Kok, 2020). To this end, Adebiyi et al., (2023) reviewed related dust size distribution studies and proposed a uniform classification for atmospheric dust particles, including the fine, coarse, super-coarse, and giant dust classes, with dust separation boundary geometric

diameters of 2.5, 10, and 62.5 μm, respectively. However, the definition of the fine-mode class as submicrometer (including the Aitken and accumulation modes) and the coarse-mode class as supermicrometer (including the coarse, super-coarse, and giant dust sub-classes of Adebiyi et al. (2023)), corresponding to boundary diameter separating the two modes of 1 μm, is related to and enforced by the experimental techniques, the outcomes, and parametrizations of Sakai et al (2010) and Järvinen et al. (2016) laboratory experiments, and thus the parametrization/methodology cannot be adapted to separate the dust size

distribution in classes of different boundary diameter between the two modes.

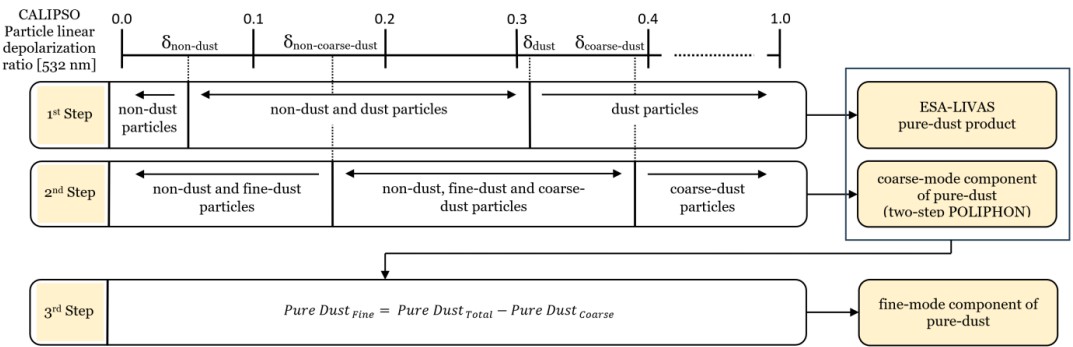

Figure 2: Illustration of the conceptual approach applied towards the derivation of the CALIPSO-based fine-mode and coarse-mode components of the total aerosol load at a near-global scale. Step 1 includes implementation of the CALIPSO-based ESA-

LIVAS pure-dust global product. Step 2 provides the coarse-mode pure-dust atmospheric component through implementation of the first step of the two-step POLIPHON method. Step 3 provides the fine-mode pure-dust component as the residual between the ESA-LIVAS pure-dust product and the coarse-mode component of pure-dust. Moreover, the figure reports on "$\delta_d$", "$\delta_{nd}$", "$\delta_{cd}$", and "$\delta_{ncd}$" depolarization at 532 nm threshold values considered here, equal to 0.31, 0.05, 0.39, and 0.16 respectively.

Mamouri and Ansmann (2014) used AERONET observations and HYSPLIT transport and dispersion model to determine a lower boundary "$\delta_{\lambda,ncd}(z)$" equal to 0.12, that would yield more accurate two-step POLIPHON separation of the fine-mode and coarse-mode dust components, for an intense event of Middle East dust advection over Limassol, Cyprus on the 28th of September 2011. In addition, the high quality of the two-step POLIPHON technique, when multi-wavelength lidar polarization

measurements are applied has been demonstrated in the case of a 3 km deep Saharan dust layer over Barbados, observed in the framework of the Saharan Aerosol Long Range Transport and Aerosol-Cloud-Interaction Experiment (SALTRACE; Weinzierl et al., 2016), on June 20th, 2014 (Mamouri and Ansmann, 2017). However, implementation of the discussed methodology to CALIPSO optical products with the objective to decouple the fine-mode and coarse-mode pure-dust atmospheric components on a near-global scale would require additional considerations, since CALIOP configuration neither

provides dual-wavelength polarization lidar profiling of the atmosphere nor concurrent sunphotometer observations. More



specifically, consideration of AERONET AOT$_f$, AOT$_c$, and FMF with the objectives of constraining the decoupling approach though assumptions on "$\delta_{\lambda,ncd}(z)$", on the lidar ratio, or on the percentages of the fine- and coarse-mode to the total aerosol load on a near-global scale, would result to uncertainties relate to (1) the comparison of fine-mode dust optical depth with AOT$_f$, an AERONET product including in addition non-dust fine-mode aerosol subtypes (e.g., biomass burning and urban haze), (2) the comparison of coarse-mode dust optical depth with AOT$_c$, an AERONET product including in addition non-dust coarse-mode aerosol subtypes (e.g., marine, pollen, volcanic ash), and (3) the fact that AERONET retrievals constrain the particle size to less than 30 μm diameter (Dubovik and King, 2000; Dubovik et al., 2000), thus missing the super-coarse and giant modes. Moreover, such implementation of AERONET would require extended assumptions further increasing the induced uncertainties (e.g., CALIPSO-AERONET collocation criteria, atmospheric homogeneity and topographical characteristics, different SNR between daytime and nighttime illumination conditions, AERONET geographical coverage, nighttime observations).

In the ESA-LIVAS pure-dust climate data record uncertainties resulting from the impact of the non-dust aerosol components to the total aerosol load (e.g., "$\delta_{\lambda,ncd}(z)$") are counter-balanced through the CALIPSO algorithm of classifying detected atmospheric aerosol features. More specifically, CALIOP's V4 L2 tropospheric aerosol subtype classification algorithm uses an approximate particle depolarization ratio "$\delta_p^{est}$", 532 nm integrated attenuated backscatter, layer top "Z$_{top}$", layer base "Z$_{base}$" and information of the underlining Earth's surface type and location to assign between seven tropospheric aerosol subtypes (Omar et al., 2009; Kim et al., 2018). While CALIOP's tropospheric aerosol subtype classification may lead to distinction ambiguities, especially in cases of aerosol mixtures (Burton et al., 2013), the use of "$\delta_p^{est}$" aerosol intensive property provides reliable information about the presence or the absence of dust in identified atmospheric features (Liu et al., 2012). Thus, following the approach established in the framework of the ESA-LIVAS pure-dust product (Amiridis et al., 2013; 2015), the fine-mode and coarse-mode dust separation technique is applied only to the "dust", "polluted dust" and "dusty marine" aerosol subtypes, while the "marine", "polluted continental/smoke", "clean continental", and "elevated smoke" atmospheric layers are neglected.

Finally, it must be mentioned that towards decoupling the pure-dust component from the total aerosol mixture, the developed methodology assumes the "dust", "polluted dust" and "dusty marine" classified atmospheric layers as external mixtures of dust and non-dust aerosol components. The external-aerosol mixture assumption does not account for coating effects possibly altering the polarization properties of the observed aerosol subtypes, resulting in misclassification and further uncertainties. However, recent chamber laboratory experiments provide evidence that dust depolarization optical properties may not be significantly influenced by thin coating of sulfuric acid, secondary organics, humidity, or aging processes (Järvinen et al.; 2016). This is of particular importance for the separation of fine-mode and coarse-mode dust on a global scale, as it suggests that the lower polarization properties of dust layers at large distances downstream from the emission sources may likely relate to gravitational settling of larger dust particles along the path and not to alterations of the polarization characteristics of dust.

The basic outcomes of our approach - till this point - include CALIOP-based profiles of backscatter coefficient at 532 nm of fine-mode pure-dust ($\beta_{\lambda,fd}(z)$), coarse-mode pure-dust ($\beta_{\lambda,cd}(z)$), and total pure-dust ($\beta_{\lambda,d}(z)$), components of the total aerosol load, along the CALIPSO orbit path at uniform 5 km horizontal and 60 m vertical resolution (Fig.4). To convert the obtained total, fine-mode, and coarse-mode dust backscatter coefficient profiles into respective extinction coefficient profiles, characteristic total, fine-mode, and coarse-mode pure-dust lidar ratio (LR) at 532 nm values are implemented (Eq. (9a-c)). In absence of extended observational/laboratory studies on the dependence of pure-dust extinction-to-backscatter ratios on the dust size distribution, we follow one of the basic assumptions of the two-step POLIPHON, that of equal LRs for total, fine-mode, and coarse-mode pure-dust LRs (Mamouri and Ansmann, 2014; 2017). Regionally characteristic pure-dust LRs at 532 nm are taken from the literature, following the ESA-LIVAS regional classification (Amiridis et al., 2013; 2015; Marinou et al., 2017; Proestakis et al., 2018), in order to facilitate implementation of the fine-coarse mode decoupling methodology (Fig.2)





without introducing ambiguities (Table 3; Fig.3). We should note though that according to performed modelling studies (Gasteiger et al., 2011; Kemppinen et al., 2015a, b), such assumption of equal LRs for total, fine-mode, and coarse-mode pure-dust LRs (Mamouri and Ansmann, 2014; 2017) may result to fine-mode and coarse-mode extinction coefficient/DOD underestimation and overestimation, respectively. Moreover, it must be mentioned that the selected approach of applying

5    regionally-dependent pure-dust LRs (Amiridis et al., 2013; 2015; Marinou et al., 2017; Proestakis et al., 2018) and not a universal pure-dust LR value is expected to reduce biases in regional studies, however at the expense of introduced uncertainties due to long-range transport of dust and discontinuities in the region borders (Kim et al., 2018).

$$\alpha_{\lambda,d}(z) = LR_{\lambda,d} * \beta_{\lambda,d}(z) \tag{9a}$$

$$\alpha_{\lambda,cd}(z) = LR_{\lambda,d} * \beta_{\lambda,cd}(z) \tag{9b}$$

$$\alpha_{\lambda,fd}(z) = LR_{\lambda,d} * \beta_{\lambda,fd}(z) \tag{9c}$$

10    In Eq. (9a-c), "$\beta_{\lambda,d}(z)$", "$\beta_{\lambda,fd}(z)$", and "$\beta_{\lambda,cd}(z)$" correspond to the total, fine-mode, and coarse-mode pure-dust backscatter coefficient at 532 nm, "$LR_{\lambda,d}(z)$" is the pure-dust Lidar Ratio at 532 nm (Table 3), and "$a_{\lambda,d}(z)$", "$a_{\lambda,fd}(z)$", and "$a_{\lambda,cd}(z)$" correspond to the total, fine-mode, and coarse-mode pure-dust extinction coefficient at 532 nm.

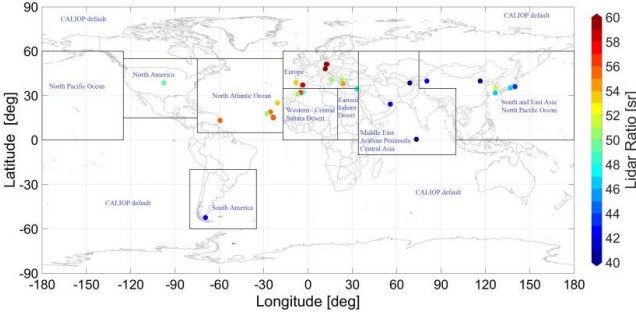

15    Figure 3: Illustration of the regional classification of pure-dust LR values applied in the present study.

Table 3: Overview of pure-dust lidar-based LR values (sr) classified under specific geographical regions of interest (Fig.3).

| Region | LR 532 nm [sr] | References | LR 532 nm [sr] |
|---|---|---|---|
| Western - Central Sahara Desert North Atlantic Ocean | 56 ± 8 | Tesche et al. (2009) | 56 ± 5 |
| | | Gross et al. (2011a) | 63 ± 6 |
| | | Gross et al. (2011b) | 62 ± 5 |
| | | Tesche et al. (2011) | 54 ± 10 |
| | | Kanitz et al. (2013) | 55 ± 5 |
| | | Kanitz et al. (2014) | 50 ± 5 |
| | | Gross et al. (2015) | 56 ± 7 |
| | | Weinzierl et al. (2016) | 55 ± 5 |
| | | Haaring et al. (2017) | 55 ± 5 |
| | | Rittmeister et al. (2017) | 55 ± 5 |
| | | Bohlmann et al. (2018) | 53 ± 2 |
| Eastern Sahara Desert | 53 ± 6 | Nisantzi et al. (2015) | 41 ± 4 |
| | | Ansmann et al. (2019) | 50 ± 10 |
| Middle East Arabian Peninsula Central Asia | 40 ± 5 | Müller et al. (2007) | 38 ± 5 |
| | | Mamouri et al. (2013) | 36.4 ± 5.9 |
| | | Nisantzi et al. (2015) | 41 ± 4 |
| | | Hofer et al. (2017) | 39.3 ± 3.6 |
| | | Filioglou et al. (2020) | 42 ± 5 |
| South and East Asia | 46 ± 7 | Liu et al. (2002) | 51 ± 9 |





| Region | | Reference | |
|---|---|---|---|
| North Pacific Ocean | | Sakai et al. (2002) | 46 ± 5 |
| | | Anderson et al. (2003) | 44 ± 8 |
| | | Sakai et al. (2003) | 47 ± 18 |
| | | Murayama et al. (2003) | 46.5 ± 10.5 |
| | | Murayama et al. (2004) | 56 ± 8 |
| | | Noh et al. (2007) | 51 ± 6 |
| | | Tesche et al. (2007) | 40 ± 5 |
| | | Noh et al. (2008) | 51 ± 6 |
| | | Jin et al. (2010) | 42 ± 3 |
| Europe | 56 ± 8 | Mattis et al. (2002) | 60 ± 10 |
| | | Ansmann et al. (2003) | 60 ± 20 |
| | | Müller et al. (2007) | 57 ± 2 |
| | | Guerrero-Rascado et al. (2009) | 57.5 ± 7.5 |
| | | Papayiannis et al. (2011) | 59 ± 11 |
| | | Preißler et al. (2011) | 53 ± 7 |
| | | Wiegner et al. (2011) | 59 ± 6 |
| | | Preißler et al. (2013) | 56 ± 8 |
| | | Soupiona et al. (2019) | 64 ± 6 |
| North America | 49 ± 9 | Burton et al. (2012) | 49 ± 9 |
| South America | 42 ± 17 | Kanitz et al. (2013) | 42 ± 17 |

Through the implementation of suitable geographically-dependent pure-dust LR values, the CALIOP-based profiles of backscatter coefficient at 532 nm of fine-mode pure-dust ($\beta_{\lambda,fd}(z)$), coarse-mode pure-dust ($\beta(z)$), and total pure-dust ($\beta_{\lambda,d}(z)$) are converted to profiles of extinction coefficient at 532 nm of fine-mode pure-dust ($a_{\lambda,fd}(z)$), coarse-mode pure-dust ($a(z)$), and total pure-dust ($a_{\lambda,d}(z)$), along the CALIPSO orbit path at uniform 5 km horizontal and 60 m vertical resolution (Fig.4). To convert the obtained total and coarse-mode extinction coefficient profiles into respective mass concentration profiles (MC) characteristic total ($c_{v,d}$) and coarse-mode ($c_{v,cd}$) volume concentration conversion factors at 532 nm (Ansmann et al., 2019) are implemented (Eq.(10a/b)). More specifically, Ansmann et al. (2019) explored dust optical and microphysical properties based on long-term AERONET observations and retrievals, and established a near-global set of climatologically representative extinction-to-volume conversion factors (Table 4). Here we utilize the Ansmann et al. (2019) total ($c_{v,d}$) and coarse-mode ($c_{v,cd}$) volume concentration conversion factors at 532 nm and typical pure-dust particle density of $\rho_d$: 2.6 gcm$^{-3}$ (Ansmann et al., 2012), to obtain the total and coarse-mode pure-dust mass concentration profiles along the CALIPSO orbit-path. Accordingly, in the framework of the present study, the fine-mode pure-dust mass concentration profiles are extracted as the residual between the total pure-dust mass concentration profiles and the coarse-mode pure-dust mass concentration profiles (Eq. (10c)) (Fig.4).

$$MC_d = \rho \cdot c_{v,d} \cdot a_d \tag{10a}$$

$$MC_{d_c} = \rho \cdot c_{v,d_c} \cdot a_{d_c} \tag{10b}$$

$$MC_{d_f} = MC_d - MC_{d_c} \tag{10c}$$

Table 4: Geographically dependent extinction coefficient at 532 nm to volume concentration conversion factors for total ($c_{v,d}$) and coarse-mode ($c_{v,cd}$) pure-dust (in 10$^{-12}$ Mm) used in the present study, as established and provided by Ansmann et al. (2019) based on long-term AERONET observations and retrievals of dust optical and microphysical properties.

| Region | $c_{v,d}$ | $c_{v,dc}$ |
|---|---|---|
| Western – Central Sahara Desert – North Atlantic Ocean – Europe | 0.68 ± 0.08 | 0.83 ± 0.09 |
| Middle East – Arabian Peninsula | 0.71 ± 0.08 | 0.86 ± 0.10 |
| Central Asia – South and East Asia – North Pacific Ocean | 0.78 ± 0.10 | 0.95 ± 0.12 |
| America – Australia | 0.89 ± 0.13 | 1.07 ± 0.14 |

Figure 4: Illustration of the methodology, from quality assurance – to backscatter coefficient at 532 nm – to – extinction coefficient at 532 nm – to mass concentration separation and conversion steps of the fine-mode and coarse-mode components of atmospheric pure-dust, components of the total aerosol load, for an indicative CALIPSO-Middle East overpass case on the 23rd of September, 2015.





### 2.2.2 Total, fine-mode, and coarse-mode pure-dust product uncertainties: backscatter coefficient 532nm, extinction coefficient 532nm and Mass Concentration

Uncertainties in the retrieval of fine-mode and coarse-mode pure-dust products, in terms of backscatter coefficient, extinction coefficient and mass concentration, are attributed mainly to three sources: (a) uncertainties in the CALIPSO L2 optical products (i.e., of backscatter coefficient and particulate depolarization ratio at 532nm, the feature type, aerosol subtype classification), (b) uncertainties in the total, coarse-mode, fine-mode pure dust decoupling methodology (i.e., of characteristic particulate depolarization ratio of pure-dust "$\delta_d$", coarse-dust "$\delta_{cd}$", non-dust "$\delta_{nd}$" aerosol categories) and (c) uncertainties in the assumption of constants, conversion factors, and constraints (i.e., of LR, extinction-to-mass concentration conversion factors). The uncertainties in the CALIPSO L2 backscatter coefficient, extinction coefficient and aerosol optical depth (AOD) are calculated in the Selective Iterated Boundary Locator (SIBYL), Scene Classification Algorithms (SCA) and Hybrid Extinction Retrieval Algorithms (HERA) processing algorithms (Vaughan et al., 2009; Winker et al., 2009), based on the assumption that the uncertainties are random, uncorrelated and produced no biases. In QA atmospheric layers (Sect.2.1.1) classified as dust, polluted dust or dusty marine, CALIPSO L2 V4 uncertainties in the backscatter coefficient 532nm are typically of the same order of magnitude, while the corresponding uncertainties in particulate depolarization ratio are typically >100%. Additional uncertainties arise from the assumptions in CALIOP aerosol subtype algorithm (Omar et al., 2009; Kim et al., 2018), and the deficiency of CALIOP to detect tenuous aerosol layers (Kacenelenbogen et al., 2011; Rogers et al., 2014). Burton et al. (2013), based on NASA B200 HSRL-1 and CALIOP coincident measurements reported on the high performance of CALIPSO V3 aerosol classification algorithm in case of dust mixtures (~80%), less good though agreement in case of polluted dust (~35%). Despite the improvements made in CALIPSO V4 (Kim et al., 2018), erroneous classification of dusty atmospheric aerosol features results in layers not processed in the pure-dust, fine-mode and coarse mode decoupling chain (Sect. 2.2.1). Moreover, CALIOP deficiency to detect tenuous aerosol layers result to nighttime and daytime negative biases of ~0.02 and <0.1 respectively in terms of AOD, attributed mainly to CALIOP minimum detection nighttime and daytime thresholds of 0.012 km$^{-1}$ and 0.067 km$^{-1}$ respectively (Toth et al., 2018). Overall, assumptions on the aerosol subtype algorithm may result to both positive or negative biases, while undetected tenuous layers result to negative biases.

The overall uncertainties induced from the application of pure-dust decoupling algorithms are extensively discussed by Shimizu et al., (2004), Tesche et al. (2009, 2011), Mamouri and Ansmann (2014, 2017), and Ansmann et al. (2019). Moreover, Amiridis et al. (2013) and Marinou et al. (2017) carried out a detailed analysis of the uncertainties in the framework of the EARLINET-optimized CALIPSO-based pure-dust product. Regarding the uncertainties induced from the pure-dust depolarization-based separation method range between 5-10% in strong dust events and 20-30% in less-pronounced dust layers (Tesche et al., 2009, 2011; Ansmann et al., 2012; Mamouri et al., 2013). The corresponding uncertainties are mainly attributed to the assumed characteristic particulate depolarization ratio values of pure-dust "$\delta_d$", coarse-dust "$\delta_{cd}$" and non-dust "$\delta_{nd}$" aerosol categories, with standard deviations considered as the basic source of information in the uncertainty analysis (Mamouri and Ansmann 2017), as will be in the present study. Moreover, extensive assessment of the effect of the variations of the fine-mode pure-dust depolarization ratio implemented in the two-step POLIPHON approach is provided by Mamouri and Ansmann (2014, 2017), considering as the optimal selection a value of 0.12. Performed simulations and sensitivity studies demonstrated the relatively low effect of the corresponding variations of $\delta_{nd+df,e}$ from 0.08 to 0.16, with relative errors induced to the fine-mode and coarse-mode pure-dust products of the order of 10%. The estimated uncertainties are in agreement with Marinou et al. (2017), reporting uncertainties in the EARLINET-optimized CALIPSO dust product as high as 8%, increasing at downwind and distant areas off the major dust sources, attributed to the effect of pure-dust and non-dust depolarization ratio parameters. Additional relative uncertainties of the order of 15-25% arise in the successive conversion of the decoupled backscatter coefficient profiles of pure-dust (total, fine-mode and coarse-mode) to corresponding extinction coefficient profiles (Eq.(9a-





c)), by applying suitable geographically-dependent dust lidar-ratios (Tesche et al., 2009; Amiridis et al., 2013; Mamouri and Ansmann, 2014, 2017; Marinou et al., 2017). It must be noted that although modelling simulation studies report on different LR values for fine-mode and coarse-mode dust (Gasteiger et al., 2011), in absence of extended pure-dust laboratory LR experiments, similar LRs for the total, coarse-mode, and fine-mode pure-dust components are assumed (Mamouri and

Ansmann, 2014, 2017). Finally, additional relative uncertainties of the order of 10-15% arise in the successive conversion of extinction coefficient profiles of pure-dust (total, fine-mode and coarse-mode) to corresponding mass concentration profiles (Table 4: Eq.(10a-c)), based on appropriate mass concentration conversion factors provided and discussed by Ansmann et al. (2019).

Overall, through the successive steps in the computation of fine-mode and coarse-mode pure-dust products, starting from the

backscatter coefficient profiles, to extinction coefficient profiles, and eventually to mass concentrations profiles, the relative uncertainties increase at each intermediate step, due to the corresponding uncertainties that must be considered. The pure-dust decoupling uncertainties reported in the literature, in the case of ground-based Raman/polarization lidar studies for moderate and high dust concentrations range between 10% and 30% in terms of backscatter coefficient, between 15% and 50% in terms of extinction coefficient and between 20% and 60% in terms of mass concentration (Mamouri and Ansmann, 2017; Ansmann

et al., 2019). However, in the case of the CALIPSO-based pure-dust products (total, coarse-mode, fine-mode) the driving factor in the uncertainties is the CALIPSO L2 backscatter coefficient and particulate depolarization ratio uncertainties, typically of the same order of magnitude with the corresponding optical products, resulting to overall uncertainties extending between 100% and 150% (Marinou et al., 2017; Proestakis et al., 2018).

**3 Consistency Check**

**3.1 Coarse-Dust and Fine-Dust Optical Depth product comparison with AERONET observations**

This section aims to evaluate the CALIPSO-based $DOD_{coarse}$ and $DOD_{fine}$ at 532 nm products through the extensive

implementation of AERONET $AOT_{coarse}$ and $AOT_{fine}$ retrievals at 532 nm, taking into consideration the unique characteristics of CALIOP and sunphotometer measurements and products, the quality assurance criteria, and the synchronization and collocation requirements. More specifically, CALIOP L2 reports aerosol and cloud measurements and products in near-vertical "sheets" of near-zero swath (~100 m footprint) on the Earth's surface, along the CALIPSO orbit-track, with 5 km horizontal resolution. On the contrary, AERONET instruments are characterized by approximately ~1.2º full angle field-of-view,

resulting in columnar pencil-like multiwavelength measurements of aerosol optical thickness between the solar disc and the sun-sky photometers (Holben et al. 1998; Omar et al., 2008). Consequently, given the natural atmospheric inhomogeneity of the aerosol fields, both in time and space, CALIOP and AERONET rarely probe the same air volumes of the atmosphere. Hence, towards extensively comparing CALIOP and AERONET columnar observations implementation of a set of constraints and criteria is a prerequisite, to ensure a robust comparison and thus to establish the quality of the CALIPSO-based $DOD_{coarse}$

and $DOD_{fine}$ at 532 nm products against AERONET $AOT_{coarse}$ and $AOT_{fine}$ retrievals at 532 nm.

One of the major factors driving comparison discrepancies in the correlative observational dataset relates to the high spatial and temporal inhomogeneity of the aerosol fields. In this study, we adopt the CALIPSO-AERONET collocation criteria established in satellite-based lidar studies, though the collocation criteria vary from study-to-study depending on the scope and requirements (Amiridis et al., 2013; Anderson et al., 2003; Omar et al., 2013; Pappalardo et al., 2010; Proestakis et al., 2019;

Schuster et al., 2012). The spatial and temporal collocation criteria applied in the framework of the present study require CALIPSO maximum overpass distance less than 80 km from the AERONET site and AERONET AOD acquisition measurements within ±60 min of the CALIPSO overpass. The selection of the spatial matching distance of CALIPSO L2 profiles within 80 km radius of the AERONET sites is justified in terms of the CALIPSO feature detection algorithms. CALIOP L2 Selective Iterated BoundarY Locator (SIBYL) processing module detects atmospheric features with L1 profile horizontal





averaging of 5, 20, and 80 km along the CALIPSO orbit-track (Vaughan et al., 2009). Thus the maximum 80 km along-track averaging resolution in the detection of atmospheric layers performed by SIBYL prohibits observations of the same atmospheric air masses by CALIOP and AERONET photometers, even in cases of CALIPSO orbit-track directly over the AERONET stations (Schuster et al., 2012). Consideration of CALIPSO distance closer than 80 km, although reported to result

in slight improvement in CALIOP-AERONET absolute biases, would be performed at the expense of larger errors due to reduction of the sample-size of synchronized measurements. Accordingly, the temporal collocation criterion between CALIOP and AERONET observations of 60 min ($\Delta t \leq 60$ min) is justified in terms of the mesoscale natural variability of aerosol in the lower-troposphere (Anderson et al., 2003; Pappalardo et al., 2010).

CALIPSO AODs at 532 nm are computed as vertical integration of the mean CALIOP extinction coefficient at 532 nm profile

with respect to height, between TOA and the elevation height of AERONET sunphotometers, thus not considering atmospheric aerosol fields possible accumulated below the AERONET sites (Amiridis et al., 2013; Schuster et al., 2012). Regarding the negative effects of clouds in the comparison cases, the use of CALIOP L2 profiles of cloud optical depth equal to zero (0) and atmospheric features of CAD score lower than −20 ensures the lowest possible contamination of CALIPSO AODs by cloud features. It must be noted that the CALIPSO-AERONET overpasses may result to non-representative CALIPSO AODs in

cases of extensive presence of clouds, due to reduction of the considered CALIPSO L2 profiles over the area (Sect.2.1.1.). To minimize the negative impact of clouds in the analysis, cases with extensive presence of clouds detected in CALIPSO L2 measurements are not considered in the comparison, through applying a lower acceptable threshold of at least eight (8) CALIPSO L2 5km cloud-free profiles (~80 km) within the 80 km distance from the AERONET stations. Accordingly, at least two (2) AERONET L2 AOD measurements are required to have been performed in the CALIPSO overpass ± 60 min temporal

window, for accepting an AERONET case.

Table 5: Requirements for the comparison between the CALIPSO-based DOD$_{coarse}$ and DOD$_{fine}$ at 532 nm products and AERONET AOT$_{coarse}$ and AOT$_{fine}$ at 532 nm retrievals.

| | | |
|---|---|---|
| 1 | CALIPSO footprint – AERONET station distance | $\leq 80$ km |
| 2 | CALIPSO – AERONET overpass temporal window | $\pm 60$ min |
| 3 | AERONET AOT measurements (#) | $\geq 2$ |
| 4 | CALIPSO L2 5km Cloud-Free profiles (#) | $\geq 8$ |
| 5 | AERONET AOT (532 nm) | $\geq 0.01$ |
| 6 | CALIOP AOD (532 nm) | $\geq 0.01$ |
| 7 | \| Relative Difference (CALIOP DOD / AERONET AOT) \| | $\leq 50\%$ |

The evaluation of the CALIPSO-based DOD$_{coarse}$ and DOD$_{fine}$ products against AERONET AOT$_{coarse}$ and AOT$_{fine}$ products should be performed in cases characterized by dust presence. The AERONET Ångström Exponent (AE) cannot be used, for the implementation of low AE to establish correlative CALIPSO-AERONET cases dominated by larger particles would drive the comparison towards excluding from the analysis submicrometer dominated pure-dust cases. The CALIPSO aerosol subtype cannot be used either as appropriate dust identifier, since it provides only qualitative information on the presence of dust. More

specifically, implementation of CALIPSO aerosol subtyping as a proxy of high dust presence would result in underestimations of CALIPSO DOD$_{coarse}$ (DOD$_{fine}$) extracted from classified layers as "dusty marine" ("polluted dust") upon comparison against AERONET AOT$_{coarse}$ (AOT$_{fine}$), since the retrievals would be based on columnar measurements of both dust and marine (smoke) aerosols. Finally, the use of CALIOP particulate depolarization ratio at 532 nm cannot be used either as a proxy for the identification of dust cases, since this optical parameter is implemented in the process of extracting the CALIPSO-based

fine-mode and coarse-mode pure-dust components, and thus the parameter does not consist an independent identifier. Thus, to facilitate the evaluation of the CALIPSO-based DOD$_{coarse}$ and DOD$_{fine}$ with AERONET AOT$_{coarse}$ and AOT$_{fine}$, only for cases of dust presence in the atmosphere, we implement the pure-dust product developed in the framework of the ESA-LIVAS activity (Amiridis et al., 2013; 2015), under the quantitative assumption of at least 50% of the observed AERONET AOT to





be related to the presence of dust, as provided by the LIVAS DOD at 532 nm product ($|\text{Rel.Dif.}_{\text{CALIOP\_DOD, AERONET\_AOT}}| \leq$ 50%). Finally, the comparison between the CALIPSO-based $\text{DOD}_{\text{coarse}}$ and $\text{DOD}_{\text{fine}}$ at 532 nm products and AERONET $\text{AOT}_{\text{coarse}}$ and $\text{AOT}_{\text{fine}}$ at 532 nm retrievals is performed only for the cases of both CALIPSO AOD at 532 nm and AERONET AOT at 532 nm at least equal to 0.01.

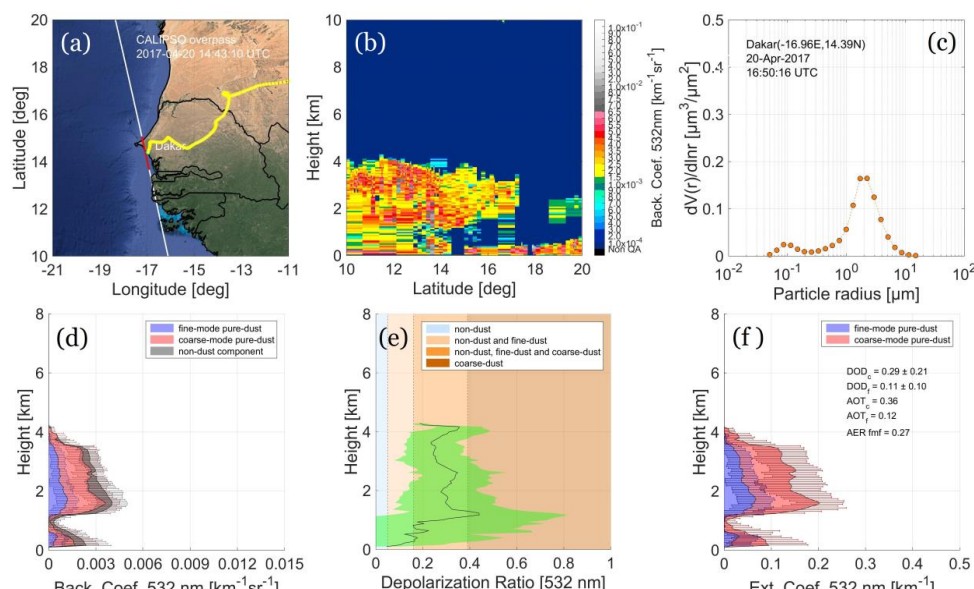

Figure 5: Major Saharan dust outbreak moving westwards over Dakar on the 20[th] of April, 2017 at ~14:43 UTC, including the CALIPSO overpass in the proximity of the AERONET-Dakar station (red line) and FLEXPART 6-day back-trajectories at 2 km at the area of interest (Lat: 14.39, Lon: -16.95) denoting the Saharan desert origin of the advected air masses (yellow line) (Fig.5a). CALIPSO L2 5 km backscatter coefficient at 532 nm cross section (Fig.5b). Column-integrated particle volume concentration as a function of particle radius observed with the AERONET sunphotometer over Dakar, Senegal (Fig.5c). Backscatter coefficient at 532 nm profiles of the coarse-mode pure-dust (red shaded area), fine-mode pure-dust (blue shaded area), and non-dust (gray shaded area) components of the total aerosol load (Fig.5d). Particulate depolarization ratio at 532 nm profile used for the decoupling of the coarse-mode pure-dust, fine-mode pure-dust, and non-dust components of the total aerosol load, as provided in Fig.5d (Fig.5e). Extinction coefficient at 532 nm profiles of the coarse-mode pure-dust (red shaded area) and fine-mode pure-dust (blue shaded area) components of the total aerosol load (Fig.5f).

Figure 5 provides an overview of the comparison for the case of a Saharan dust outbreak reaching Dakar on the 20[th] of April, 2017 (Fig.5a). FLEXPART v10.4 (FLEXible PARTicle) Lagrangian dispersion model (Stohl et al., 2005; Ignacio Pisso et al., 2019) 6-day air masses back-trajectories at 2 km in the area of interest (Lat: 14.39, Lon: -16.95) are performed. The initial and boundary conditions for the FLEXPART runs are produced with 3-hourly meteorological data from the National Centers for Environmental Prediction (NCEP) Global Forecast System (GFS) provided at 0.5°×0.5° spatial resolution, and 41 model pressure levels. FLEXPART has been used in a large number of similar studies on long-range atmospheric transport (Stohl et al., 2005, Solomos et al., 2019; Kampouri et al., 2021). According to the back-trajectories, the advected air masses were of north-northwestern Saharan origin (Fig.5a-yellow line). According to the CALIPSO overpass (Fig.5a-red line) in the proximity of the AERONET-Dakar sunphotometer (Fig.5c), the observed aerosol layer extended vertically between 1 and 4 km a.m.s.l. and was dominated by the presence of Saharan dust, as corroborated by both the cross section of backscatter coefficient at 532 nm (Fig.5b) and the mean particulate depolarization ratio at 532 nm profile (Fig.5e). Fig.5d provides the pure-dust coarse-





mode (red shaded area) and fine-mode (blue shaded area) components, following decoupling from the total aerosol load, including the non-dust aerosol component (gray shaded area) in terms of profiles of backscatter coefficient at 532 nm. Columnar-integrated pure-dust coarse-mode (red shaded area) and fine-mode (blue shaded area) extinction coefficient at 532 nm profiles yield CALIPSO-based DOD$_{coarse}$ and DOD$_{fine}$ at 532 nm equal to 0.29±0.21 and 0.11±0.1 respectively, optical depth values in good agreement with AERONET-Dakar AOT$_{coarse}$ and AOT$_{fine}$ of 0.36 and 0.12 values respectively (Fig.5f). Enforcing the collocation criteria and constraints described above (Table 5), to account for the spatiotemporal atmospheric homogeneity and sampling differences between the two systems, yields 1737 CALIPSO-AERONET coincidences fulfilling the requirements over the globe for a fourteen-year period (06/2006 - 05/2019). The CALIPSO-AERONET coincidences, extracted under the condition of dust presence in the atmosphere, populate mainly the arid areas of the globe and the downwind to the source region areas. In this section we provide an assessment of the derived CALIPSO-based fine-mode and coarse-mode DODs at 532 nm against the corresponding AERONET fine-mode and coarse-mode AOTs at 532 nm, retrieved as discussed in Sections 2.2.2-2.2.3 and as the presented above case for the CALIPSO - AERONET-Dakar on the 20$^{th}$ of April, 2017 (Fig.5).

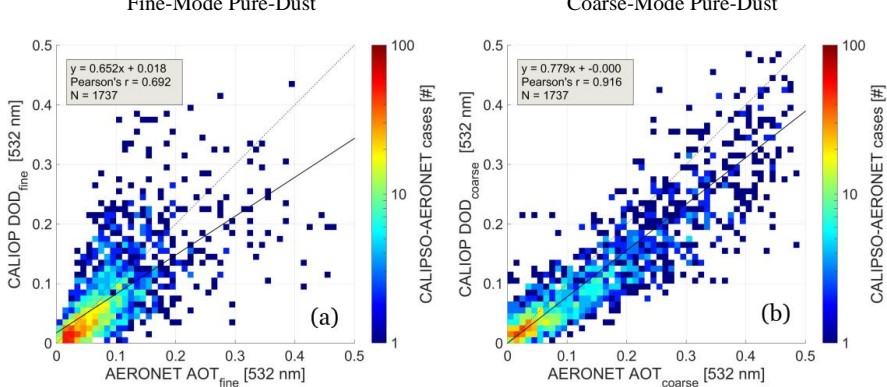

Figure 6: Evaluation of the CALIPSO-based DOD$_{fine}$ and DOD$_{coarse}$ at 532 nm products against AERONET AOT$_{fine}$ and AOT$_{coarse}$ at 532 nm retrievals respectively, in the form of 2D histogram density scatterplots.

Figure 6 evaluates the total dataset of correlative CALIPSO-based fine-mode and coarse-mode DODs and AERONET-based fine-mode and coarse-mode AOTs in the form of 2D histogram density scatterplots with a bin-step equal to 0.01. Data pairs on the delineated by the black color one-to-one dashed line correspond to perfect optical depth agreement between the CALIPSO-based fine/coarse DOD modes and the respective AERONET fine/coarse AOT modes. Linear regression (solid black line) for the submicrometer category reveals a good agreement between the two datasets, of slope 0.652, offset 0.018, and Pearson's correlation coefficient of the order of 0.692, though there is a tendency of deviating from the one-to-one line with increasing aerosol load (i.e., underestimation). With respect to the coarse-mode category the agreement between the two datasets improves substantially, with linear fit of slope 0.779, interception close to -0.002, and Pearson's correlation coefficient equal to 0.916. An apparent feature revealed through the evaluation of the CALIPSO-based fine-mode and coarse-mode DODs at 532 nm products against the AERONET retrievals of fine-mode and coarse-mode AOTs at 532 nm is the increase of the degree of dispersion with increasing AERONET AOT values (Fig.6). More specifically, the general tendency is for data pairs to be closer to the AERONET axis, resulting to an overall CALIPSO fine/coarse DOD underestimation compared to the respective AERONET fine/coarse AOT modes.



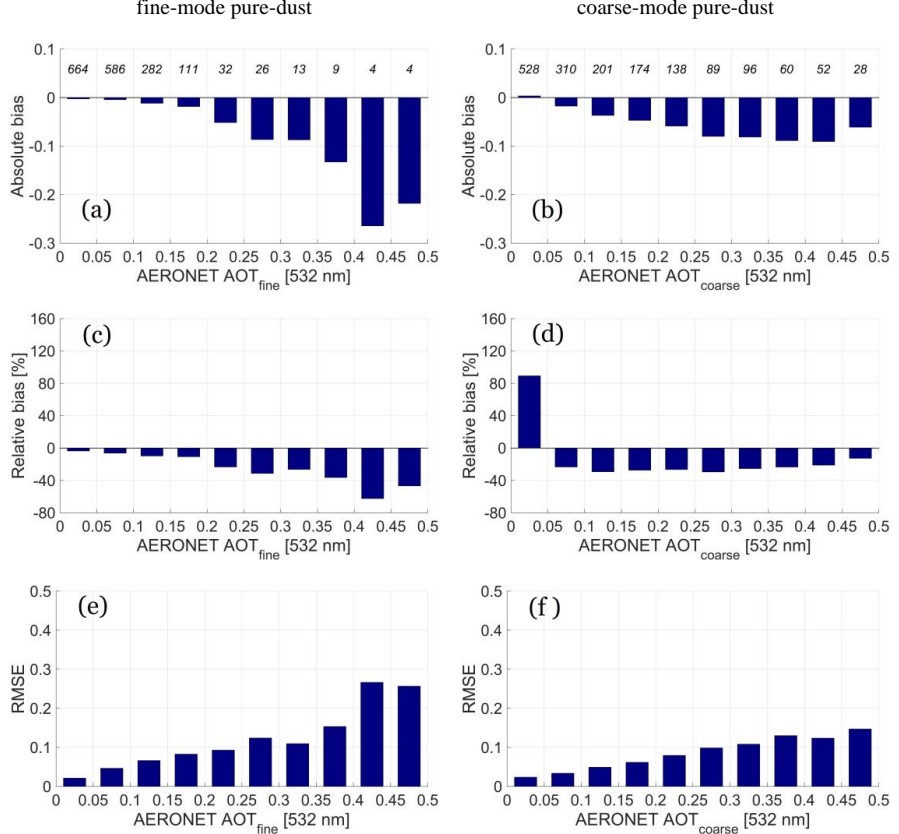

Figure 7: CALIPSO-based fine-mode (left column) and coarse-mode (right column) DOD at 532 nm absolute biases (Fig.7a/b), relative biases (Fig.7c/d), and Root Mean Square Errors - RMSE (Fig.7e/f) by AERONET fine-mode and coarse-mode AOT at 532 nm class of equal increment step of 0.05 for the AOT at 532 nm range extending between 0 and 0.5.

Figure 7 provides quantitatively the CALIPSO-based fine-mode and coarse-mode DOD at 532 nm absolute biases (Fig.7a/b), relative biases (Fig.7c/d), and Root Mean Square Errors - RMSE (Fig.7e/f) by AERONET fine-mode and coarse-mode AOT at 532 nm class of equal increment step of 0.05 for the AOT at 532 nm range extending between 0 and 0.5. Overall, with respect to the CALIPSO-based fine-mode DOD at 532 nm, the evaluation reveals negative mean absolute biases (−0.012), mean relative biases (-7.38%), and RMSE (0.059). Overall, with respect to the CALIPSO-based coarse-mode DOD at 532 nm,

the evaluation reveals negative mean absolute biases (-0.036), positive mean relative biases (9.55%), and RMSE (0.076). With respect to the CALIPSO-based coarse-mode DOD at 532 nm it must be noted though that the mean relative bias is driven positive by the observed patterns in the range of small DOD values (0-0.05), while for EARLINET AOT classes larger than 0.05 negative relative biases are observed (Fig.7d). Moreover, the analysis shows that the CALIPSO-based fine-mode and coarse-mode DOD absolute biases (Fig.7a/b) are more clearly affected for larger AOD values (increasing), while mean relative

absolute biases demonstrate significantly less variability, remaining relatively constant (Fig.7c/d).

It must be noted that the revealed here CALIPSO fine/coarse-mode DOD at 532 nm underestimation constitutes an expected feature. More specifically, it is well-documented that a discrepancy - underestimation - exists in CALIPSO extinction coefficient profiles and AODs at 532 nm, established against correlative observations by Aqua-MODIS AODs (e.g., Kacenelenbogen et al., 2011; Kittaka et al., 2011; Redemann et al., 2012; Kim et al., 2013; Ma et al., 2013), AERONET-

derived AOTs (Schuster et al., 2012; Omar et al., 2013; Amiridis et al., 2013; Toth et al., 2018), and airborne lidar observations


(Rogers et al., 2014). These studies attribute the apparent CALIPSO AOD at 532 nm underestimation to a number of factors, including - among others - misclassification of cloud layers as aerosol (and vice versa), erroneous sub-classification of the classified atmospheric layers as aerosol, incorrect selection of the aerosol subtype lidar ratios, limited or restricted penetration/attenuation of CALIOP beam within thick aerosol layers, and profiles frequently populated with retrieval fill values

(RFVs) due to failure to detect diffuse and tenuous aerosol layers of SNR below CALIOP minimum detection thresholds. The CALIPSO fine/coarse-mode DOD at 532 nm underestimation with respect to AERONET AOT at 532 nm is further enhanced by the fine-mode, coarse-mode, and total pure-dust decupling algorithm (Sect.2.2.1.). The induced underestimation features relate to the evaluation of $DOD_{fine}$ with $AOT_f$, an AERONET retrieval including non-dust fine-mode aerosol subtypes (e.g., biomass burning and urban haze), and to the evaluation of $DOD_{coarse}$ with $AOT_c$, an AERONET product including non-dust

coarse-mode aerosol subtypes (e.g., marine, pollen, volcanic ash).

Table 6: CALIPSO-based fine-mode (submicrometer) and coarse-mode (supermicrometer) DOD at 532 nm overall statistic metrics established on the basis of the reference AERONET fine-mode and coarse-mode AOTs at 532 nm, including absolute biases ($B_{abs}$), relative biases ($B_{rel}$), and Root Mean Square Error (RMSE), Correlation Coefficient, Slope ($S_{fit}$) and Interception

($I_{fit}$) of liner regression fit.

| Mode (N: 1737) | Evaluation Dataset | Reference Dataset | $B_{abs}$ | $B_{rel}$ (%) | RMSE | Cor. Coef. | $S_{fit}$ | $I_{fit}$ |
|---|---|---|---|---|---|---|---|---|
| Submicrometer | $DOD_{fine}$ | $AOT_{fine}$ | -0.012 | -7.38 | 0.059 | 0.692 | 0.652 | 0.018 |
| Supermicrometer | $DOD_{coarse}$ | $AOT_{coarse}$ | -0.036 | 9.55 | 0.076 | 0.916 | 0.779 | -0.002 |

Overall, the CALIPSO-based DOD–AERONET AOT evaluation comparison corroborates on the good performance of the lidar-based algorithms developed with the objective of decoupling the fine-mode, coarse-mode, and total pure-dust components of the total aerosol load (Shimizu et al., 2004; Mamouri and Ansmann, 2014; 2017; Tesche et al., 2009), the high quality of

the already established ESA-LIVAS pure-dust database, cornerstone of the present study (Amiridis et al., 2013; 2015; Marinou et al., 2017; Proestakis et al., 2018), and the quality of the established CALIPSO-based products of fine-mode and coarse-mode pure-dust atmospheric components, in terms of extinction coefficient profiles and DODs at 532 nm.

**3.2 Total, fine-mode, and coarse-mode Pure-Dust mass concentration product - validation against airborne**
**in-situ measurements**

In August 2015 a large-scale collaborative field campaign, the "Ice in Clouds Experiment–Dust" (ICE–D), including a subcomponent, the "AERosol properties–Dust" (AER–D) experiment, was conducted over the Eastern tropical Atlantic Ocean, in the broader region extending between Cabo Verde and Canary Islands. The main science objective of the AER–D campaign

evolved around a better characterization of airborne mineral dust optical and microphysical properties (Marenco et al., 2018), while ICE–D aimed at studying dust-cloud interactions and the efficiency of airborne mineral dust to act as CCN and IN (Liu et al., 2018). In the framework of AER-D and ICE-D sixteen research flights were carried out with the Facility for Airborne Atmospheric Measurements (FAAM) BAe-146 research aircraft. The research flight b920 conducted on August 7[th], 2015, aimed to perform highly spatially and temporally coordinated measurements with CATS onboard the ISS, targeting the

thorough validation of CATS products (McGill et al., 2015; Pauly et al., 2019; Yorks et al., 2016, 2019).
AER–D and ICE–D FAAM flights deployed remote sensing and in-situ instrumentation. Remote sensing measurements were conducted, amongst others, by a Leosphere ALS450 elastic backscatter lidar system, designed to acquire vertical profiles of aerosol and clouds at 355 nm in a nadir-viewing geometry. The system specifications are provided by Marenco et al. (2011) and Chazette et al. (2012), and the processing algorithms, starting from the level of the lidar beam returns to the determination

of the lidar ratio, the derivation of the extinction coefficient profiles, and the corresponding AOD values, are described in Marenco (2013), Marenco et al. (2014), and O'Sullivan et al (2020). In-situ measurements are described in Ryder et al. (2018)





and include aerosol size distribution measurements using a combination of a wing-mounted PCASP (Passive Cavity Aerosol Spectrometer Probe; Osborne et al., 2011), a CDP (Cloud Droplet Probe; Lance et al., 2010) and a 2DS (Two-Dimensional Stereo probe), sampling a combined size distribution from 0.1 to 100 μm in diameter. Here we use airborne lidar measurements were carried out during straight FAAM flights performed on a constant altitude between two locations ("runs"-R), and in-situ

of aerosol size distribution measurements from aircraft ascents and descents ("profiles"-P).

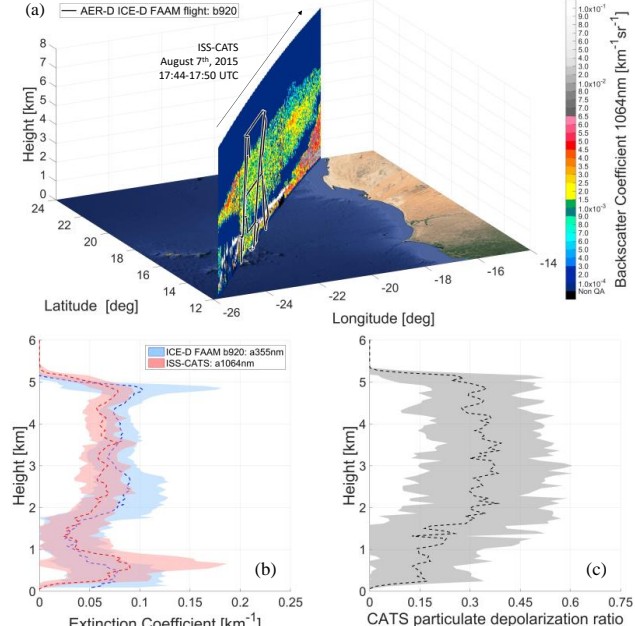

Figure 8: FAAM b920 – ISS-CATS underflight on the 7th of August 2015 in the proximity of Praia-Cabo Verde (a), FAAM b920 lidar and ISS-CATS mean extinction coefficient profiles at 355m and 1064nm respectively (b), and ISS-CATS mean particulate depolarization ratio profile at 1064nm (c). In panel (a) the aircraft track is shown in black.

Fig.8a shows the flight-track of b920 (black line), demonstrating both the spatially and temporally highly coordinated flight trajectory to the ISS orbit-track. The flight performed both airborne lidar measurements and air mass sampling at different altitudes. In addition, Fig.8a provides the structure of the atmospheric aerosol scene in the broader area extending between Cabo Verde and Canary Islands on the 7th of August 2015, as probed by ISS-CATS and based on L2 5km backscatter

coefficient at 1064 nm profiles (17:44-17:50 UTC). With respect to the vertical structure of the aerosol layers, the profiles of extinction coefficient obtained by the airborne lidar at 355 nm and by CATS at 1064 nm, in terms extinction coefficient, are in good agreement. Both extinction coefficient profiles report values in the range between 0.03 and 0.08 km-1, with FAAM-lidar values well within the variability of the atmospheric scene as observed by ISS-CATS. The aerosol observed by both the airborne and spaceborne lidar systems were dominated by dust, extending between 2 and 5 km a.m.s.l. in the Saharan Air

Layer (SAL), originating from convective activity over central Algeria, while a mixture of dust and marine and broken clouds was present within the Marine Boundary Layer (MBL). The CATS mean particulate depolarization ratio at 1064nm lies in the range between 0.3 and 0.4, confirming through the highly depolarizing optical properties of the non-spherical dust particles, the presence of dust aerosols in the atmosphere (Fig.8c). Thus, it is evident that the FAAM b920 and ISS–CATS collocated measurements were performed in presence of a deep dust layer within the Saharan Air Layer (SAL), transported westwards

by the predominantly easterly winds. The FAAM flight on August 7th, 2015, provides therefore an ideal case for establishing the quality of the satellite-based lidar fine-mode and coarse-mode pure-dust mass concentration products due to (1) the


implementation of both remote sensing and in-situ airborne measurements, (2) the highly collocated FAAM underflight of the ISS orbit-track, and (3) the significant presence of atmospheric dust within the SAL.

Towards the accurate derivation of the fine-mode and coarse-mode pure-dust mass concentration profiles, based on CATS backscatter coefficient and particulate depolarization ratio at 1064 nm, the implementation of suitable decoupling parameters

(Eq. (5) and Eq. (7)) and conversion factors (Eq. (13) and Eq. (14)) at the wavelength of 1064 nm is required. Towards this objective, the pure-dust $"\delta_d"$, coarse-mode pure-dust $"\delta_{cd}"$ and non-dust $"\delta_{nd}"$ particulate depolarization ratio values at 1064 nm are set equal to 0.27, 0.28 and 0.05, respectively (Freudenthaler et al., 2009; Haaring et al., 2017; Mamouri and Ansmann, 2017). CATS L2 5 km quality assured profiles between latitudes 14.61°N and 17.92 °N, following the latitudinal geographical coverage of FAAM b920 flight, were used. Accordingly, height profiles of pure-dust and coarse-mode pure-dust extinction

coefficient at 1064 nm are obtained (Eqs. (10) – (11)), by multiplying the pure-dust and coarse-mode pure-dust backscatter coefficients profiles at 1064 nm with a dust lidar ratio of 67 sr (Mamouri and Ansmann, 2017). To convert the obtained profiles of pure-dust and coarse-mode pure-dust extinction coefficient at 1064 nm into mass concentration profiles, conversion factors $"c_{v,d}"$ and $"c_{v,d_c}"$ equal to 0.73 and 0.72 are utilized, as calculated for Cabo Verde and Barbados in Mamouri and Ansmann (2017). As a final step, the profiles of fine-mode pure-dust mass concentration are estimated as the residual between the profiles

of total pure-dust mass concentration and the coarse-mode pure-dust mass concentration (Eq. (14)). Accordingly, the airborne in-situ mean aerosol size distribution in terms of mass concentration for aerosol particles with diameter between 0.1 and 100 μm is averaged based on P2 and P7 of the FAAM b920 flight, selected on the basis of the vertical extent, covering the full range of the SAL.

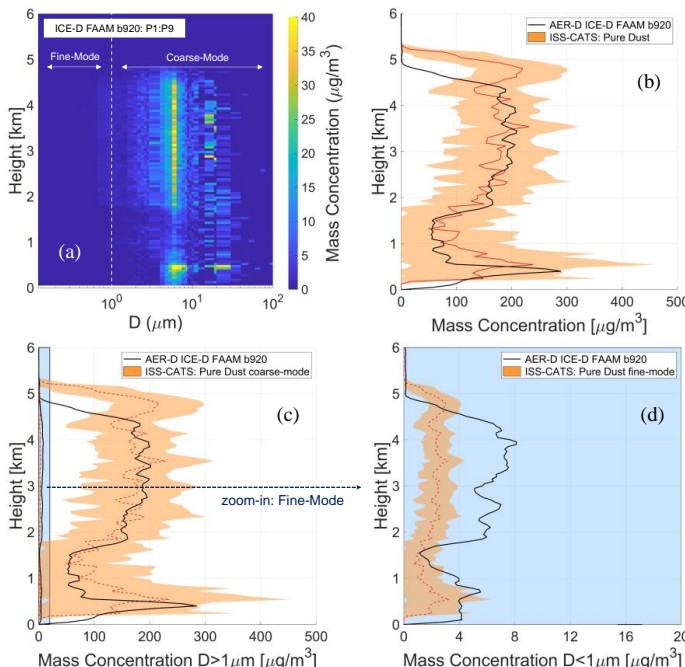

Figure 9: FAAM b920 airborne in-situ measurements of mass concentration in the proximity of Praia Island on the 7$^{th}$ of August 2015 based on (averaged) P2 and P7 (Fig.9a). FAAM b920 in-situ (black line) and ISS-CATS (orange line) mass concentration profiles are provided for the total (Fig.9b), supermicrometer (Fig.9c), and submicrometer atmospheric aerosol classes (Fig.9d).





Here, the fine-mode and coarse-mode pure-dust mass concentration profiles obtained based on ISS-CATS backscatter coefficient and particulate depolarization ratio at 1064 nm profiles through the separation technique (Sect.2.2) are validated against FAAM b920 airborne in-situ full aerosol size distribution (Fig.9). From the airborne in-situ measurements it is evident that the dust layer is dominated by particles of supermicrometer diameter, consisting ~99.73% by mass (Fig.9a). Moreover, it

is evident that CATS total pure-dust mass concentration and the airborne in-situ mass concentration profiles are in good agreement, both in the vertical extent and quantitatively, with values extending between 150 and 170 $\mu gm^{-3}$ within SAL and reaching as high as 250-300 $\mu gm^{-3}$ within the MBL (Fig.9b). The comparison is allowed even though the airborne in-situ measurements consist of both dust and non-dust aerosols, due to the dominant dust presence, as it is confirmed by the CATS particulate depolarization ratio at 1064 nm observations (Fig.8c). The high agreement between the CATS total pure-dust mass

concentration and the airborne in-situ mass concentration profiles is crucial, since discrepancies would not allow further intercomparison between the airborne fine-mode and coarse-mode profiles of mass concentration and the corresponding satellite-derived fine-mode pure-dust and coarse-mode pure-dust profiles of mass concentration (Fig.8c-d).

CATS estimated coarse-mode pure-dust concentration (163.3±31.8 $\mu gm^{-3}$; averaged over the altitude range 1.5 to 5 km a.m.s.l.) and the in-situ supermicrometer mass concentration (149.4±55.2 $\mu gm^{-3}$) are within 10% (Fig.9c), and very close to the total

in-situ mass concentration measured (154.8±57.1 $\mu gm^{-3}$). However, CATS fine-mode pure-dust concentration is underestimated (2.3±0.4 $\mu gm^{-3}$; averaged over the altitude range 0 to 1.5 km a.m.s.l.), ~58% lower of the airborne in-situ fine-mode mass concentration (5.4±2.1 $\mu gm^{-3}$) in SAL, although the performance increases significantly in the MBL region (Fig.9d). According to Ryder et al. (2018) and the provided size-resolved composition both in the SAL and the MBL for b920, no dust at sizes d<0.5 $\mu$m was present in the atmosphere between Cabo Verde and Canary Islands on the 7th of August 2015.

However, the fine-mode and coarse-mode dust decoupling algorithm is established on the basis of a boundary diameter separating the two modes of 1 $\mu$m, thus the low dust fine-mode concentrations apparent both in the SAL and the MBL for b920 most probably correspond to dust of diameter between 0.5 and 1 $\mu$m. Overall, CATS fine-mode pure-dust mass concentration underestimation may be partially attributed to the natural atmospheric inhomogeneity of the aerosol fields and partially to CATS deficiency to detect tenuous aerosol layers during sunlight illumination conditions due to low signal-to-noise ratio

(CATS 7.2 daytime minimum detectable backscatter 1064 nm: $1.30\times10^{-3} \pm 0.24\times10^{-3}$ km$^{-1}$sr$^{-1}$ for cirrus clouds; Yorks et al., 2016), introducing negative biases of the order of 20%−30% for daytime observations (Proestakis et al., 2019). However, the overall good performance of the fine-mode and coarse-mode pure-dust decoupling methodology is corroborated by the fine-mode to total mass concentration fractions, being in good agreement and of the same order of magnitude (CATS: ~1.4%, b920 in-situ: ~3.5%). Finally, both the spaceborne-lidar derived and the airborne fine and coarse modes are in good agreement with

respect to the vertical extent of the probed dust layer, characterized by similar profile characteristics, both in the SAL and MBL.

## 4 Three-dimensional distribution and temporal evolution of the fine-mode and coarse-mode components of atmospheric dust

The primary products of the present study consist of the fine-mode and coarse-mode components of atmospheric pure-dust, provided in quality-assured vertical profiles of the (1) backscatter coefficient at 532 nm, (2) extinction coefficient at 532 nm, and (3) mass concentration, with the original L2 horizontal (5 km) and vertical (60 m) resolution of CALIPSO, respectively (Sect.2.2.1., Fig.4). However, further processing of all L2 granules makes feasible the provision of an advanced four-

dimensional (4D) reconstruction of the total aerosol load, along with its components, at a near-global scale and over a long-term period. As such, the present section aims through the long-term horizontal, vertical, and temporal distributions of the two dust modes to demonstrate this potential of the established geo-information products, established with the objective to allow for further advancements of our EO-based capacity to observe and understand the submicrometer and supermicrometer modes of pure-dust in the atmosphere.





### 4.1 Horizontal distribution

With respect to the horizontal distribution of aerosol, the optical depth is the parameter most frequently used by spaceborne

passive sensors to quantify the columnar aerosol load under cloud-free sky conditions. Since CALIOP is not a passive sensor but an active system, in order to process the L2 extinction coefficient profiles into optical depth, specific steps have to be undertaken. More specifically, initially and for the needs of the present study, a near-global uniform grid of 1° latitude by 1° longitude is established (between 70°S and 70°N). Accordingly, the L3 processing algorithm iterates through all the CALIPSO L2 overpass granules within each grid and for the period between 06/2006 and 12/2021. As a next step, temporal averaging is

accomplished by averaging all the grid-aggregated quality-screened L2 extinction coefficient at 532 nm profiles within each grid of spatial resolution 1◦×1◦ over a time period of interest (e.g., of annual, seasonal, or monthly temporal resolution). Finally, the mean AOD, DOD, DOD$_{fine-mode}$, and DOD$_{coarse-mode}$ at 532 nm are computed by vertical integration of the gridded mean total aerosol, pure-dust, fine-mode pure-dust, and coarse-mode pure-dust extinction coefficient at 532 nm profiles, respectively. Figure 10 demonstrates the outcomes of these processing steps, and more specifically, the near-global annual-

mean horizontal distribution of the CALIPSO-based AOD at 532 nm in 1◦×1◦ spatial resolution (Fig.10a), the ESA-LIVAS DOD at 532 nm product (Amiridis et al., 2013; 2015; Marinou et al., 2017; Proestakis et al., 2018; Fig.10b), and the coarse-mode (Fig.10c) and fine-mode (Fig.10d) components of DOD at 532 nm. Moreover, Figure 10 provides the coarse-mode pure-dust fraction (CMF; coarse-mode pure-dust optical depth to total pure-dust optical depth fraction; Fig.10e) and the fine-mode pure-dust fraction (FMF; fine-mode pure-dust optical depth to total pure-dust optical depth fraction; Fig.10f), for areas of

annual-mean DOD at 532 nm greater than 0.01.

With respect to the optical depth outcomes, we begin with the examination of the near-global long-term horizontal distribution of CALIOP AOD at 532 nm, to initially provide the generic characteristics of the net effect of dust and non-dust aerosol subtypes in the atmosphere (Fig.10a). According to the observed AOD features, significant spatial variability is evident, especially over land and the Northern Hemisphere, where the major sources of natural aerosol and major sources of

anthropogenic aerosol are present (Winker et al., 2013). More specifically, mineral dust particles emitted from the vast arid and semi-arid regions of the planet (i.e., Saharan, Arabian, Taklimakan, Gobi deserts), biomass burning aerosols from tropical grasslands and forests (i.e., Amazonian region, sub-Sahel, Gulf of Guinea regions), volcanic emissions, sea-salt marine particles, and aerosol emissions related to anthropogenic activity, especially over the major urban, densely-populated, and heavily industrialized areas of the planet (i.e., East Asia, Indo-Gangetic Plain), result in high annual-mean AOD values,

frequently exceeding 0.6, apparent not only over the source regions, but over distances of thousands of kilometers downwind as well.

The effect of dust aerosol on optical depth, decoupled and isolated from the effect of all non-dust aerosol subtypes, is provided by the well-established ESA-LIVAS pure-dust product (Fig.10b). Overall, the pure-dust product of the ESA-LIVAS database has verified and demonstrated capacity to fully reveal the sources of dust, to quantitatively describe the three-dimensional

evolution of dust in the atmosphere, from source to sink, and to efficiently provide the seasonality of activation of dust source regions and the four-dimensional transition of dust transport pathways (Amiridis et al., 2013; Marinou et al., 2017; Proestakis et al., 2018; Aslanoğlu et al., 2022).

The effects of the coarse-mode and fine-mode components of pure-dust on optical depth, decoupled and isolated from the effect of all non-dust aerosol subtypes, are illustrated in Figs. 10c and 10d, respectively. In addition, Figure 11 provides the

seasonal coarse-mode (left column) and fine-mode (right column) components of DOD at 532 nm, grouped for "December-January-February (DJF)" (Fig.11a/b), "March-April-May (MAM)" (Fig.11c/d), "June-July-August (JJA)" (Fig.11e/f), and "September-October-November (SON)" (Fig.11g/h). The objective here is to provide and discuss through the coarse-mode



and fine-mode pure-dust products characteristics of dust sources and transport, reporting at the same time noticeable features of the datasets.

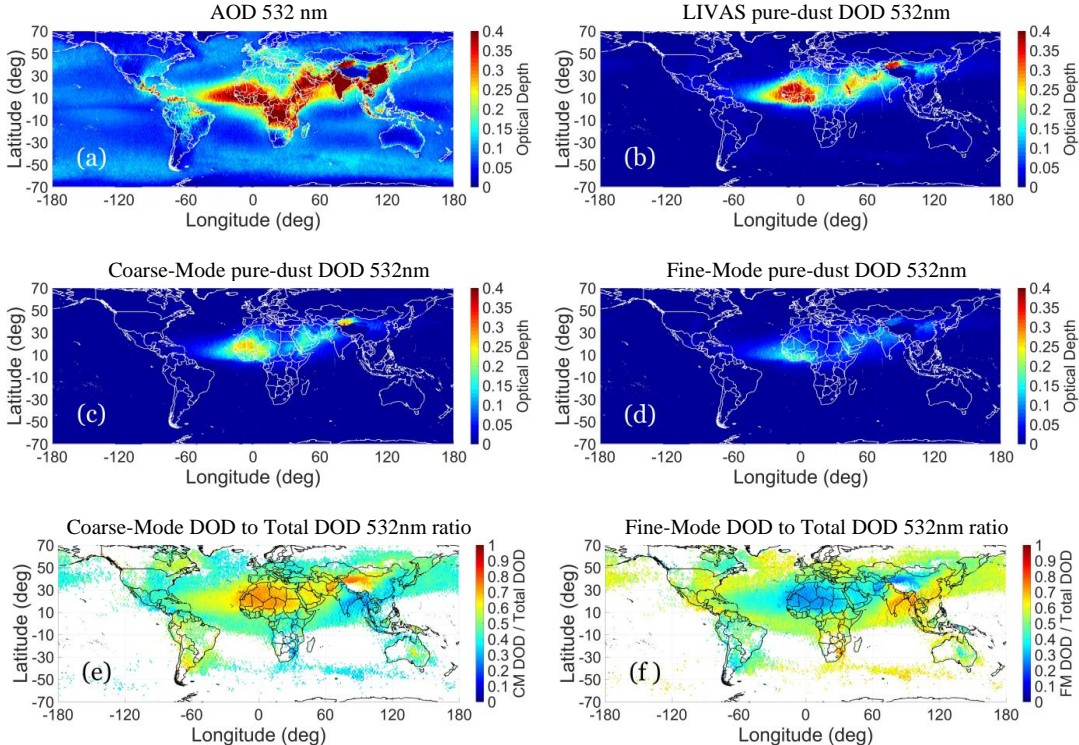

Figure 10: Near-global annual-mean horizontal distribution of the CALIPSO-based AOD at 532 nm (a), the ESA-LIVAS DOD
at 532 nm product (b), the coarse-mode DOD at 532 nm (c), the fine-mode DOD at 532 nm (d), the coarse-mode pure-dust
fraction (e), and the fine-mode pure-dust fraction (f), provided in 1°×1° spatial resolution and for the period 06/2006-12/2021.
In (e) and (f) DOD at 532 nm cases of load lower than 0.01 are not shown.

Towards these objectives, a generic apparent feature is that the horizontal distributions of both the $DOD_{coarse-mode}$, and $DOD_{fine-mode}$ are capable of efficiently delineating both the sources of dust and the dust transport pathways (Figs.10c/d). Overall, well-documented geographical patterns are evident in all four seasons, and although between the fine-mode and coarse-mode DODs few differences in terms of spatial distribution are revealed, the observed features significantly vary in magnitude. In general, the most intense values of the coarse-mode and fine-mode components of DOD at 532 nm are observed over the desert areas of the planet and especially over the dust belt, though the extent of the observed features highly varies with regional meteorology and seasonality (Fig.11).

With respect to the desert areas, the Saharan Desert is characterized by particularly intense loads of dust persistently present in the atmosphere throughout the year, though of remarkable spatial and intrannual variability. The expected west-to-east DOD gradient is apparent in both the $DOD_{coarse-mode}$ and $DOD_{fine-mode}$ products, with higher values evident to the west and lower values to the east of the domain. More specifically, over the central and west parts of the desert substantially high annual-mean $DOD_{coarse-mode}$ (~0.3) and $DOD_{fine-mode}$ (~0.12) values are observed, while over the central-east parts of the desert significantly lower annual-mean $DOD_{coarse-mode}$ (~0.14) and $DOD_{fine-mode}$ (~0.06) values are evident (Fig.10c/d). With respect to seasonality, remarkable intense loads of dust are observed primarily during JJA, as high as $DOD_{coarse-mode}$~0.6 and $DOD_{fine-mode}$~0.25 (Figs.11e/f), and secondarily during MAM, as high as $DOD_{coarse-mode}$~0.39 and $DOD_{fine-mode}$~0.17 (Figs.11c/d). In contrast,





during DJF (Figs.11a/b) and SON (Figs.11g/h) remarkably less dust activity is evident, almost uniformly over the entire Saharan Desert, with DOD$_{coarse-mode}$ and DOD$_{fine-mode}$ values not exceeding ~0.25 and ~0.15, respectively. The particularly intense loads of dust observed over the Saharan Desert, especially between March and August, are mainly attributed to the Saharan heat low and the west African monsoon system in spring and summer (Schepanski et al., 2017). At the same time, the latitudinal migration of the Intertropical Convergence Zone (ITCZ; Schneider et al., 2014) and the intensification of the trade winds (easterlies) favour the long-distance transport of massive loads of mineral dust aerosol across the tropical Atlantic Ocean within the SAL (Kanitz et al., 2014) as far as the Caribbean Sea (Prospero, 1999), as evident by both DOD$_{coarse-mode}$ and DOD$_{fine-mode\ products}$. Short-distance transport of fine-mode and coarse-mode dust layers originating from the Saharan Desert is also evident, especially over the Mediterranean Sea (Gkikas et al., 2013 ; 2015; 2016 ; 2022 ; Marinou et al., 2017; Aslanoğlu et al., 2022) under the impact of cyclone formation at the Gulf of Genoa - North African coast (e.g., Trigo et al., 1999; Maheras et al., 2001), over the Gulf of Guinea in the South (Ben-Ami et al., 2009), and over the Red Sea in the East (Banks et al., 2017; Li et al., 2018).

With respect to Middle East and Central Asia, dust activity is more pronounced in the dust-belt region extending between the eastern parts of the Saharan Desert and the Himalaya orographic barrier, a zone encompassing several major natural sources of dust, including the great Arabian Desert, the Thar Desert, and the arid and semi-arid regions of Ethiopia, Somalia, Iran, Iraq, and Afghanistan (Ginoux et al., 2012; Hamidi et al., 2013; Proestakis et al., 2018). These desert areas are clearly mapped through systematically high annual-mean DODcoarse-mode (~0.16) and DODfine-mode (~0.11) values (Fig.10c/d), though the observed features are subject to high spatial and intrannual variability (Fig.11). With respect to seasonality, remarkable intense loads of dust are observed primarily during JJA, as high as DODcoarse-mode~0.35 and DODfine-mode~0.23 (Figs.11e/f), and secondarily during MAM, as high as DODcoarse-mode~0.3 and DODfine-mode~0.15 (Figs.11c/d), although the activation mechanisms of the desert regions in the area differ (Prospero et al., 2002). More specifically, dust emission and transport over the Arabian Desert, Thar Desert, and the deserts of Ethiopia and Somalia is mainly attributed to the west Indian monsoon activity (Vinoj et al., 2014), while the relatively limited emission and transport of dust layers over Dasht-e Lut Desert and the arid areas of Iran, Iraq and Afghanistan is mainly attributed to convective episodes (Karami et al., 2017). With respect to the spatial variability, this zone is characterized by extensive inhomogeneities in the observed fine-mode and coarse-mode dust aerosol loads. Higher annual-mean values are evident over the Arabian Peninsula and the Middle East mainland (DOD$_{coarse-mode}$~0.15 and DOD$_{fine-mode}$~0.09) and lower values over the mainland of the southeast Asia and the broader Bay of Bengal (DOD$_{coarse-mode}$~0.03 and DOD$_{fine-mode}$~0.04). In addition, the expected and evident west-to-east gradient in the pure-dust products is attributed primarily to increased wet deposition of dust aerosol particles during the monsoon season of the year and secondarily to gravitational settling (Lau et al., 1988; Lau et al., 2006). The long-range transport of dust is more evident during MAM (Figs.11c/d) and JJA (Figs.11e/f) pre-monsoon seasons (Dey et al., 2004), when dust plumes emitted into the atmosphere are carried as far as the Indo-Gangetic Plain, and exceptionally as far as the Bay of Bengal (Mao et al., 2011; Proestakis et al., 2018; Ramaswamy et al., 2018).

To the east of the Himalaya orographic barrier, the dominant natural sources of mineral dust delineated by the fine-mode and coarse-mode pure-dust products are the very arid Taklimakan Desert in northwestern China (Liu et al., 2008; Ge et al., 2014) and the vast semi-arid Gobi Desert located eastwards in the same latitudinal band across northern China and Mongolia (Prospero et al., 2002; Gong et al., 2006; Proestakis et al., 2018). Over this zone of the dust belt, the favorable topographical features of the Tarim Basin and the strong midlatitude cyclonic systems which develop over the Mongolian Plateau (Bory et al., 2003; Yu et al., 2008) mobilize dust emission sources resulting to uplift and formation of elevated dust layers, especially during the spring and summer seasons (Proestakis et al., 2018), evident also by the high values of DOD$_{coarse-mode}$ (~0.3) and DOD$_{fine-mode}$ (~0.1). The dust layers suspended in the atmosphere are accordingly captured by the strong westerly jet in the upper troposphere and transported over the eastern Asia region (Zhang et al., 2003; Yu et al., 2019), across the broader northern Pacific Ocean (Duce et al., 1980; Shaw, 1980; Yu et al., 2008), and exceptionally as far as the western coast of the United



States (Husar et al., 2001). In contrast, during DJF (Figs.11a/b) and SON (Figs.11g/h) remarkable more limited dust activity is evident, almost uniformly over the entire East Asia region, as also shown by the pure-dust products, with $DOD_{coarse-mode}$ and $DOD_{fine-mode}$ values not exceeding ~0.25 and ~0.14, respectively.

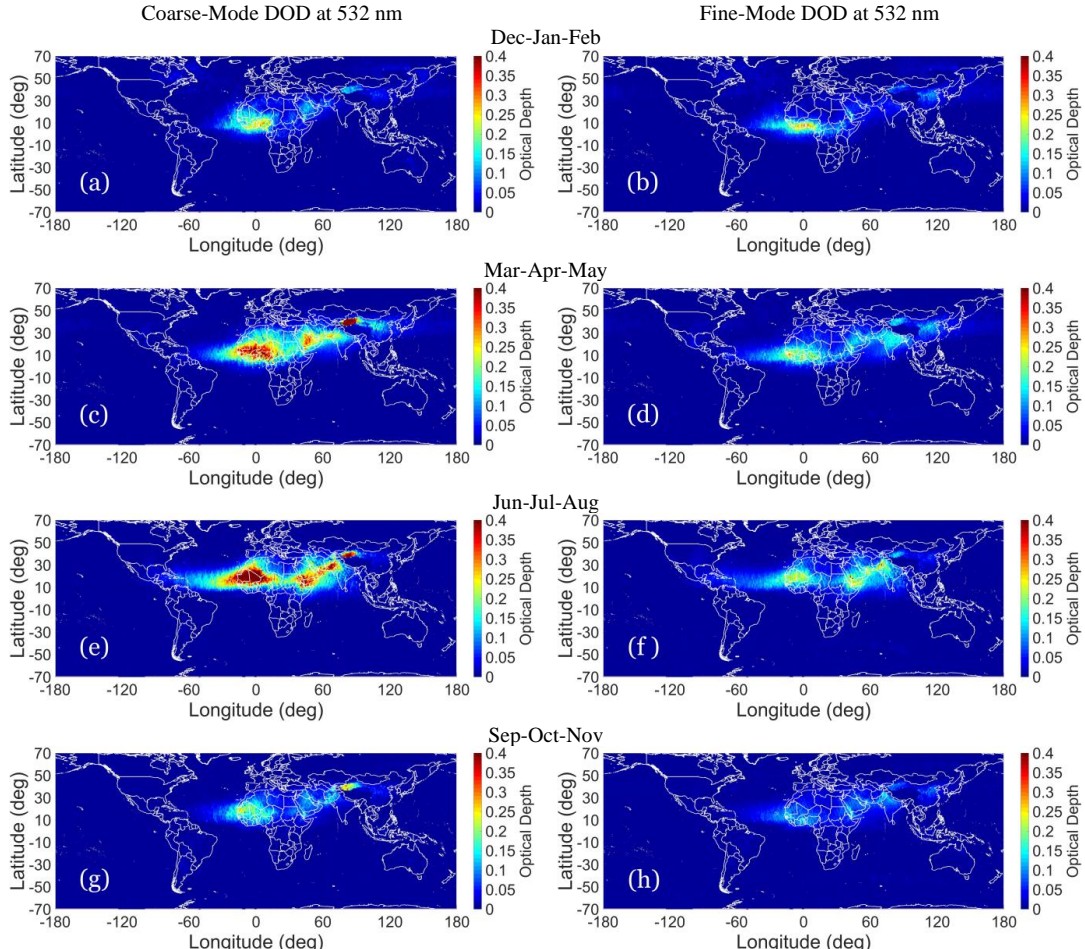

Figure 11: Seasonal coarse-mode (left column) and fine-mode (right column) components of DOD at 532 nm, grouped for "December-January-February (DJF)" (Fig.11a/b), "March-April-May (MAM)" (Fig.11c/d), "June-July-August (JJA)" (Fig.11e/f), and "September-October-November (SON)" (Fig.11g/h).

In addition to the dust-belt arid and semi-arid areas stretching from the Saharan Desert, through Middle East and Central Asia, to the Taklimakan and Gobi Deserts of East Asia, several other natural sources of dust are delineated and revealed by the pure-dust products, though of lower dust emission strength comparable to the major deserts of the planet. Such areas include the Mojave and Sonoran deserts of southwestern United States - northwest Mexico (Hand et al., 2017), and the Australian (Prospero et al., 2002), southern Africa (Bryant et al., 2007), and southern America (Gassó and Torres, 2019) desert areas located in the Southern Hemisphere. In contrast to the major sources of dust, over these areas remarkably more limited dust load is observed, with annual-mean $DOD_{coarse-mode}$ and $DOD_{fine-mode}$ values not exceeding ~0.03 and ~0.02, respectively.

Finally, two more noticeable features are worth mentioning. The first one relates to the preferential removal of larger dust particles during long-range transport, evident in both the CMF (Fig.10e) and the FMF (Fig.10f). In terms of optical depth,



close to the major natural dust sources, the supermicrometer component contributes to ~70% of the total dust load, falling to ~40% following removal processes during long-range transport of dust layers over distances of thousands of kilometers. This apparent feature corroborates with outcomes of the SALTRACE campaign, reporting that the fraction of removed dust particles is size-dependent, shape-dependent, and increases with increasing particle size (Weinzierl et al., 2016). The second one relates

to the sharp decrease (increase) of the CMF (FMF) over densely populated and heavily industrialized areas of the globe (i.e., Indian peninsula, China, and the broader South and East Asia region). Although subject to extended scientific discussion and hard to quantify, the current consensus is that atmospheric dust is composed of natural dust (~75%) and anthropogenic dust (~25%), a categorization established on the basis of dust sources and emission mechanisms (Penner et al. 1994; Tegen and Fung, 1995; Ginoux et al.; 2012). More specifically, anthropogenic activity over erodible soils or areas of little vegetation results in

locally emitted anthropogenic dust into the atmosphere, mainly confined within the PBL. The contribution of anthropogenic dust over areas of major anthropogenic activity results in perturbations of atmospheric dust accumulation load compared to the surrounding areas affected mainly by long-range transport of dust, a feature apparent in the modification of the fine-mode / coarse-mode to pure-dust fractions over the regions of extensive anthropogenic activity (Fig.10e/f).

**4.2 Vertical distribution and temporal evolution**

Emphasis in this climatology of the established products, however, is not only on the horizontal distribution of the fine-mode and coarse-mode components of pure-dust, as provided in terms of $DOD_{fine-mode}$ and $DOD_{coarse-mode}$ at 532 nm, respectively (Figures 10 and 11). The most unique information provided by CALIOP is the vertical structure of atmospheric constituents

detected along the CALIPSO orbit path. Vertically averaging all the CALIPSO-based L2 grid-aggregated quality-screened backscatter coefficients at 532 nm, extinction coefficients at 532 nm, and mass concentration profiles of the two-modes within each grid of spatial resolution 1∘×1∘ over a time period of interest (e.g., of annual, seasonal, or monthly temporal resolution), provides in addition to the horizontal distribution, the vertical distribution and temporal evolution of the two-modes, and allows the datasets to be established under a four-dimensional database. This capacity of the established products is illustrated in

Figure 12, based on the study case of the broader Indian subcontinent and the New Delhi megacity area. More specifically, Figure 12 provides the vertical structure of the coarse-mode (left-column) and fine-mode (right-column) components of atmospheric pure-dust, through long-term (15.5 yrs) annual-mean mass concentration cross sections over the Indian peninsula (Fig.12a/b), and timeseries of the temporal evolution of the coarse-mode (Fig.12c) and fine-mode (Fig.12d) pure-dust components, in terms of seasonal-mean mass concentration profiles, over the New Delhi megacity area, for the period 06/2006-

12/2021. Moreover, Figure 12 provides the coarse-mode pure-dust to total pure-dust ratio (Fig.12e) and the fine-mode pure-dust to total pure-dust ratio (Fig.12f). Based on the four-dimensional (4D) reconstruction of the atmosphere in terms of fine-mode and coarse-mode pure-dust components, as illustrated in Figure 12, several interesting characteristics of atmospheric dust are revealed, especially with respect to emission sources and transport, both in the FT and the PBL.

With respect to dust layers in the free-troposphere, the Indian subcontinent is largely affected by the major sources of dust

located to the west of the peninsula, including among others, the great Arabian, Dasht-e Lut, and Thar deserts (Hamidi et al., 2013; Proestakis et al., 2018; Gkikas et al., 2022). The layers of dust in the free-troposphere are mainly observed during the pre-monsoon season (Dey et al., 2004), and more specifically during MAM (Figs.11c/d) and JJA (Figs.11e/f), mainly attributed to the west Indian pre-monsoon activity (Vinoj et al., 2014), when dust plumes emitted into the atmosphere are carried as far as the Indo-Gangetic Plain, and exceptionally as far as the Bay of Bengal (Mao et al., 2011; Proestakis et al., 2018; Ramaswamy

et al., 2018). This seasonality of long-range transport of dust layers over India, originating from the major deserts located to the west of the peninsula, is revealed mainly by the supermicrometer mode of atmospheric pure-dust in terms of seasonal-mean mass concentration profiles over the New Delhi megacity area (Fig.12c). More specifically, the coarse-mode pure-dust product reveal an almost clear wave-like seasonal variation, with higher values observed during the pre-monsoon period





(Fig.12c). Moreover, due to the proximity of the New Delhi area to the major desert dust sources to the west, primarily to Thar desert and secondarily to Dasht-e Lut, dust aerosol transport over this area is revealed to take place not only in the free-troposphere but in the PBL as well. In addition, although the long-range transport of dust layers of natural origin occurs mainly in the free-troposphere, a fraction of this dust due to gravitational settling is expected to enter the PBL. Overall, the coarse-mode pure-dust product yields annual-mean mass concentration of $53.94\pm16.98$ μg/m$^3$, for the New Delhi megacity area, as computed based on the lower 1500 m of the atmosphere. Similarly computed, the coarse-mode pure-dust mass concentration over the New Delhi area is estimated for DJF equal $19.08\pm5.36$ μg/m$^3$, for MAM equal $86.88\pm18.72$ μg/m$^3$, for JJA equal $114.11\pm11.35$ μg/m$^3$, and for SON equal $47.44\pm10.15$ μg/m$^3$ (Table 7).

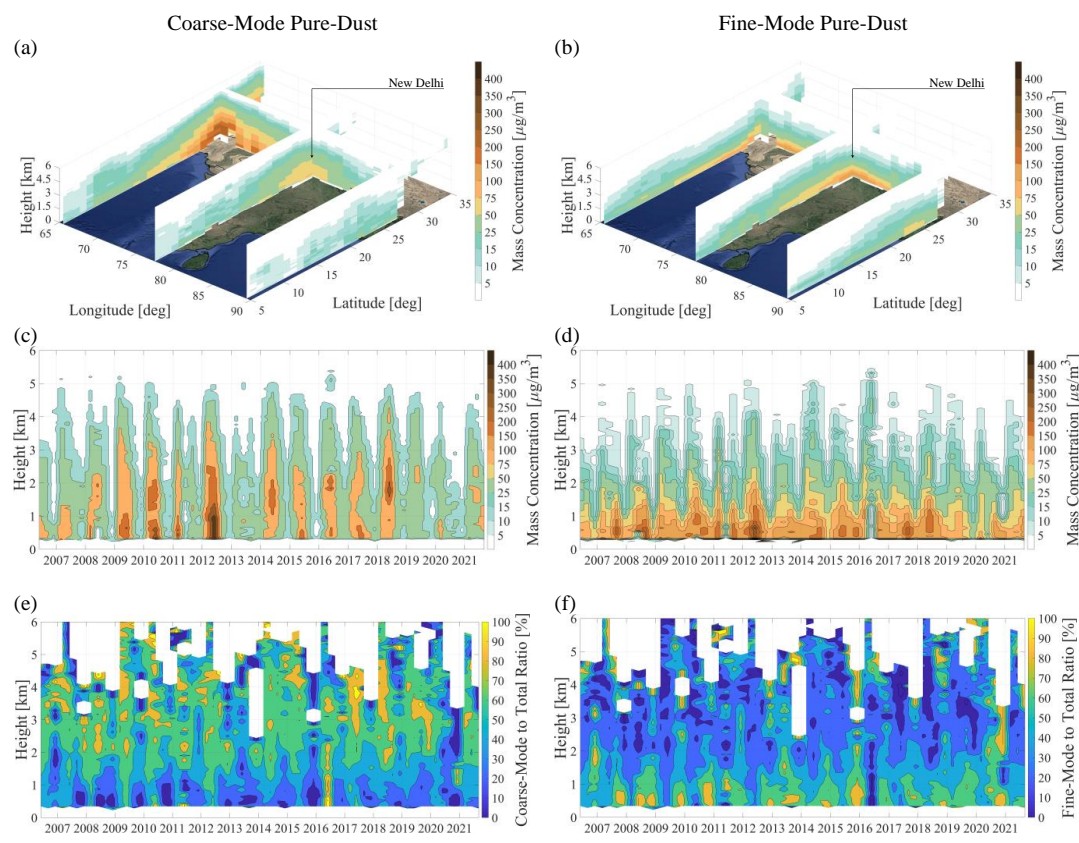

Figure 12: Coarse-mode (a) and fine-mode (b) pure-dust annual-mean mass concentration cross sections over the Indian peninsula (Lat: 28.5°N - Lon: 76.5°E - 06/2006-12/2021), seasonal-mean mass concentration time-series over the New Delhi megacity area (Lat: 27°-29°N - Lon: 76°-78°E) for the coarse-mode (c) and fine-mode (d), and coarse-mode pure-dust to total pure-dust ratio (e) and the fine-mode pure-dust to total pure-dust ratio (f) for New Delhi.

With respect to dust layers in the PBL, however, atmospheric dust is the net contribution of dust particles emitted and transported from areas outside the region of interest and dust particles emitted from within the region of interest, resulting in increased complexity with respect to atmospheric aerosol features. More specifically, the current consensus is that atmospheric dust is categorized into natural dust and anthropogenic dust, a classification established on the basis of dust sources and emission mechanisms (Penner et al. 1994; Tegen and Fung, 1995). Natural dust is suspended in the atmosphere by mechanisms that involve dust devils (Koch and Renno, 2005), "haboobs" (Knippertz et al., 2007), pressure gradients (Klose et al., 2010)





and low-level jets (LLJ; Fiedler et al., 2013), developed over dry and unvegetated desert regions (Prospero et al., 2002). Anthropogenic dust is related directly or/and indirectly to human activity, involving, among others, transportation, infrastructure, building and road construction (Moulin and Chiapello, 2006; Chen et al., 2018), and deterioration of extended soil surfaces and change in land use due to deforestation, grazing (Ginoux et al., 2012), urbanization, and agriculture (Tegen et al 1996). According to studies, the magnitude of atmospheric dust classified as anthropogenic is significant. Ginoux et al. (2012) reported that on a global scale ~25% of the total dust load is of anthropogenic origin. More recent, studies report that over densely populated and heavily industrialized urban areas of developing countries anthropogenic dust accounts to as much as ~70% of total dust load (Huang et al., 2015; Chen et al., 2019).

This is the case of the Indian peninsula (Huang et al., 2015; Guan et al., 2016; Chen et al., 2018). More specifically, long-term near-surface in-situ measurements of $PM_{2.5}$ conducted in the framework of the Surface PARTiculate mAtter Network (SPARTAN; https://www.spartan-network.org/; last access: 14/08/2023) over several megacities in the broader region (Delhi/Kanpur-India; Dhaka-Bangladesh) reported crustal material of high Zinc (Zn) to Aluminum (Al) ratios (Snider et al 2015, 2016). Zn is mainly of anthropogenic origin (Councell et al., 2004; Begum et al., 2010), whereas Al is mainly of natural origin (Zhang et al., 2006). As such, the high Zn:Al ratios detected in crustal materials consists of a clear indicator that a high percentage of fine-mode dust suspended within the PBL is of anthropogenic origin. The ongoing, continuous, and of high intensity anthropogenic activity over the Indian peninsula results in high emissions of anthropogenic dust all year round. This characteristic is revealed by the fine-mode pure-dust product, as utilized in order to provide the seasonal evolution of the submicrometer mode of atmospheric pure-dust in terms of mass concentration over the Delhi megacity area (Fig.12d). More specifically, the fine-mode pure-dust product reveals an almost continuous and of low seasonal variability behavior within the PBL, characterized by the presence of a fine-mode pure-dust background. Due to dust advection from natural sources and the variability of climate related variables such as soil moisture content and precipitation, perturbations and fluctuations in the fine-mode pure-dust component with the PBL are also present. Overall, the fine-mode pure-dust product yields annual-mean mass concentration of 76.55±34.16 µg/m3, for the New Delhi megacity area, as computed based on the lower 1500 m of the atmosphere more and based on more than 15-years of Earth-Observations. Similarly computed, the fine-mode pure-dust mass concentration over the New Delhi area is estimated for DJF equal 50.46±33.76 µg/m3, for MAM equal 79.27±30.05 µg/m3, for JJA equal 83.99±23.24 µg/m3, and for SON equal 96.95±42.75 µg/m3 (Table 7). Moreover, it is estimated that in the New-Delhi area the fine-mode pure-dust mass concentration values - approximately two days out of three - exceed the World Health Organization (WHO) Air Quality Guidelines (AQGs) for $PM_{2.5}$ at 24h-hour mean (https://www.who.int/publications-detail-redirect/WHO-SDE-PHE-OEH-06-02, last access: 23/08/2023).

Table 7: Annual- and seasonal- mean, median, and SD of the coarse-mode and fine-mode components of pure-dust within the lowest 1.5km of the atmosphere in the New-Delhi megacity area, including percentage of cases when the pure-dust fine-mode mean mass concentration exceeds WHO AQG for $PM_{2.5}$ – 24h-hour mean.

|  | Mode | Mean | SD | Median | Exceeding WHO AQG for $PM_{2.5}$ – 24h-hour mean |
|---|---|---|---|---|---|
|  |  | (µg/m$^3$) | (µg/m$^3$) | (µg/m$^3$) | (%) |
| Annual | coarse-mode | 53.94 | 16.98 | 57.82 | - |
|  | fine-mode | 76.55 | 34.16 | 77.95 | 66.11 |
| DJF | coarse-mode | 19.08 | 5.36 | 20.53 | - |
|  | fine-mode | 50.46 | 33.76 | 41.64 | 54.45 |
| MAM | coarse-mode | 86.88 | 18.72 | 90.12 | - |
|  | fine-mode | 79.27 | 30.05 | 81.92 | 76.39 |
| JJA | coarse-mode | 114.11 | 11.35 | 110.15 | - |
|  | fine-mode | 83.99 | 23.24 | 91.79 | 55.01 |
| SON | coarse-mode | 47.44 | 10.15 | 47.98 | - |
|  | fine-mode | 96.95 | 42.75 | 103.12 | 74.85 |



The reported two-modes pure-dust mass concentration product tackles several challenges, especially with respect to potential applications in addressing the impact of dust on human health. To date, several studies aiming to explore the negative effects of dust on air quality and human health frequently rely on passive remote sensing (De Longueville et al., 2010; Deroubaix et al., 2013; Katra et al., 2014; Prospero et al., 2014). However, it is considered particularly challenging to retrieve the elevation

and extent of dust aerosol layers residing in the atmosphere by means of passive remote sensing, hampering the ability to resolve the dust load within the PBL with high accuracy, where the main anthropogenic activity takes place. Even more significant is the fact that the majority of dust-related health disorders and their intensity depend primarily on dust PSD and secondarily on the total load of dust. More specifically, only the fine-mode of dust particles is inhalable deep into the lungs and alveoli, increasing morbidity and mortality rates among populations (Brook et al., 2010; Martinelli et al., 2013; Goudie et

al., 2014). Recent studies reveal that the size distribution of airborne dust particles range from 0.1 to more than 100 μm in diameter (Weinzierl et al., 2016; Ryder et al., 2018), reflecting also the limitations on addressing the impact of dust on induced health disorders by means of integrated parameters corresponding to the entire PSD, since only the inhalable component of dust is of high significance for air quality. Finally, atmospheric aerosol models extensively used to provide the spatiotemporal information support on dust emission, transport, deposition, and vertical structure (Textor et al., 2006; Astitha et al., 2012;

Randles et al., 2017; Inness et al., 2019), usually employ static land cover types to classify arid and semi-arid regions as dust sources (Ginoux et al., 2001). However, the implementation of static emission inventories leads to large uncertainties, especially in regard to non-considered anthropogenic dust emission fluxes (Ginoux et al., 2012) over highly-industrialized and densely-populated regions of the Earth, leading to underestimations on the risk factor of dust impact on human health. Thus, the fine-mode pure-dust product established here is considered of paramount importance, since it offers pure-observational

evidence on the fine-mode component of dust residing within the PBL and over extended geographical regions and temporal periods, thus enhancing our EO-based capacity to assess and understand the complex role of inhalable dust particles to induced disorders on human health.

**5. Summary and conclusions**

A new, multiyear, and near-global reconstruction of the Earth's atmosphere, in terms of fine-mode and coarse-mode components of atmospheric pure-dust, is provided. The primary datasets consist of the submicrometer (particles with diameter less than 1 μm) and supermicrometer (particles with diameter greater than 1 μm) modes of atmospheric pure-dust, provided in quality-assured profiles of (1) backscatter coefficient at 532 nm, (2) extinction coefficient at 532 nm, and (3) mass

concentration, with the original L2 horizontal (5 km) and vertical (60 m) resolution of CALIOP, along the CALIPSO orbit-path (Winker et al., 2009). Further processing of all CALIPSO-based L2 quality-screened backscatter coefficient at 532 nm, extinction coefficient at 532 nm, and mass concentration fine-mode and coarse-mode pure-dust profiles within a pre-defined grid (e.g., here of spatial resolution 1∘×1∘) and over a selected time period of interest (e.g., here of annual and seasonal temporal resolution), allows the primary datasets to be established under a four-dimensional climate data record, and synergistically to

provide the horizontal distribution, vertical distribution, and temporal evolution of the two-modes of atmospheric pure-dust. There are three main enablers that lie behind the established fine-mode and coarse-mode pure-dust database. The first one lies in laboratory studies reporting on the distinct light-depolarizing properties of the submicrometer and supermicrometer modes of pure-dust (Sakai et al., 2010; Järvinen et al., 2016) and that allows the separation of the two components from depolarization lidar. The second one lies in the well-established family of POLIPHON algorithms, and more specifically, on the potential of

the one-step POLIPHON method to decouple the pure-dust component from the total aerosol load (Shimizu et al., 2004; Tesche et al., 2009) and on the potential of the two-step POLIPHON method to decouple the two-modes of pure-dust (Mamouri and Ansmann, 2014; 2017; Ansmann et al., 2019). The third and final one lies in the simplicity of the pure-dust retrieval algorithms. Straightforward and applicable to single-wavelength lidar observations, the algorithms can be utilized in satellite-based EO as





long as profiles of backscatter coefficient and particulate depolarization ratio are provided. To date, the pure-dust decoupling methodology (Shimizu et al., 2004; Tesche et al., 2009) has been applied towards the development of a robust, multiyear, four-dimensional, and near-global pure-dust dataset, established in the framework of the ESA-LIVAS database (Amiridis et al., 2013; 2015; Marinou et al., 2017; Proestakis et al., 2018), demonstrating the feasibility of applying the polarization-based

algorithms to satellite-based EO products.

Motivated by these studies, the present work provides, for the first time, a multiyear near-global four-dimensional representation of the fine-mode and coarse-mode components of atmospheric pure-dust. To this point, the work includes an overview of the methodology, assumptions, and adaptations made, primarily in order to facilitate the applicability of the technique to CALIPSO-CALIOP and ISS-CATS optical products and special characteristics, and secondarily to allow the

implementation of the ESA-LIVAS pure-dust product without introducing ambiguities to the developed datasets. Overall, the core concept of the applied methodology evolves around the assumption that the fine-mode pure-dust backscatter coefficient at 532 nm is the residual between the ESA-LIVAS pure-dust backscatter coefficient at 532 nm (Amiridis et al., 2013; 2015; Marinou et al., 2017; Proestakis et al., 2018), decoupled from the total aerosol load based on the pure-dust decoupling methodology (Shimizu et al., 2004; Tesche et al., 2009), and the pure-dust coarse-mode backscatter coefficient at 532 nm,

decoupled from the total aerosol load based on the first-step of the two-step POLIPHON (Mamouri and Ansmann, 2014; 2017). Accordingly, the study implements suitable regionally-dependent lidar-derived pure-dust LRs to convert the total, fine-mode, and coarse-mode pure-dust profiles of backscatter coefficient at 532 nm to total, fine-mode, and coarse-mode pure-dust profiles of extinction coefficient at 532 nm, along the CALIPSO orbit-path at uniform 5 km horizontal and 60 m vertical resolution. As a follow up step, the obtained total and coarse-mode extinction coefficient profiles are converted into respective mass

concentration profiles through the implementation of characteristic regionally-dependent volume concentration conversion factors (Ansmann et al.; 2019). As a final step, the fine-mode pure-dust mass concentration profiles are extracted as the residual between the total pure-dust mass concentration profiles and the coarse-mode pure-dust mass concentration profiles.

The quality of the dust products is justified by using AERONET fine-mode and coarse-mode AOTs converted at 532 nm and AER-D campaign airborne in-situ aerosol size distributions as reference datasets, during atmospheric conditions characterized

by dust presence. More specifically, match-up reference datasets used as a basis to ensure the assessment of the range of validity and robustness of the products, include both existing climatologies and in-situ field campaign observations. As a first step, the CALIPSO-based $DOD_{coarse}$ and $DOD_{fine}$ at 532 nm products were evaluated against AERONET $AOT_{coarse}$ and $AOT_{fine}$ at 532 nm retrievals. Evaluation was conducted for identified cases dominated by the presence of dust, as reported by the ESA-LIVAS pure-dust product. With respect to the submicrometer category, a fair good agreement between the two datasets was

revealed (slope: 0.652, offset: 0.018, Pearson's correlation coefficient: 0.692), although a strong tendency of increasing fine-mode pure-dust underestimation with increasing dust aerosol load was also apparent. With respect to the supermicrometer category, the analysis revealed a substantially better agreement between the two datasets (slope: 0.779, interception: -0.002, Pearson's correlation coefficient: 0.916). As a second step, the AER-D FAAM b920 research flight conducted on August 7th, 2015 with the objective to perform highly spatially and temporally coordinated measurements with the CATS lidar onboard

the ISS, was utilized. The case is ideal for establishing the quality of the satellite-based lidar fine-mode and coarse-mode pure-dust mass concentration products and assessing their accuracy due to (1) the implementation of both remote sensing and in-situ airborne measurements by FAAM b920, (2) the collocated FAAM underflight of the ISS orbit-track, and (3) the significant presence of atmospheric dust within SAL. The comparison against the airborne in-situ PSD revealed that the developed fine-mode and coarse-mode mass concentration profile products have the capacity to provide, both qualitatively and quantitatively,

the structure of atmospheric dust layers in terms of the two-dust modes. With respect to the supermicrometer category, agreement within 10% (and well within the uncertainties) was revealed between the coarse-mode pure-dust mass concentration product (163.3±31.8 μgm⁻³) and the reference in-situ supermicrometer mass concentration (149.4±55.2 μgm⁻³). With respect to the submicrometer category, the fine-mode pure-dust mass concentration product was underestimated (2.3±0.4 μgm⁻³),





~58% lower than the airborne in-situ fine-mode mass concentration (5.4±2.1 μgm$^{-3}$) in the SAL, while the performance was substantially better in the MBL region. The overall good performance of the decoupling methodology is further corroborated by the fine-mode to total mass concentration fraction, being in good agreement and of the same order of magnitude (CATS: ~1.4, b920 in-situ: ~3.5).

The demonstration of the fine-mode and coarse-mode climatological database involved two main discussion aspects. The first one aimed to present and briefly discuss the capacity of the established database to provide the horizontal distribution of the fine-mode and coarse-mode components of pure-dust, in terms of optical depth. Towards this objective, the annual and seasonal DOD$_{coarse-mode}$ and DOD$_{fine-mode}$ patterns at a near-global scale (70°S-70°N) and for the period extending between 06/2006 and 12/2021, were discussed. The second one aimed to illustrate the protentional of the established datasets to reveal the full four-

dimensional structure of the two-modes of dust (horizontal, vertical, temporal), based on the Delhi megacity as part of the broader Indian peninsula. For this purpose, annual-mean cross-sections and the seasonal-mean temporal evolution of the two-modes of pure-dust, in terms of mass concentration profiles, were discussed. Overall, the cases demonstrated the potential of the fine-mode and coarse-mode pure-dust products (1) to reveal the natural and anthropogenic sources of dust, (2) to both qualitatively and quantitatively describe the three-dimensional evolution of dust in the atmosphere, from source to sink, to (3)

efficiently provide the seasonal activation cycle of dust source regions, and (4) the seasonal shift and four-dimensional transition of dust transport pathways.

The present work aims through the presented fine-mode and coarse-mode pure-dust datasets to contribute to the next generation of atmospheric aerosol geo-information products. Future work includes further optimization of the fine-mode and coarse-mode pure-dust climate database through revision of the basic assumptions (i.e., DeLiAn database; Floutsi et al., 2023), while long-

term overarching objective is the development of a novel and unprecedented climate data record of the two-modes of dust suspended in the atmosphere, through the integration of the fine-mode and coarse-mode dust products derived by future Earth Explorer satellite missions (i.e., ESA-EarthCARE and NASA-AOS I1/I2/P1). At present, the CALIPSO-based near-global datasets cover a multi-year period spanning between June 2006 and December 2021. Exploitation of the dust datasets facilitates - among others - dust climatological studies, time-series, and trend analysis over extensive geographical domains, validation

of atmospheric dust models and reanalysis datasets, assimilation activities, investigation of the role of airborne dust on radiation, quantification and decoupling of the natural-anthropogenic dust system, and an advanced database to address the effect of the inhalable component of airborne dust on air quality. Overall, the present database envisages to provide a support system that will allow advancements on our EO-based capacity to assess and understand the impact of dust on climate, environmental conditions, and human health under the ongoing Climate Change.

### Data availability

The CALIPSO lidar level 1B and level 2 data products are publicly available from the Atmospheric Science Data Center at NASA Langley Research Center (https://earthdata.nasa.gov/eosdis/daacs/asdc, last access: 24/08/2023). The FAAM aircraft

datasets collected during the AER-D campaign is available from the British Atmospheric Data Centre, Centre for Environmental Data Analysis (http://www.ceda.ac.uk/; last access: 24/08/2023). CATS browsed images and data products are freely distributed via the CATS website at http://cats.gsfc.nasa.gov/data/ (last access: 24/08/2021). The LIVAS Pure-Dust database is available upon personal communication with E. Proestakis (proestakis@noa.gr), E. Marinou (elmarinou@noa.gr), and/or V. Amiridis (vamoir@noa.gr). The L2 pure-dust fine-mode and coarse-mode data set is available at

10.5281/zenodo.10389741.


**Author contributions**

EP designed the study. AG treated AERONET data and provided assistance on interpretation of the validation outcomes. TG provided critical assistance in algorithm development and processing. AK provided FLEXPART back-trajectory analysis and interpretation of outcomes. ED provided assistance in the treatment of airborne observations. CLR and FM provided the data from the airborne in situ measurements. VA, EM, and EP provided the LIVAS dataset and helped in interpretation of the results. EP wrote the manuscript. EP, AG, and TG developed the codes. EP and AG analyzed the results. AG, AG, TG, AK, ED, CLR, FM, EM, and VA provided critical feedback and reviewed and edited the manuscript.

**Competing interests.**

At least one of the (co-)authors is a member of the editorial board of AMT.

**Acknowledgements**

E. Proestakis acknowledges support by the AXA Research Fund for postdoctoral researchers under the project entitled "Earth Observation for Air-Quality - Dust Fine-Mode (EO4AQ-DustFM)". We would like to thank the NASA CALIPSO team and NASA/LaRC/ASDC for making the CALIPSO products available, which have been extensively used in the present study. We would like to thank ESA for supporting the LIVAS developments (contract no. 4000104106/11/NL/FF/fk). We thank the PANhellenic GEophysical observatory of Antikythera (PANGEA) for providing access to the LIVAS data used in this study and their computational center. CLR was funded by NERC grant NE/M018288/1. A. Gkikas acknowledges support by the Hellenic Foundation for Research and Innovation (H.F.R.I.) under the "2nd Call for H.F.R.I. Research Projects to support Post-Doctoral Researchers" (project acronym: ATLANTAS, project number: 544). Airborne data were obtained using the United Kingdom BAe-146 Atmospheric Research Aircraft, which at the time was flown by Directflight Ltd., managed by the Facility for Airborne Atmospheric Measurements (FAAM), and was a joint entity of the Natural Environment Research Council (NERC) and the Met Office. The staff of the Met Office Observations Based Research team, the universities of Leeds, Manchester, and Hertfordshire, FAAM, Directflight Ltd, Avalon Engineering, BAE Systems are thanked for their dedication in making AER-D a success.

**Financial support**

The research study was supported and funded by AXA Research Fund for postdoctoral researchers under the project entitled "Earth Observation for Air-Quality - Dust Fine-Mode (EO4AQ-DustFM)". The Met Office (Exeter, United Kingdom) funded and supported the AER-D campaign. The LIVAS climate data record was established under the auspices of the ESA-ESTEC project: Lidar Climatology of Vertical Aerosol Structure for Space-Based LIDAR Simulation Studies (LIVAS) contract no. 4000104106/11/NL/FF/fk.

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
