# Peer review of "A near-global multiyear climate data record of the fine-mode and coarse-mode components of atmospheric pure-dust"

_Atmospheric Measurement Techniques, 2023_

## Referee Comment (RC1)

This study is ambitious. The authors extend a well-established technique for separating the fine and coarse modes of dust particles, based on AERONET data and lidar polarization measurements. In this manuscript, the authors adapt this approach for lidar measurements from space. Estimating dust particle mass concentration from an elastic backscatter lidar involves numerous assumptions about region-dependent lidar ratios, conversion factors, etc., which the authors acknowledge and discuss thoroughly. It's worth noting that the authors are well-known experts in lidar measurements. The manuscript is well-written, includes detailed introduction, long reference list, and is definitely suitable for publication in AMT. I have just several technical comments.

p.10 ln.19 "urban haze and biomass-burning smoke, with depolarizing effects of 1-4% at 532 nm". Actually it can be much higher. Depolarization of smoke and urban particles can be up to 10%. (Veselovskii, I., Hu, Q., Goloub, P., Podvin, T., Barchunov, B., and Korenskii, M.: Combining Mie–Raman and fluorescence observations: a step forward in aerosol classification with lidar technology, Atmos. Meas. Tech., 15, 4881–4900, 2022. https://doi.org/10.5194/amt-15-4881-2022) and references therein. In upper troposphere depolarization of smoke at 532 nm can reach 15%. All this introduces uncertainties in separation of dust and smoke. Depolarization of pollen at 532 nm can reach 35%.

p.10 ln 22. "(Noh et al.; 2013)". More recent references about pollen depolarization should be added.
Bohlmann, S., Shang, X., Giannakaki, E., Filioglou, M., Saarto, A., Romakkaniemi, S. and Komppula, M.: Detection and characterization of birch pollen in the atmosphere using multi-wavelength Raman lidar in Finland, Atmos. Chem. Phys. 19, 14559–14569, 2019. doi.org/10.5194/acp-19-14559-2019.

Atmos. Chem. Phys., 21, 7083–7097, 2021  https://doi.org/10.5194/acp-21-7083-2021

p.11 ln.1. " In Eq. (6), "$\beta_{\lambda,ncd}(z)$" and "$\beta_{\lambda,cd}(z)$" correspond to the non-coarse-mode aerosol (i.e., non-dust and fine-mode dust)" I am confused. Because in p.11 ln.22 authors write "we assume mean linear depolarization effects of "$\delta_{ncd}$" and "$\delta_{cd}$" equal to $0.16 \pm 0.02$ and $0.39 \pm 0.03$" Does it mean that smoke (it is non-dust) has depolarization 16%? These definitions should be clarified.

Fig.2. Depolarization 1.0 is confusing. Reader may have feeling that dust depolarizations extends up to 1.0

Table 4. The reference should be added:
Atmos. Meas. Tech., 16, 1951–1970, 2023 https://doi.org/10.5194/amt-16-1951-2023

Fig.5b. Labels on right axes is difficult to read.

---

## Author Comment (AC1)

This study is ambitious. The authors extend a well-established technique for separating the fine and coarse modes of dust particles, based on AERONET data and lidar polarization measurements. In this manuscript, the authors adapt this approach for lidar measurements from space. Estimating dust particle mass concentration from an elastic backscatter lidar involves numerous assumptions about region-dependent lidar ratios, conversion factors, etc., which the authors acknowledge and discuss thoroughly. It's worth noting that the authors are well-known experts in lidar measurements. The manuscript is well-written, includes detailed introduction, long reference list, and is definitely suitable for publication in AMT. I have just several technical comments.

The authors would like to thank the reviewer for his time, comments and suggestions. We did our best to incorporate the proposed changes and corrections in the revised manuscript, aiming at improving the presented paper. Following, you will find our responses, one by one to the comments addressed.

Kind regards,

Emmanouil Proestakis

1. p.10 ln.19 "urban haze and biomass-burning smoke, with depolarizing effects of 1-4% at 532 nm". Actually, it can be much higher. Depolarization of smoke and urban particles can be up to 10%. (Veselovskii, I., Hu, Q., Goloub, P., Podvin, T., Barchunov, B., and Korenskii, M.: Combining Mie–Raman and fluorescence observations: a step forward in aerosol classification with lidar technology, Atmos. Meas. Tech., 15, 4881–4900, 2022. https://doi.org/10.5194/amt-15-4881-2022) and references therein. In upper troposphere depolarization of smoke at 532 nm can reach 15%. All this introduces uncertainties in separation of dust and smoke. Depolarization of pollen at 532 nm can reach 35%.

The authors agree with the reviewer, the section of the depolarization ratio of the aerosol subtype classes considered in the study could be enhanced by including the research outcomes reported by Veselovskii et al. (2022) and Bohlmann et al. (2021). The paragraph was re-written in order to provide more information. Below we provide the part of the section which was vastly rephrased and extended.

[revised manuscript text omitted]

We agree with the reviewer that the manuscript would improve by including the recommended reference. Added in the manuscript.

3. p.11 ln.1. "In Eq. (6), "$\beta_{\lambda,ncd}(z)$" and "$\beta_{\lambda,cd}(z)$" correspond to the non-coarse-mode aerosol (i.e., non-dust and fine-mode dust)" I am confused. Because in p.11 ln.22 authors write "we assume mean linear depolarization effects of "$\delta ncd$" and "$\delta cd$" equal to 0.16 ± 0.02 and 0.39 ± 0.03" Does it mean that smoke (it is non-dust) has depolarization 16%? These definitions should be clarified.

The two-step POLIPHON technique (Mamouri and Ansmann, 2014; 2017) here is applied to the CALIPSO dusty aerosol subtypes (Winker et al., 2010), hence to the "dust", "dusty-marine", and "polluted dust" (Omar et al., 2009; Kim et al., 2018). Hence, since the aerosol subtypes of interest are the dusty mixtures, it is assumed always to have a dust component and a non-dust aerosol component in the aerosol layer, present as external mixtures (Tesche et al., 2009; Ansmann et al., 2019). Here, as complement aerosol subtypes to the dust aerosol subtype (total, fine-mode, and coarse-mode dust) in the CALIPSO-based dusty mixtures are considered (1) the marine aerosol class in the "dusty-marine"

aerosol subtype mixture and (2) the smoke or pollen or volcanic ash or urban or continental aerosol classes in the "polluted-dust" aerosol subtype mixture (Amiridis et al., 2013; Marinou et al., 2017; Proestakis et al., 2018). More specifically, the two-step POLIPHON technique assumes that the backscattered signal by an external aerosol mixture "$\beta_{\lambda,p}(z)$" corresponds to the summation of the cross and parallel return signals from the (1) non-dust, (2) fine-mode dust, and (3) the coarse-mode dust aerosol components, as discussed above (Mamouri and Ansmann, 2014; 2017). Thus, with respect to the implementation of the first-step of the two-steps of the POLIPHON technique, towards determination of the atmospheric coarse-mode dust aerosol component, knowledge of the non-coarse-mode dust aerosol and coarse-mode dust light-depolarization characteristics, thus of "$\delta_{ncd}$" and "$\delta_{cd}$" respectively, is required. The "$\delta_{ncd}$" corresponds to the particulate depolarization ratio of the non-coarse-mode dust component, thus for the aforementioned terms (#1) and (#2). The "$\delta_{cd}$" term corresponds to the particulate depolarization ratio of the coarse-mode dust aerosol component, thus of the aforementioned term (#3).

Since "$\delta_{ncd}$" includes broader aerosol non-dust aerosol subtype classes and since the CALIPSO aerosol subtype classification algorithm does not provide information on the complement-to-dust aerosol subtype class in the aerosol mixture, in order to apply the two-step POLIPHON technique a representative universal mean particulate depolarization ratio at 532 nm for the non-dust aerosol subtype and for the fine-mode dust particulate depolarization ratio at 532 nm have to be included. Following an increasing number of observations reporting on the particulate depolarization ratio at 532 nm of different aerosol subtype classes (i.e., DeLiAn database; Floutsi et al., 2023 and references therein) and laboratory experiments reporting on the particulate depolarization ratio at 532 nm of the fine- coarse- mode of dust (Sakai et al., 2010; Järvinen et al., 2016) assumptions are made on "$\delta_{ncd}$" and "$\delta_{cd}$". According to the published outcomes of the studies, the term "$\delta_{ncd}$", the total effect on depolarization of the fine-mode of dust and the non-dust aerosol component (i.e., marine or smoke or pollen or volcanic ash or urban or continental) in the external aerosol mixture is assumed of ~0.16 ± 0.02 in the two-step POLIPHON technique applied to optical products of CALIPSO (Sect. 2.2.).

4. Fig.2. Depolarization 1.0 is confusing. Reader may have felt that dust depolarizations extend up to 1.0.

The authors have followed the official CALIPSO conversions in terms of colorbars, colormaps, and scale of CALIOP optical products. An indicative example is provided here-in-after, for the Godzilla dust transport event over the North Atlantic Ocean (June 2020), as provided by CALIPSO lidar browse images. In terms of depolarization ratio at 532 nm the official and suggested scale extends from lower than 0.0 to higher values than 1.0. This is the reason behind the lower limit of 0.0 and the upper limit of 1.0 in Fig.2.

(a)

[Figure]

[Figure]

Figure 01: CALIPSO nighttime granule (a), Total Attenuated Backscatter at 532 nm (b), and Depolarization Ratio at 532 nm (c) for the Godzilla dust transport event on the 18th of June 2020.

5. Table 4. The reference should be added: Atmos. Meas. Tech., 16, 1951–1970, 2023 https://doi.org/10.5194/amt-16-1951-2023.

We agree with the reviewer that the manuscript and the discussion would improve by including the recommended reference. Reference is added in the manuscript.

6. Fig.5b. Labels on right axes is difficult to read.

According to the reviewer's comment, the figure is changed as follows:

[Figure]

[Figure]

Figure 5: Major Saharan dust outbreak moving westwards over Dakar on the 20[th] of April, 2017 at ~14:43 UTC, including the CALIPSO overpass in the proximity of the AERONET-Dakar station (red line) and FLEXPART 6-day back-trajectories at 2 km at the area of interest (Lat: 14.39, Lon: -16.95) denoting the Saharan desert origin of the advected air masses (yellow line) (Fig.5a). CALIPSO L2 5 km backscatter coefficient at 532 nm cross section (Fig.5b). Column-integrated particle volume concentration as a function of particle radius observed with the AERONET sunphotometer over Dakar, Senegal (Fig.5c). Backscatter coefficient at 532 nm profiles of the coarse-mode pure-dust (red shaded area), fine-mode pure-dust (blue shaded area), and non-dust (gray shaded area) components of the total aerosol load (Fig.5d). Particulate depolarization ratio at 532 nm profile used for the decoupling of the coarse-mode pure-dust, fine-mode pure-dust, and non-dust components of the total aerosol load, as provided in Fig.5d (Fig.5e). Extinction coefficient at 532 nm profiles of the coarse-mode pure-dust (red shaded area) and fine-mode pure-dust (blue shaded area) components of the total aerosol load (Fig.5f).

---

## Author Comment (AC2)

The authors present a very detailed uncertainty analysis regarding space lidar observations of dust and the potential to separate the fine-mode and the coarse-mode fraction of the dust aerosol. The paper is rather long, and I had the feeling it should be shortened. On the other hand, AMT should cover such long and detailed papers dealing with quality assurance studies. I have some minor remarks, only.

The authors would like to thank the referee for his review for his time and for the interesting and at the same time substantial comments and suggestions. We tried, and did our best, to incorporate the proposed changes and corrections in the revised manuscript, aiming at improving the presented paper. Following, our responses may be found, addressing one by one to the comments raised.

Kind regards,

Emmanouil Proestakis

1. p1, l37: Improve wording: …the quality of the dust products is justified … What do you want to say? The quality can be good, excellent, poor e.t.c… but what means …. justified?

   The authors agree with the reviewer that the word justified was confusing and not proper for the manuscript, thus the sentence was re-written in order to be clearer, as follows.
   From:
   > "The quality of the dust products is justified by using AERONET fine-mode and coarse-mode aerosol optical thickness (AOT) interpolated to 532 nm and AERosol properties – Dust (AER-D) campaign airborne in-situ particle size distributions (PSDs) as reference datasets, during atmospheric conditions characterized by dust presence".

   To:
   > "The quality of the CALIPSO-based fine-mode and coarse-mode dust products is assessed through the use of AERONET fine-mode and coarse-mode aerosol optical thickness (AOT) interpolated to 532 nm and AERosol properties – Dust (AER-D) campaign airborne in-situ particle size distributions (PSDs) as reference datasets, during atmospheric conditions characterized by dust presence".

2. p2, l9: The research community prefers to denote 'ice nuclei' meanwhile 'ice-nucleating particles (INPs)'.

   Following the reviewer's comment the terminology is adapted accordingly.

3. p2, l32: life cycle.

   Following the reviewer's comment the sentence is corrected.

4. p2, l33: One should cite the lidar paper on dust devils and convective plumes of Ansmann et al., Tellus, 2009, in this context.

   We agree with the reviewer that the manuscript would be improved and enriched by including the recommended reference. The following reference is added in the manuscript: "Ansmann, A., Tesche, M., Groß, S., Freudenthaler, V., Seifert, P., Hiebsch, A., Schmidt, J., Wandinger, U., Mattis, I., Müller, D. and Wiegner, M.: The 16 April 2010 major volcanic ash plume over central Europe: EARLINET lidar and AERONET photometer observations at Leipzig and Munich, Germany, Geophysical Research Letters, 37(13), doi:10.1029/2010GL043809, 2010.".

5.  p3, l22: … remote sensing of the atmospheric aerosols. The technique is able……

    The sentence is modified according to the reviewer's recommendation.

6.  p4, l36: I would add recent modeling studies of Saito et al. Please check whether these authors also provide information on size (fine, coarse) dependence of the dust depolarization ratio.

    Study is included in the manuscript.

7.  p5, l3: Again, what do you mean? … Section 3 provides justification of the validity…

    As mentioned in Comment #1, the use of the word justified was confusing and not proper for the manuscript. The sentence is re-written as follows.
    From:
    "Section 3 provides justification of the validity of the fine-mode and coarse-mode components of pure-dust, on the basis of long-term AERONET observations (Sect. 3.1) and airborne in-situ measurements (Sect. 3.2) as reference datasets.".
    To:
    "In Section 3 the quality of the CALIPSO-based fine-mode and coarse-mode pure-dust products is addressed, against long-term AERONET observations (Sect. 3.1) and airborne in-situ measurements (Sect. 3.2) as reference datasets.".

8.  p5-p18: Section 2 is rather long… Are all the mentioned details needed?

    The authors have tried to cover the pre-processing of the datasets and the application of the two-step POLIPHON technique to CALIPSO optical products as comprehensively as possible. Detailing of the method and how it is transferred to CALIPSO optical products, including detailing the assumptions and limitations, is crucial in order to ensure transparency and reproducibility of the outcomes and also to allow for further advances in the future. Though we have done our best towards a robust discussion in this section several aspects would need even more in-depth discussion, since they are inherently complex. Towards this challenge the manuscript aims to provide in every discussed aspect an extended and comprehensive reference list aiming to point towards a broad range of high-level references of core and related scientific studies and resources to facilitate further exploitation of the applications, and at the same time to give credit to the research community.

9.  p8, l 27: The definition of the fine and coarse mode (< and > 1 µm in diameter) should be already given in the introduction, as early as possible.

    As commented by the reviewer, the definition of the fine- and coarse- modes of dust as sub- and supper-micrometer in terms of diameter is provided as early as possible in the manuscript, and more specifically, in the first two lines of the "Abstract Section": "A new four-dimensional, multiyear, and near-global climate data record of the fine-mode (submicrometer in terms of diameter) and coarse-mode (supermicrometer in terms of diameter) components of atmospheric pure-dust, is presented." (Pg. 1 - lines 20-21 of with-track-changes manuscript version); and in the "Introduction Section": "Overarching objective of the present study consists of the separation of the pure-dust submicrometer (fine-mode) and supermicrometer (coarse-mode) components of the dust aerosol load, in order to provide an accurate near-global and multiyear description of (1) the temporal distribution, (2) the three-dimensional spatial and vertical distribution, and (3) the seasonal and spatial transition of fine/coarse-mode dust transport pathways in terms of range, height and intensity." (Pg. 4 - lines 39-43 of with-track-changes manuscript version).

10. p10, l5: Please check the two papers of Hofer et al., ACP 2020a, b (Tajikistan lidar observations), and the Hu et al. ACP paper in 2020 (Taklamakan lidar observations).

We would like to thank the reviewer for reminding us the Hofer et al. 2020a/b studies, with which the study is enriched. Hofer et al. (2020a/b), on the basis of vertically resolved long-term lidar aerosol measurements conducted in the framework of the Central Asian Dust Experiment (CADEX) at Dushanbe, Tajikistan (March 2015-August 2016) provided dust optical properties at 532 nm, including among others, extinction-to-backscatter ratios (lidar ratios), linear depolarization ratios, and backscatter- and extinction-related Ångström. More specifically, Hofer et al. (2020a/b) reported dust lidar ratios at 532 nm accumulating around 35–40 sr and typical dust depolarization ratios of 0.30–0.35 at 532 nm. In addition, Hu et al. (2020) reported characterization of Taklamakan dust optical properties, on the basis of dust aerosol observations conducted in April 2019 in Kashi, China. More specifically, Hu et al. (2020) reported dust lidar ratios at 532 nm accumulating around $45 \pm 7$ sr and particle linear depolarization ratios about $0.36 \pm 0.05$ at 532 nm. In the present study, we use dust lidar ratios of $40 \pm 5$ sr and of $46 \pm 7$ sr for Central Asia and for East Asia, respectively, and a global mean particle linear depolarization ratio of $0.31 \pm 0.04$ at 532 nm. Thus, the dust lidar ratio and particulate depolarization ratio values at 532 nm used in the present study fall well within the dust lidar ratios and particulate depolarization ratios values at 532 nm reported by Hofer et al. (2020a/b) and Hu et al. (2020). We would like to inform the reviewer that future release version of the CALIPSO-based fine- and coarse- modes of pure-dust will incorporate the DeLiAn (Floutsi et al., 2023) reported values, thus will be further benefited by the statistical values of dust lidar ratio and particulate depolarization ratio values at 532 nm reported therein, including also the reported values by Hofer et al. (2020a/b) and Hu et al. (2020).
The following studies are included in manuscript:
- Hofer, J., Ansmann, A., Althausen, D., Engelmann, R., Baars, H., Abdullaev, S. F., and Makhmudov, A. N.: Long-term profiling of aerosol light extinction, particle mass, cloud condensation nuclei, and ice-nucleating particle concentration over Dushanbe, Tajikistan, in Central Asia, Atmos. Chem. Phys., 20, 4695–4711, https://doi.org/10.5194/acp-20-4695-2020, 2020a.
- Hofer, J., Ansmann, A., Althausen, D., Engelmann, R., Baars, H., Fomba, K. W., Wandinger, U., Abdullaev, S. F., and Makhmudov, A. N.: Optical properties of Central Asian aerosol relevant for spaceborne lidar applications and aerosol typing at 355 and 532 nm, Atmos. Chem. Phys., 20, 9265–9280, https://doi.org/10.5194/acp-20-9265-2020, 2020b.
- Hu, Q., Wang, H., Goloub, P., Li, Z., Veselovskii, I., Podvin, T., Li, K., and Korenskiy, M.: The characterization of Taklamakan dust properties using a multiwavelength Raman polarization lidar in Kashi, China, Atmos. Chem. Phys., 20, 13817–13834, https://doi.org/10.5194/acp-20-13817-2020, 2020.

11. p12, l10: Mineral particles with diameters >62.5 µm are no longer dust particles. The size of 62.5µm defines the minimum diameter of the sand particle range.

The discussion on mineral dust particles and the corresponding classification in terms of size as provided in the present study is of high importance, since the definition/terminology of the fine- and coarse-modes of dust varies extensively in the literature and between different groups and organizations, since mineral dust particles suspended into the atmosphere - in terms of diameter size - span from less than 0.1 µm to more than 100 µm in diameter. However, with respect to dust particles with diameter > 62.5 µm and the corresponding classification as sand particles, the terminology is not characterized by such inconsistencies in the literature. As mentioned by the reviewer, in general giant dust particles of diameter > 62.5 µm are considered sand-sized particles (Adebiyi et al., 2023 and references therein). Thus, the reviewer's comment is included in the manuscript. The sentence is modified
From:

"To this end, Adebiyi et al., (2023) reviewed related dust size distribution studies and proposed a uniform classification for atmospheric dust particles, including the fine, coarse, super-coarse, and giant dust classes, with dust separation boundary geometric diameters of 2.5, 10, and 62.5 μm, respectively."

To

"To this end, Adebiyi et al., (2023) reviewed related dust size distribution studies and proposed a uniform classification for atmospheric dust particles, including the fine, coarse, super-coarse, and giant (sand-sized) dust classes, with dust separation boundary geometric diameters of 2.5, 10, and 62.5 μm, respectively."

12. p14, Figure 3 is rather small.

We agree with the reviewer that the size of Figure 3 was rather small. The figure size is increased.

[Figure]

Figure 3: Illustration of the regional classification of pure-dust LR values applied in the present study. Domain abbreviations are provided in Table 3.

13. p14, Table 3, please check and add: Hofer et al. 2020b on depol and lidar ratios, and Hu et al. 2020.

According to the reviewer's suggestion the manuscript is enriched by the following studies, included in Table 3.

1) Hofer, J., Ansmann, A., Althausen, D., Engelmann, R., Baars, H., Abdullaev, S. F., and Makhmudov, A. N.: Long-term profiling of aerosol light extinction, particle mass, cloud condensation nuclei, and ice-nucleating particle concentration over Dushanbe, Tajikistan, in Central Asia, Atmos. Chem. Phys., 20, 4695–4711, https://doi.org/10.5194/acp-20-4695-2020, 2020a.

2) Hofer, J., Ansmann, A., Althausen, D., Engelmann, R., Baars, H., Fomba, K. W., Wandinger, U., Abdullaev, S. F., and Makhmudov, A. N.: Optical properties of Central Asian aerosol relevant for spaceborne lidar applications and aerosol typing at 355 and 532 nm, Atmos. Chem. Phys., 20, 9265–9280, https://doi.org/10.5194/acp-20-9265-2020, 2020b.

3) Hu, Q., Wang, H., Goloub, P., Li, Z., Veselovskii, I., Podvin, T., Li, K., and Korenskiy, M.: The characterization of Taklamakan dust properties using a multiwavelength Raman polarization lidar in Kashi, China, Atmos. Chem. Phys., 20, 13817–13834, https://doi.org/10.5194/acp-20-13817-2020, 2020.

Table 3: Overview of pure-dust lidar-based LR values (sr) classified under specific geographical regions of interest (Fig.3).

| Region | LR 532 nm [sr] | References | LR 532 nm [sr] |
|---|---|---|---|
| Western - Central Sahara Desert (W-C SD) North Atlantic Ocean (NAO) | 56 ± 8 | Tesche et al. (2009) | 56 ± 5 |
| | | Gross et al. (2011a) | 63 ± 6 |
| | | Gross et al. (2011b) | 62 ± 5 |
| | | Tesche et al. (2011) | 54 ± 10 |
| | | Kanitz et al. (2013) | 55 ± 5 |
| | | Kanitz et al. (2014) | 50 ± 5 |
| | | Gross et al. (2015) | 56 ± 7 |
| | | Weinzierl et al. (2016) | 55 ± 5 |
| | | Haaring et al. (2017) | 55 ± 5 |
| | | Rittmeister et al. (2017) | 55 ± 5 |
| | | Bolhmann et al. (2018) | 53 ± 2 |
| Eastern Sahara Desert (ESD) | 53 ± 6 | Nisantzi et al. (2015) | 41 ± 4 |
| | | Ansmann et al. (2019) | 50 ± 10 |
| Middle East Arabian Peninsula Central Asia (ME/AP/CA) | 40 ± 5 | Müller et al. (2007) | 38 ± 5 |
| | | Mamouri et al. (2013) | 36.4 ± 5.9 |
| | | Nisantzi et al. (2015) | 41 ± 4 |
| | | Hofer et al. (2017) | 39.3 ± 3.6 |
| | | Filioglou et al. (2020) | 42 ± 5 |
| | | Hofer et al. (2020a/b) | 37.7±2.1 |
| South and East Asia (S-EA) North Pacific Ocean (NPO) | 46 ± 7 | Liu et al. (2002) | 51 ± 9 |
| | | Sakai et al. (2002) | 46 ± 5 |
| | | Anderson et al. (2003) | 44 ± 8 |
| | | Sakai et al. (2003) | 47 ± 18 |
| | | Murayama et al. (2003) | 46.5 ± 10.5 |
| | | Murayama et al. (2004) | 56 ± 8 |
| | | Noh et al. (2007) | 51 ± 6 |
| | | Tesche et al. (2007) | 40 ± 5 |
| | | Noh et al. (2008) | 51 ± 6 |
| | | Jin et al. (2010) | 42 ± 3 |
| | | Hu et al., (2020) | 45 ± 7 |
| Europe (EU) | 56 ± 8 | Mattis et al. (2002) | 60 ± 10 |
| | | Ansmann et al. (2003) | 60 ± 20 |
| | | Müller et al. (2007) | 57 ± 2 |
| | | Guerrero-Rascado et al. (2009) | 57.5 ±7.5 |
| | | Papayiannis et al. (2011) | 59 ± 11 |
| | | Preißler et al. (2011) | 53 ± 7 |
| | | Wiegner et al. (2011) | 59 ± 6 |
| | | Preißler et al. (2013) | 56 ± 8 |
| | | Soupiona et al. (2019) | 64 ± 6 |
| North America (NA) | 49 ± 9 | Burton et al. (2012) | 49 ± 9 |
| South America (SA) | 42 ± 17 | Kanitz et al. (2013) | 42 ± 17 |

14. Section 3: Consistency check.

Following the reviewer's comment the title is modified.

15. Figure 5: This is one of the key figures. Therefore, the panels must be larger. I suggest to have two top panels (a, b), then two center panels (e, f), and finally two bottom panels (d, c).

According to the reviewer's comment, the figure is changed as follows:

[Figure]

Figure 5: Major Saharan dust outbreak moving westwards over Dakar on the 20[th] of April, 2017 at ~14:43 UTC, including the CALIPSO overpass in the proximity of the AERONET-Dakar station (red line) and FLEXPART 6-day back-trajectories at 2 km at the area of interest (Lat: 14.39, Lon: -16.95) denoting the Saharan desert origin of the advected air masses (yellow line) (Fig.5a). CALIPSO L2 5 km backscatter coefficient at 532 nm cross section (Fig.5b). Column-integrated particle volume concentration as a function of particle radius observed with the AERONET sunphotometer over Dakar,

Senegal (Fig.5c). Backscatter coefficient at 532 nm profiles of the coarse-mode pure-dust (red shaded area), fine-mode pure-dust (blue shaded area), and non-dust (gray shaded area) components of the total aerosol load (Fig.5d). Particulate depolarization ratio at 532 nm profile used for the decoupling of the coarse-mode pure-dust, fine-mode pure-dust, and non-dust components of the total aerosol load, as provided in Fig.5d (Fig.5e). Extinction coefficient at 532 nm profiles of the coarse-mode pure-dust (red shaded area) and fine-mode pure-dust (blue shaded area) components of the total aerosol load (Fig.5f).

16. Figure 6: My spontaneous question was: Who is right? AERONET or CALIOP? The correlation line in the left panel (a) does not provide a clear answer. The agreement between AERONET and CALIOP (scattered data cloud) is good. And, one should not forget: There is always a non-dust fine-mode contribution to the AERONET AOT. The right panel shows good agreement as well when keeping the impact of the uncertainty in the lidar ratio assumption into consideration.

Sect. 3.1 aims to evaluate the CALIPSO-based $DOD_{coarse}$ and $DOD_{fine}$ at 532 nm products through implementation of AERONET $AOT_{coarse}$ and $AOT_{fine}$ retrievals at 532 nm. As mentioned by the reviewer, the agreement between AERONET and CALIOP (scattered data cloud) is good. More specifically, the CALIPSO-based DOD–AERONET AOT evaluation intercomparison corroborates on (1) the good performance of the one-step and two-step POLIPHON techniques developed with the objective of decoupling the fine-mode, coarse-mode, and total pure-dust components of the total aerosol load (Mamouri and Ansmann, 2014; 2017; Tesche et al., 2009), (2) the high quality of the ESA-LIVAS pure-dust Climate Data Record (Amiridis et al., 2013; 2015; Marinou et al., 2017; Proestakis et al., 2018), and (3) the quality of the established CALIPSO-based products of fine-mode and coarse-mode pure-dust atmospheric components, in terms of extinction coefficient profiles and DODs at 532 nm. However, as mentioned by the reviewer, there are several aspects, assumptions, and limitations that are included and discussed when intercomparison/validation/evaluation activities between satellite-based and ground-based systems are performed. In the case of the CALIPSO-based DOD–AERONET AOT evaluation intercomparison, one of the significant aspects propagating in the statistical analysis is the non-dust components included in AERONET $AOT_{coarse}$ and $AOT_{fine}$ retrievals at 532 nm. More specifically, as mentioned by the reviewer and also discussed in the manuscript, the apparent underestimation features relate to an extend to the evaluation of $DOD_{fine}$ with $AOT_f$, an AERONET retrievals including non-dust fine-mode aerosol subtypes (e.g., biomass burning and urban haze), and to the evaluation of $DOD_{coarse}$ with $AOT_c$, an AERONET product including non-dust coarse-mode aerosol subtypes (e.g., marine, pollen, volcanic ash) - (Pg. 25 - lines 5-8 of with-track-changes manuscript version). To reduce the impact on the evaluation of these non-dust components, the CALIPSO-based DOD–AERONET AOT evaluation intercomparison is performed in quality-assured cases characterized by high presence of atmospheric dust. More specifically, to facilitate the evaluation of the CALIPSO-based $DOD_{coarse}$ and $DOD_{fine}$ datasets with AERONET $AOT_{coarse}$ and $AOT_{fine}$ retrievals, only for cases of dust presence in the atmosphere. Towards this objective the pure-dust product developed through the one-step POLIHPON applied to CALIPSO optical products - the ESA-LIVAS pure-dust CDR (Amiridis et al., 2013; 2015) - is implemented. Moreover, it should be noted that the overall revealed CALIPSO fine/coarse-mode DOD at 532 nm underestimation constitutes an expected feature. It is well-documented that an underestimation exists in CALIPSO extinction coefficient profiles and AODs at 532 nm, established against correlative observations by Aqua-MODIS AODs (e.g., Kacenelenbogen et al., 2011; Kittaka et al., 2011; Redemann et al., 2012; Kim et al., 2013; Ma et al., 2013), AERONET-derived AOTs (i.e., Schuster et al., 2012; Omar et al., 2013; Amiridis et al., 2013; Toth et al., 2018), and airborne lidar observations (Rogers et al., 2014). These studies attribute the CALIPSO AOD at 532 nm underestimation to (1) misclassification of cloud layers as aerosol (and vice versa), (2) erroneous sub-classification of the classified atmospheric layers as aerosol, (3) incorrect selection of the aerosol subtype lidar ratios, (4) limited or restricted penetration/attenuation of CALIOP beam within thick aerosol layers, and (5) profiles frequently populated with retrieval fill values (RFVs) due to failure to detect diffuse and tenuous aerosol layers of SNR below CALIOP minimum detection thresholds.

However, there are also additional reasons, assumptions, and limitations impacting the outcomes, including not only the above-mentioned CALIPSO-related factors, but also factors that are related to AERONET, assumptions and limitations of the POLIPHON technique (i.e. mean particulate depolarization ratios of the assumed aerosol subtypes), and in addition the lidar ratio assumptions which are taken into consideration, as mentioned by the reviewer. This is the reason why, towards enhancing the quality-testing of the dust products, the authors have performed the activities related to the total, fine-mode, and coarse-mode pure-dust mass concentration products' validation against airborne in-situ measurements (Sect. 3.2).

17. Figure 8: The 1064 nm particle or volume depol ratio is quite high. Above 30%! Other observations at 1064nm did not show that, except observations (e.g. of Burton et al.) directly in the dust source region. Also, in the marine boundary layer the depol ratio is likewise high. It is summer (August) and the marine layer should not be contaminated with dust so much. Did you check the quality of the CATS depolarization measurement?

In the case of the spatially and temporarily highly coordinated FAAM b920 – ISS-CATS underflight on the 7$^{th}$ of August 2015 in the proximity of Praia-Cabo Verde, a particularly dense Saharan Dust layer over the North Atlantic Ocean was probed. CATS, being a satellite-based lidar system at 1064 nm, is characterized by challenges upon the retrieval of aerosol/cloud optical properties different in comparison to ground-based lidar systems, which their high quality/maintenance/and manual operation makes them ideal to be used towards evaluation of the optical products and of the algorithms' performance as reference systems (i.e., EARLINET/Proestakis et al., 2019, PollyNET/Rebecca et al., 2019).
It is documented that at night, CATS V3-00 median 1064 nm ATB relative uncertainty is 7% in cloud/aerosol layers, while CATS V3-00 median daytime, such in the case of FAAM b920 – ISS-CATS underflight, 1064 nm ATB relative uncertainty is as high as 21% in cloud/aerosol layers (Rebecca et al., 2019). Moreover, coincident flights with the airborne Cloud Physics Lidar (CPL) instrument (McGill et al., 2002) showed that CATS data agree to within 25% with CPL. In addition, CATS ATB has been compared against ground-based EARLINET/PollyNET systems and found to be within 20%, and against CALIOP 1064 nm and found within ~18%. The L1B ATB uncertainties are mentioned here since the linear volume total depolarization ratio is defined as the ratio of perpendicular total (Rayleigh plus particle) backscatter to parallel total backscatter. Moreover, the mentioned uncertainties in L1B are crucial as L2 uncertainties of optical products are generally higher. In the framework of the present study, towards decoupling the pure-dust fine- and coarse- modes from the total aerosol load the L2 optical products are used (i.e., profiles of particulate depolarization ratio and backscatter coefficient) and not the L1B (i.e., Total Attenuated Backscatter and Volume Depolarization Ratio), at 532 nm and at 1064 nm, for CALIOP and CATS respectively, along the CALIPSO and ISS orbit tracks. With respect to pre-processing of the satellite-based products, quality assurance criteria reported in the literature and suggested in ATBDs are enforced, described, and discussed for CALIPSO-CALIOP in Sect.2.1.1 and for ISS-CATS in Sect.2.1.2. In general, the satellite-based lidar systems yield higher variability in optical products compared to ground-based systems, however reasonable considering the high challenges and the unique characteristics of the satellite systems.
To demonstrate the good performance and consistency of CALIOP and CATS backscatter coefficients and particulate depolarization ratios, keeping in mind the above-mentioned reported uncertainties in the literature, an indicative case of CALIPSO and CATS is provided in the following Figure. More specifically, the following Figure provides an indicative example of CALIOP at 532nm nm and CATS 1064nm observations of the same dust event, on the 3$^{rd}$ of February 2017, within 50 km and 30 minutes spatial and temporal distance collocation criteria respectively, between the CALIPSO and the ISS overpasses (upper Figure). The second line of the Figure provides CALIPSO (left) total backscatter coefficient at 532 nm and CATS (right) total backscatter coefficient at 1064 nm. The third line of the following Figure provides the CALIPSO (left) particulate depolarization ratio at 532 nm and the CATS (right) particulate depolarization ratio at 1064 nm. Finally, the lower part of the Figure provides the

CALIOP 532 and 1064 nm against CATS 1064 nm intercomparison, in terms of Total Backscatter Coefficient (left), Depolarization Ratio (center), and Extinction Coefficient (right).

[Figure]

Figure: Indicative example of CALIOP at 532nm nm and CATS 1064nm observations of the same dust event, on the 3rd of February 2017, within 50 km and 30 minutes spatial and temporal distance collocation criteria respectively, between the CALIPSO and the ISS overpasses (upper Figure). The second line of the Figure provides CALIPSO (left) total backscatter coefficient at 532 nm and CATS (right) total backscatter coefficient at 1064 nm. The third line of the Figure provides CALIPSO (left) particulate depolarization ratio at 532 nm and CATS (right) particulate depolarization ratio at 1064 nm. Finally, the lower part of the Figure provides the CALIOP 532 and 1064 nm against CATS 1064 nm intercomparison, in terms of Total Backscatter Coefficient (left), Depolarization Ratio (center), and Extinction Coefficient (right).

Furthermore, to provide to the reviewer an overview of the significantly higher variability in terms of particulate depolarization ratios at 1064 nm in satellite-based EO, the following Figure provides the entire CATS daytime (left panel) and nighttime (right panel) aerosol-classified quality-assured particulate depolarization ratios at 1064 nm along the ISS flight track, starting from 10th of February 2015 until the 30th of October 2017, when the system suffered an unrecoverable fault.

[Figure]

Figure: Distributions of CATS daytime (left panel) and nighttime (right panel) aerosol classified quality-assured particulate depolarization ratios at 1064 nm along the ISS flight track starting from 10th of February 2015 until the 30th of October 2017.

One reason for the extended manuscript, as mentioned by the reviewer, is the challenges the authors had to face when detailing all the challenges, assumptions, and limitations arising from applying the two-step POLIPHON to satellite lidar-based EO. Here, though the comment of the reviewer and the present discussion evolves around ISS-CATS, fluctuations are also apparent to CALIOP optical products. The following Figure provides CALIOP nighttime (left panel) aerosol-classified quality-assured particulate depolarization ratios at 532 nm along the CALIPSO flight track, for 2010, over the North Atlantic Ocean domain. Prior applying the POLIPHON technique to CALIOP optical products, both the backscatter coefficient at 523 nm and the particulate depolarization ratio at 532 nm undergo a ±1 bin vertical smoothing (only in the case of QA aerosol features). The effect of this approach is presented in the right panel, where the two predominant aerosol subtypes over the North Atlantic Ocean, the marine and the dust aerosol subtypes, in terms of particulate depolarization ratio at 532 nm, are revealed.

[Figure]

Figure: Distributions of CALIOP nighttime aerosol classified quality-assured particulate depolarization ratios at 532 nm along the CALIPSO flight track over the North Atlantic Ocean domain for 2010 (left panel) and following a ±1 bin vertical smoothing (right panel) prior applying the POLIPHON techniques.

18. p25, l5, I cannot find Eqs. (13) and (14)!

   Numbering in the equations is corrected.

19. p25, l10: Eq. (10) and (11) are now Eq.(9a) to (9c)?

   Numbering in the equations is corrected.

20. Figure 9: The in-situ observations in Fig. 9d show a pronounced fine mode, in contrast to the lidar fine-mode observations. Maybe this is an indication for the dominating anthropogenic pollution from Africa. The in-situ observations provide the total fine mode mass concentration. Is there an option to even distinguish fine dust from fine-mode aerosol pollution in the in-situ observation?
   *and*
21. p26, l15-l31: Again, the discussion of the fine mode aerosol is difficult. It remains tricky to find a good solution for the different contributions of fine-mode pollution and fine-mode dust to the overall fine mode aerosol fraction.

   As detailed in the well-established one-step and two-step family of POLIPHON the strong advantage of the methods is that they allow decupling of a highly depolarizing aerosol component from a low depolarizing aerosol component assuming external aerosol mixtures, in terms of backscatter coefficients. Further decoupling most probably is very challenging. On the basis of multi-wavelength Raman ground-based lidar systems accompanied by concurrent and collocated sunphotometer measurements may provide the capacity under a number of assumptions to further decouple the fine-mode pollution and fine-mode dust components from the total aerosol load. Possible support and implementation of atmospheric aerosol reanalysis models may be of high value towards this objective. However, CALIOP was a dual-wavelength polarization-sensitive elastic backscatter lidar system, capable of transmitting linear polarized light pulses at 532 and 1064 nm, and performing range-resolved measurements of the backscattered signals by atmospheric features, and specifically, of the parallel and perpendicular components of the backscattered photons at 532 nm with respect to the polarization plane of CALIOP emitted beam, and the total backscatter intensity at 1064 nm (Winker et al., 2009).

Implementation of the 1064 nm towards extracting the Color Ratio is challenging, due to significant fluctuations in the corresponding 1064 nm optical products. Moreover, implementation of the three channel (8.65, 10.6, 12.05 µm) Imaging Infrared Radiometer (IIR; Garnier et al., 2017) onboard CALIPSO may also be not a choice offering critical advantages, since IIR is less sensitive to smaller particles in comparison to larger particles. Even in the case of decoupling the fine-mode dust and the fine-mode non dust aerosol components, the challenge of the aerosol subtype classification would have to be overcome in order to apply suitable lidar ratios and compute the profiles of extinction coefficient from the backscatter coefficient profiles, without taking into consideration the size-dependance and wavelength-dependance of lidar ratios of different aerosol subtypes. Moreover, distinguish of the fine dust from fine-mode aerosol pollution in the in-situ observation, is not an option in this case.

The present version (V4) of CALIPSO algorithm, in terms of aerosol-subtype classification, utilizes (1) altitude, (2) location, (3) surface type, and (3) estimated particulate depolarization ratio, and (4) the integrated attenuated backscatter ($\gamma'$) to identify the aerosol subtype (Omar et al., 2009; Kim et al., 2018), while in comparison to the previous version (V3) the integrated attenuated color ratio (1064-to-532nm) is not used for aerosol subtyping due to the low signal-to-noise ratio (SNR), especially in the daytime, making it an inconsistent discriminator among tropospheric aerosol types (Kim et al., 2018). The final classification results to atmospheric features categorized as "tropospheric aerosol" between "marine", "dust", "polluted continental/smoke", "clean continental", "polluted dust", "elevated smoke" and "dusty marine" (Omar et al., 2009; Kim et al., 2018), and in the case of "stratospheric aerosol" between "PSC aerosol", "volcanic ash", and "sulfate/other" (Kar et al., 2019). Assuming due to the implementation of depolarization measurements in the aerosol-subtype classification frequently correct identification of atmospheric aerosol layers between as dusty and non-dusty, the fine-mode and coarse-mode dust separation technique is applied only to the "dust", "polluted dust" and "dusty marine" aerosol subtypes, while the "marine", "polluted continental/smoke", "clean continental", and "elevated smoke" atmospheric layers are neglected, as non-dusty.

With respect to the highly coordinated FAAM b920 – ISS-CATS underflight on the 7th of August 2015 in the proximity of Praia-Cabo Verde, a particularly dense Saharan Dust layer over the North Atlantic Ocean was probed. In general, the export of dust layers entrained into the atmosphere over the Saharan Desert westwards and the corresponding transport across the broader Atlantic Ocean is largely controlled by the prevailing wind systems and the regional meteorology, shaping the major dust transport pathways (Adams et al., 2012; Ben-Ami et al., 2012; Amiridis et al., 2013; Marinou et al., 2017; H. Yu et al., 2019; Proestakis et al., 2024), largely modulating the westwards atmospheric transport of Saharan dust layers by the seasonal latitudinal migration of the Intertropical Convergence Zone (ITCZ; Schneider et al., 2014). Due to the regional meteorology, the broader area over Cabo-Verde, where the FAAM b920 research flight took place, is largely affected by dust and marine aerosol subtypes, while non-dust and non-marine dust aerosol subtypes are rarely present.

This is corroborated here below by fourteen years of CALIPSO observations over the area of FAAM b920 research flight. More specifically, the following Figure provides all CALIPSO overpasses within an area of ~700 km from Cabo Verde (red color overpasses: daytime/blue color: nighttime overpasses, the fourteen years (~14 yrs) long-term (2007-2021) mean particulate depolarization ratio at 532 nm, the feature type classified atmospheric layers, and the aerosol subtype classification of the atmospheric layers classified as aerosol (Kim et al., 2018; Kar et al., 2018).

According to the long-term observations, the presence of non-dust and non-marine aerosol subtypes is rare in the area, thus further decoupling of the fine-mode dust and fine-mode non-dust would not have a significant impact in the present region of the globe. However, such advances in terms of algorithms would be extremely interesting upon development and implementation towards long-term near global climate data records, upon implementation to satellite-based EO.

CALIPSO-Cabo Verde overpasses

[Figure]

Particulate Depolarization Ratio at 532 nm - mean

[Figure]

CALIPSO Feature-Type - Total

[Figure]

CALIPSO Aerosol-Subtype - Total

[Figure]

Figure ##: CALIPSO overpasses within an area of ~700 km from Cabo Verde (red color overpasses: daytime/blue color: nighttime overpasses), the fourteen years (~14 yrs) long-term (2007-2021) mean particulate depolarization ratio at 532 nm, the feature type classified atmospheric layers, and the aerosol subtype classification of the atmospheric layers classified as aerosol.

22. Section 4: Three-dimensional distribution …

Following the reviewer's comment the title is modified.

23. p31, l25: India is meanwhile the most polluted sub-continent. So, again it will be complicated to get a clear picture in terms of coarse mode dust, fine-mode dust, and dominating fine-mode anthropogenic pollution. The separation of coarse mode dust is the most straightforward approach. But then the discussion becomes quite speculative.

The authors agree with the reviewer on the complexity of the broader area of the Indian subcontinent and the challenges related to applying the two-step POLIPHON to satellite-based EOs over the region, with the overarching objective of decoupling the coarse mode dust and fine-mode dust from the total

aerosol load. According to the authors experience, the most robust, extensively used, and validated/evaluated version of the family of the POLIPHON technique is the one-step POLIPHON (since less assumptions are required), which is applied in the ESA-LIVAS resulting to extracting the pure-dust component from the total aerosol load. Moreover, the authors agree with the reviewer that the separation of the coarse-mode dust is the most straightforward approach as a following step. Based on the above-mentioned points, the fine-mode component here is computed by subtracting from the total pure-dust the coarse-mode pure-dust component. Following separation of the components, the interpretation and discussion, under all the discussion on the strengths, assumptions, and limitation of applying POLIPHON to CALIPSO, is still challenging, resulting to the extended manuscript, as mentioned by the reviewer.

24. p31, l42: Why should only the coarse-mode dust fraction show a wave-like seasonal cycle? The fine-mode dust fraction will show the same, I guess.

The Indian subcontinent is affected in terms of long-range dust transport by the major sources of great Arabian, Dasht-e Lut, and Thar located to the west of the peninsula (Hamidi et al., 2013; Proestakis et al., 2018; Gkikas et al., 2022). The long-range transport over the area the takes place both in the Free Troposphere and within the PBL. The long-range transport is more pronounced during the pre-monsoon season (Dey et al., 2004, Vinoj et al., 2014), transport which is characterized by high seasonality, as revealed by both the supermicrometer and the submicrometer atmospheric pure-dust products, shown in Fig.12c and Fig.12d, respectively. This seasonality in the coarse-mode pure-dust product is revealed as an almost clear wave-like seasonal variation mainly in the Free Troposphere. This is in line with the reviewer's comment, that both the coarse-mode dust fraction and the fine-mode dust fraction should show a wave-like seasonal cycle. More specifically, this common characteristic is stated following correction of the manuscript in the present version (Pg. 34 - lines 14-15 of with-track-changes manuscript version). With respect to dust layers in the PBL, however, the seasonality although present is less apparent, since atmospheric dust is the net contribution of dust particles emitted and transported from areas outside the region of interest and dust particles emitted from within the region of interest, to an extend resulting from anthropogenic activity. As discussed in the manuscript, anthropogenic dust, involving transportation, infrastructure, building and road construction (Moulin and Chiapello, 2006; Chen et al., 2018), deterioration of extended soil surfaces and change in land use due to deforestation, grazing (Ginoux et al., 2012), urbanization, and agriculture (Tegen et al 1996) is estimated to contribute significant to the total atmospheric dust burden, with studies reporting an extended range, between ~25% of the total dust load (Ginoux et al.; 2012) and as high as ~70% of total dust load (Huang et al., 2015; Chen et al., 2019). Anthropogenic dust emission, though significant, lacks the strength of the atmospheric mechanisms developed over the arid and hyper arid regions of the globe (Koch and Renno, 2005, Knippertz et al., 2007; Ansmann et al., 2009; Klose et al., 2010; Fiedler et al., 2013), thus are expected to penetrate PBL and be released into the FT less frequently, resulting in increased complexity with respect to atmospheric aerosol features observed within the PBL than the clear wavelike characteristics observed in the FT.

25. p32, l3. Why should long range transport over the continent take place in the free troposphere, only, and not in the PBL as well? Over the oceans, this seems to be the case (dust transport in the free troposphere above the colder marine boundary layer). But over the continents? There is ONE dust layer, to my opinion, from the surface up to probably 3-4 km height (up to the PBL height over subtropical India). Figure 12c corroborates my opinion.

We agree with the reviewer long-range transport over the continent takes place both in the FT and within the PBL. In line with the reviewer's comment, this expected feature is mentioned in the manuscript (Pg. 34 - lines 14-15 of with-track-changes manuscript version).

26. Figure 12e vs 12f is confusing! There is a relatively high coarse-mode dust fraction above 2 km height, but not a high fine-mode fraction. What is the reason for this contrast? And the opposite is found for the near surface heights (up to 1.5 to 2 km). The fine-mode anthropogenic aerosol should clearly dominate in the highly polluted boundary layer.

The reviewer's comment is related to the previous comments, thus discussed therein. In addition, in terms of the present comment, the authors agree with the reviewer that Figures 12e and 12f were confusing to a reader, thus the corresponding Figures and the corresponding discussion are removed from the revised manuscript.

27. p33, l9-l29: Keeping my question just mentioned in mind, I find the discussion quite speculative. Please, provide a bit more careful, more hypothetical debate.

The section of the manuscript mentioned by the reviewer is modified in order to be less speculative, as follows.
From:

[revised manuscript text omitted]